# Differentially Private Clustering: Tight Approximation Ratios

**Badih Ghazi**
Google Research
Mountain View, CA
badihghazi@gmail.com

**Ravi Kumar**
Google Research
Mountain View, CA
ravi.k53@gmail.com

**Pasin Manurangsi**
Google Research
Mountain View, CA
pasin@google.com

## Abstract

We study the task of differentially private clustering. For several basic clustering problems, including Euclidean DensestBall, 1-Cluster, $k$-means, and $k$-median, we give efficient differentially private algorithms that achieve essentially the *same* approximation ratios as those that can be obtained by any non-private algorithm, while incurring only small additive errors. This improves upon existing efficient algorithms that only achieve some large constant approximation factors.

Our results also imply an improved algorithm for the Sample and Aggregate privacy framework. Furthermore, we show that one of the tools used in our 1-Cluster algorithm can be employed to get a faster quantum algorithm for ClosestPair in a moderate number of dimensions.

## 1 Introduction

With the significant increase in data collection, serious concerns about user privacy have emerged. This has stimulated research on formalizing and guaranteeing strong privacy protections for user-sensitive information. Differential Privacy (DP) [DMNS06, DKM+06] is a rigorous mathematical concept for studying user privacy and has been widely adopted in practice [EPK14, Sha14, Gre16, App17, DKY17, Abo18]. Informally, the notion of privacy is that the algorithm's output (or output distribution) should be mostly unchanged when any one of its inputs is changed. DP is quantified by two parameters $\epsilon$ and $\delta$; the resulting notion is referred to as pure-DP when $\delta = 0$, and approximate-DP when $\delta > 0$. See Section 2 for formal definitions of DP and [DR14, Vad17] for an overview.

Clustering is a central primitive in unsupervised machine learning [XW08, AC13]. An algorithm for clustering in the DP model informally means that the cluster centers (or the distribution on cluster centers) output by the algorithm should be mostly unchanged when any one of the input points is changed. Many real-world applications involve clustering sensitive data. Motivated by these, a long line of work has studied clustering algorithms in the DP model [BDMN05, NRS07, FFKN09, GLM+10, MTS+12, WWS15, NSV16, NCBN16, SCL+16, FXZR17, BDL+17, NS18, HL18, NCBN16, NS18, SK18, Ste20]. In this work we focus on several basic clustering problems in the DP model and obtain efficient algorithms with tight approximation ratios.

**Clustering Formulations.** The input to all our problems is a set $X$ of $n$ points, each contained in the $d$-dimensional unit ball. There are many different formulations of clustering. In the popular $k$-means problem [Llo82], the goal is to find $k$ centers minimizing the clustering cost, which is the sum of squared distances from each point to its closest center. The $k$-median problem is similar to $k$-means except that the distances are not squared in the definition of the clustering cost.[1] Both problems are NP-hard, and there is a large body of work dedicated to determining the best possible

| Reference | $w$ | $t$ | Running time |
|---|---|---|---|
| [NSV16], $\delta > 0$ | $O(\sqrt{\log n})$ | $O(\frac{\sqrt{d}}{\epsilon} \cdot \text{poly} \log \frac{1}{\delta})$ | $\text{poly}(n, d, \log \frac{1}{r})$ |
| [NS18], $\delta > 0$ | $O(1)$ | $\tilde{O}_{\epsilon,\delta}(\frac{\sqrt{d}}{\epsilon} \cdot n^{0.1} \cdot \text{poly} \log \frac{1}{\delta})$ | $\text{poly}(n, d, \log \frac{1}{r})$ |
| Exp. Mech. [MT07], $\delta = 0$ | $1 + \alpha$ | $O_\alpha(\frac{d}{\epsilon} \cdot \log \frac{1}{r})$ | $O\left(\left(\frac{1}{\alpha r}\right)^d\right)$ |
| Theorem 6, $\delta = 0$ | $1 + \alpha$ | $O_\alpha\left(\frac{d}{\epsilon} \cdot \log\left(\frac{d}{r}\right)\right)$ | $(nd)^{O_\alpha(1)} \text{poly} \log \frac{1}{r}$ |
| Theorem 6, $\delta > 0$ | $1 + \alpha$ | $O_\alpha\left(\frac{\sqrt{d}}{\epsilon} \cdot \text{poly} \log\left(\frac{nd}{\epsilon\delta}\right)\right)$ | $(nd)^{O_\alpha(1)} \text{poly} \log \frac{1}{r}$ |

Table 1: Comparison of $(\epsilon, \delta)$-DP algorithms for $(w, t)$-approximations for DensestBall given $r$.

approximation ratios achievable in polynomial time (e.g. [Bar96, CCGG98, CGTS02, JV01, JMS02, AGK+04, KMN+04, AV07, LS16, ACKS15, BPR+17, LSW17, ANSW17, CK19]), although the answers remain elusive. We consider approximation algorithms for both these problems in the DP model, where a $(w, t)$-approximation algorithm outputs a cluster whose cost is at most the sum of $t$ and $w$ times the optimum; we refer to $w$ as the *approximation ratio* and $t$ as the *additive error*. It is important that $t$ is small since without this constraint, the problem could become trivial. (Note also that without privacy constraints, approximation algorithms typically work with $t = 0$.)

We also study two even more basic clustering primitives, DensestBall and 1-Cluster, in the DP model. These underlie several of our results.

**Definition 1** (DensestBall)**.** *Given $r > 0$, a $(w, t)$-approximation for the* DensestBall *problem is a ball $B$ of radius $w \cdot r$ such that whenever there is a ball of radius $r$ that contains at least $T$ input points, $B$ contains at least $T - t$ input points.*

This problem is NP-hard for $w = 1$ [BS00, BES02, She15]. Moreover, approximating the largest number of points within any ball of radius of $r$ and up some constant factor is also NP-hard [BES02]. On the other hand, several polynomial-time approximation algorithms achieving $(1 + \alpha, 0)$-approximation for any $\alpha > 0$ are known [AHPV05, She13, BES02].

DensestBall is a useful primitive since a DP algorithm for it allows one to "peel off" one important cluster at a time. This approach has played a pivotal role in a recent fruitful line of research that obtains DP approximation algorithms for $k$-means and $k$-median [SK18, Ste20].

The 1-Cluster problem studied, e.g., in [NSV16, NS18] is the "inverse" of DensestBall, where instead of the radius $r$, the target number $T$ of points inside the ball is given. Without DP constraints, the computational complexities of these two problems are essentially the same (up to logarithmic factors in the number of points and the input universe size), as we may use binary search on $r$ to convert a DensestBall algorithm into one for 1-Cluster, and vice versa.[2] These two problems are generalizations of the MinimumEnclosingBall (aka MinimumBoundingSphere) problem, which is well-studied in statistics, operations research, and computational geometry.

As we elaborate below, DensestBall and 1-Cluster are also related to other well-studied problems, such as learning halfspaces with a margin and the Sample and Aggregate framework [NRS07].

**Main Results.** A common highlight of most of our results is that for the problems we study, our algorithms run in polynomial time (in $n$ and $d$) and obtain tight approximation ratios. Previous work sacrificed one of these, i.e., either ran in polynomial time but produced sub-optimal approximation ratios or took time exponential in $d$ to guarantee tight approximation ratios.

(i) For DensestBall, we obtain for any $\alpha > 0$, a pure-DP $(1 + \alpha, \tilde{O}_\alpha(\frac{d}{\epsilon}))$-approximation algorithm and an approximate-DP $(1 + \alpha, \tilde{O}_\alpha(\frac{\sqrt{d}}{\epsilon}))$-approximation algorithm.[3] The runtime of our algorithms is $\text{poly}(nd)$. Table 1 shows our results compared to previous work. To solve DensestBall with DP,

we introduce and solve two problems: efficient list-decodable covers and private sparse selection. These could be of independent interest.

(ii) For 1-Cluster, informally, we obtain for any $\alpha > 0$, a pure-DP $(1 + \alpha, \tilde{O}_\alpha(\frac{d}{\epsilon}))$-approximation algorithm running in time $(nd)^{O_\alpha(1)}$. We also obtain an approximate-DP $(1 + \alpha, \tilde{O}_\alpha(\frac{\sqrt{d}}{\epsilon}))$-approximation algorithm running in time $(nd)^{O_\alpha(1)}$. The latter is an improvement over the previous work of [NS18] who obtain an $(\tilde{O}(1 + \frac{1}{\phi}), \tilde{O}_{\epsilon,\delta}(n^\phi\sqrt{d}))$-approximation. In particular, they do not get an approximation ratio $w$ arbitrarily close to 1. Even worse, the exponent $\phi$ in the additive error $t$ can be made close to 0 only at the expense of blowing up $w$. Our algorithm for 1-Cluster follows by applying our DP algorithm for DensestBall, along with "DP binary search" similarly to [NSV16].

(iii) For $k$-means and $k$-median, we prove that we can take any (not necessarily private) approximation algorithm and convert it to a DP clustering algorithm with essentially the same approximation ratio, and with small additive error and small increase in runtime. More precisely, given any $w^*$-approximation algorithm for $k$-means (resp., $k$-median), we obtain a pure-DP $(w^*(1 + \alpha), \tilde{O}_\alpha(\frac{kd+k^{O_\alpha(1)}}{\epsilon}))$-approximation algorithm and an approximate-DP $(w^*(1 + \alpha), \tilde{O}_\alpha(\frac{k\sqrt{d}+k^{O_\alpha(1)}}{\epsilon}))$-approximation algorithm for $k$-means (resp., $k$-median). (The current best known non-private approximation algorithms achieve $w^* = 6.358$ for $k$-means and $w^* = 2.633$ for $k$-median [ANSW17].) Our algorithms run in time polynomial in $n$, $d$ and $k$, and improve on those of [SK18] who only obtained some large constant factor approximation ratio independent of $w^*$.

It is known that $w^*$ can be made arbitrarily close to 1 for (non-private) $k$-means and $k$-median if we allow fixed parameter tractable[4] algorithms [BHPI02, DLVKKR03, KSS04, KSS05, Che06, FMS07, FL11]. Using this, we get a pure-DP $(1 + \alpha, \tilde{O}_\alpha(\frac{kd+k^2}{\epsilon}))$-approximation, and an approximate-DP $(1 + \alpha, \tilde{O}_\alpha(\frac{k\sqrt{d}+k^2}{\epsilon}))$-approximation. The algorithms run in time $2^{O_\alpha(k \log k)}\mathrm{poly}(nd)$.

**Overview of the Framework.** All of our DP clustering algorithms follow this three-step recipe:

(i) Dimensionality reduction: we randomly project the input points to a low dimension.

(ii) Cluster(s) identification in low dimension: we devise a DP clustering algorithm in the low-dimensional space for the problem of interest, which results in cluster(s) of input points.

(iii) Cluster center finding in original dimension: for each cluster found in step (ii), we privately compute a center in the original high-dimensional space minimizing the desired cost.

**Applications.** Our DP algorithms for 1-Cluster imply better algorithms for the Sample and Aggregate framework of [NRS07]. Using a reduction from 1-Cluster due to [NSV16], we get an algorithm that privately outputs a stable point with a radius not larger than the optimal radius than by a $1 + \alpha$ factor, where $\alpha$ is an arbitrary positive constant. For more context, please see Section 5.2.

Moreover, by combining our DP algorithm for DensestBall with a reduction of [BS00, BES02], we obtain an efficient DP algorithm for agnostic learning of halfspaces with a constant margin. Note that this result was already known from the work of Nguyen et al. [NUZ20]; we simply give an alternative proof that employs our DensestBall algorithm as a blackbox. For more on this and related work, please see Section 5.3.

Finally, we provide an application of one of our observations outside of DP. In particular, we give a faster (randomized) history-independent data structure for dynamically maintaining ClosestPair in a moderate number of dimensions. This in turn implies a faster *quantum* algorithm for ClosestPair in a similar setting of parameters.

**Organization.** Section 2 contains background on DP and clustering. Our algorithms for DensestBall are presented in Section 3, and those for $k$-means and $k$-median are given in Section 4. Applications to 1-Cluster, Sample and Aggregate, agnostic learning of halfspaces with a margin, and ClosestPair are described in Section 5. We conclude with some open questions in Section 6. All missing proofs are deferred to the Appendix.

## 2 Preliminaries

**Notation.** For a finite universe $\mathcal{U}$ and $\ell \in \mathbb{N}$, we let $\binom{\mathcal{U}}{\leq \ell}$ be the set of all subsets of $\mathcal{U}$ of size at most $\ell$. Let $[n] = \{1, \ldots, n\}$. For $v \in \mathbb{R}^d$ and $r \in \mathbb{R}_{\geq 0}$, let $\mathcal{B}(v, r)$ be the ball of radius $r$ centered at $v$. For $\kappa \in \mathbb{R}_{\geq 0}$, denote by $\mathbb{B}_\kappa^d$ the quantized $d$-dimensional unit ball with discretization step $\kappa$.[5] We throughout consider closed balls.

**Differential Privacy (DP).** We next recall the definition and basic properties of DP. Datasets $\mathbf{X}$ and $\mathbf{X}'$ are said to be neighbors if $\mathbf{X}'$ results from removing or adding a single data point from $\mathbf{X}$.[6]

**Definition 2** (Differential Privacy (DP) [DMNS06, DKM+06]). *Let $\epsilon, \delta \in \mathbb{R}_{\geq 0}$ and $n \in \mathbb{N}$. A randomized algorithm $\mathcal{A}$ taking as input a dataset is said to be $(\epsilon, \delta)$-differentially private if for any two neighboring datasets $\mathbf{X}$ and $\mathbf{X}'$, and for any subset $S$ of outputs of $\mathcal{A}$, it holds that $\Pr[\mathcal{A}(\mathbf{X}) \in S] \leq e^\epsilon \cdot \Pr[\mathcal{A}(\mathbf{X}') \in S] + \delta$. If $\delta = 0$, then $\mathcal{A}$ is said to be $\epsilon$-differentially private.*

We assume throughout that $0 < \epsilon \leq O(1)$, $0 < \alpha < 1$, and when used, $\delta > 0$.

**Clustering.** Since many of the proof components are common to the analyses of $k$-means and $k$-median, we will use the following notion, which generalizes both problems.

**Definition 3** (($k, p$)-Clustering). *Given $k \in \mathbb{N}$ and a multiset $\mathbf{X} = \{x_1, \ldots, x_n\}$ of points in the unit ball, we wish to find $k$ centers $c_1, \ldots, c_k \in \mathbb{R}^d$ minimizing $\mathrm{cost}_\mathbf{X}^p(c_1, \ldots, c_k) := \sum_{i \in [n]} \left( \min_{j \in [k]} \|x_i - c_j\| \right)^p$. Let $\mathrm{OPT}_\mathbf{X}^{p,k}$ denote[7] $\min_{c_1, \ldots, c_k \in \mathbb{R}^d} \mathrm{cost}_\mathbf{X}^p(c_1, \ldots, c_k)$. A $(w, t)$-approximation algorithm for $(k, p)$-Clustering outputs $c_1, \ldots, c_k$ such that $\mathrm{cost}_\mathbf{X}^p(c_1, \ldots, c_k) \leq w \cdot \mathrm{OPT}_\mathbf{X}^{p,k} + t$. When $\mathbf{X}$, $p$, and $k$ are unambiguous, we drop the subscripts and superscripts.*

Note that $(k, 1)$-Clustering and $(k, 2)$-Clustering correspond to $k$-median and $k$-means respectively. It will also be useful to consider the *Discrete $(k, p)$-Clustering* problem, which is the same as in Definition 3, except that we are given a set $\mathcal{C}$ of "candidate centers" and we can only choose the centers from $\mathcal{C}$. We use $\mathrm{OPT}_\mathbf{X}^{p,k}(\mathcal{C})$ to denote $\min_{c_{i_1}, \ldots, c_{i_k} \in \mathcal{C}} \mathrm{cost}_\mathbf{X}^p(c_{i_1}, \ldots, c_{i_k})$.

**Centroid Sets and Coresets.** A centroid set is a set of candidate centers such that the optimum does not increase by much even when we restrict the centers to belong to this set.

**Definition 4** (Centroid Set [Mat00]). *For $w, t > 0, p \geq 1$, $k, d \in \mathbb{N}$, a set $\mathcal{C} \subseteq \mathbb{R}^d$ is a $(p, k, w, t)$-centroid set of $\mathbf{X} \subseteq \mathbb{R}^d$ if $\mathrm{OPT}_\mathbf{X}^{p,k}(\mathcal{C}) \leq w \cdot \mathrm{OPT}_\mathbf{X}^{p,k} + t$. When $k$ and $p$ are unambiguous, we simply say that $\mathcal{C}$ is a $(w, t)$-centroid set of $\mathbf{X}$.*

A coreset is a (multi)set of points such that, for any possible $k$ centers, the cost of $(k, p)$-Clustering of the original set is roughly the same as that of the coreset (e.g., [HM04]).

**Definition 5** (Coreset). *For $\gamma, t > 0, p \geq 1, k \in \mathbb{N}$, a set $\mathbf{X}'$ is a $(p, k, \gamma, t)$-coreset of $\mathbf{X} \subseteq \mathbb{R}^d$ if for every $\mathcal{C} = \{c_1, \ldots, c_k\} \subseteq \mathbb{R}^d$, we have $(1 - \gamma) \cdot \mathrm{cost}_\mathbf{X}^p(\mathcal{C}) - t \leq \mathrm{cost}_{\mathbf{X}'}^p(\mathcal{C}) \leq (1 + \gamma) \cdot \mathrm{cost}_\mathbf{X}^p(\mathcal{C}) + t$. When $k$ and $p$ are unambiguous, we simply say that $\mathbf{X}'$ is a $(\gamma, t)$-coreset of $\mathbf{X}$.*

## 3 Private DensestBall

In this section, we obtain pure-DP and approximate-DP algorithms for DensestBall.

**Theorem 6.** *There is an $\epsilon$-DP (resp., $(\epsilon, \delta)$-DP) algorithm that runs in time $(nd)^{O_\alpha(1)} \cdot$ poly $\log(1/r)$ and, w.p.*[8]  *0.99, returns a $\left(1 + \alpha, O_\alpha\left(\frac{d}{\epsilon} \cdot \log\left(\frac{d}{r}\right)\right)\right)$-approximation (resp., $\left(1 + \alpha, O_\alpha\left(\frac{\sqrt{d}}{\epsilon} \cdot \text{poly} \log\left(\frac{nd}{\epsilon\delta}\right)\right)\right)$-approximation) for DensestBall.*

To prove this, we follow the three-step recipe from Section 1. Using the Johnson–Lindenstrauss (JL) lemma [JL84] together with the Kirszbraun Theorem [Kir34], we project the input to $O((\log n)/\alpha^2)$ dimensions in step (i). It turns out that step (iii) is similar to (ii), as we can repeatedly apply a low-dimensional DensestBall algorithm to find a center in the high-dimensional space. Therefore, the bulk of our technical work is in carrying out step (ii), i.e., finding an efficient, DP algorithm for DensestBall in $O((\log n)/\alpha^2)$ dimensions. We focus on this part in the rest of this section; the full proof with the rest of the arguments can be found in Appendix D.2.

## 3.1 A Private Algorithm in Low Dimensions

Having reduced the dimension to $d' = O((\log n)/\alpha^2)$ in step (i), we can afford an algorithm that runs in time $\exp(O_\alpha(d')) = n^{O_\alpha(1)}$. With this in mind, our algorithms in dimension $d'$ have the following guarantees:

**Theorem 7.** *There is an $\epsilon$-DP (resp., $(\epsilon, \delta)$-DP) algorithm that runs in time $(1 + 1/\alpha)^{O(d')}$poly $\log(1/r)$ and, w.p. 0.99, returns a $\left(1 + \alpha, O_\alpha\left(\frac{d'}{\epsilon} \log\left(\frac{1}{r}\right)\right)\right)$-approximation (resp., $\left(1 + \alpha, O_\alpha\left(\frac{d'}{\epsilon} \log\left(\frac{n}{\epsilon\delta}\right)\right)\right)$-approximation) for DensestBall.*

As the algorithms are allowed to run in time exponential in $d'$, Theorem 7 might seem easy to devise at first glance. Unfortunately, even the Exponential Mechanism [MT07], which is the only known algorithm achieving approximation ratio arbitrarily close to 1, still takes $\Theta_\alpha(1/r)^{d'}$ time, which is $\exp(\omega(d'))$ for $r = o(1)$. (In fact, in applications to $k$-means and $k$-median, we set $r$ to be as small as $1/n$, which would result in a running time of $n^{\Omega(\log n)}$.) To understand, and eventually overcome this barrier, we recall the implementation of the Exponential Mechanism for DensestBall:

- Consider any $(\alpha r)$-cover[9] $C$ of the unit ball $\mathcal{B}(0, 1)$.
- For every $c \in C$, let $score[c]$ be the number of input points lying inside $\mathcal{B}(c, (1 + \alpha)r)$.
- Output a point $c^* \in C$ with probability $\frac{e^{(\epsilon/2) \cdot score[c^*]}}{\sum_{c \in C} e^{(\epsilon/2) \cdot score[c]}}$.

By the generic analysis of the Exponential Mechanism [MT07], this algorithm is $\epsilon$-DP and achieves a $\left(1 + \alpha, O_\alpha\left(\frac{d'}{\epsilon} \log\left(\frac{1}{r}\right)\right)\right)$-approximation as in Theorem 7. The existence of an $(\alpha r)$-cover of size $\Theta\left(\frac{1}{\alpha r}\right)^{d'}$ is well-known and directly implies the $\Theta_\alpha(1/r)^{d'}$ running time stated above.

Our main technical contribution is to implement the Exponential Mechanism in $\Theta_\alpha(1)^{d'}$poly $\log \frac{1}{r}$ time instead of $\Theta_\alpha(1/r)^{d'}$. To elaborate on our approach, for each input point $x_i$, we define $S_i$ to be $C \cap \mathcal{B}(x_i, (1 + \alpha)r)$, i.e., the set of all points in the cover $C$ within distance $(1 + \alpha)r$ of $x_i$. Note that the score assigned by the Exponential Mechanism is $score[c] = \{i \in [n] \mid c \in S_i\}$, and our goal is to privately select $c^* \in C$ with as large a score as possible. Two main questions remain: (1) How do we find the $S_i$'s efficiently? (2) Given the $S_i$'s, how do we sample $c^*$? We address these in the following two subsections, respectively.

### 3.1.1 Efficiently List-Decodable Covers

In this section, we discuss how to find $S_i$ in time $(1 + 1/\alpha)^{O(d')}$. Motivated by works on error-correcting codes (see, e.g., [Gur06]), we introduce the notion of *list-decodability* for covers:

**Definition 8** (List-Decodable Cover). *A $\Delta$-cover is* list-decodable *at distance $\Delta' \geq \Delta$ with list size $\ell$ if for any $x \in \mathcal{B}(0, 1)$, we have that $|\{c \in C \mid \|c - x\| \leq \Delta'\}| \leq \ell$. Moreover, the cover is* efficiently list-decodable *if there is an algorithm that returns such a list in time* poly$(\ell, d', \log(1/\Delta))$.

We prove the existence of efficiently list-decodable covers with the following parameters:

**Lemma 9.** *For every $0 < \Delta < 1$, there exists a $\Delta$-cover $C_\Delta$ that is efficiently list-decodable at any distance $\Delta' \geq \Delta$ with list size $(1 + \Delta'/\Delta)^{O(d')}$.*

In this terminology, $S_i$ is exactly the decoded list at distance $\Delta' = (1 + \alpha)r$, where $\Delta = \alpha r$ in our cover $C$. As a result, we obtain the $(1 + 1/\alpha)^{O(r)}$ bound on the time for computing $S_i$, as desired.

The proof of Lemma 9 includes two tasks: (i) bounding the size of the list and (ii) coming up with an efficient decoding algorithm. It turns out that (i) is not too hard: if our cover is also an $\Omega(\Delta)$-packing[10], then a standard volume argument implies the bound in Lemma 9. However, carrying out (ii) is more challenging. To do so, we turn to lattice-based covers. A *lattice* is a set of points that can be written as an integer combination of some given basis vectors. Rogers [Rog59] (see also [Mic04]) constructed a family of lattices that are both $\Delta$-covers and $\Omega(\Delta)$-packings. Furthermore, known lattice algorithms for the so-called *Closest Vector Problem* [MV13] allow us to find a point $c \in C_\Delta$ that is closest to a given point $x$ in time $2^{O(d')}$. With some more work, we can "expand" from $c$ to get the entire list in time polynomial in $\ell$. This concludes the outline of our proof of Lemma 9.

### 3.1.2  SparseSelection

We now move to (2): given $S_i$'s, how to privately select $c^*$ with large $score[c^*] = |\{i \mid c^* \in S_i\}|$?

We formalize the problem as follows:

**Definition 10** (SparseSelection)**.** *For $\ell \in \mathbb{N}$, the input to the $\ell$-SparseSelection problem is a list $S_1, \ldots, S_n$ of subsets, where $S_1, \ldots, S_n \in \binom{C}{\leq \ell}$ for some finite universe $C$. An algorithm solves $\ell$-SparseSelection with additive error $t$ if it outputs a universe element $\hat{c} \in C$ such that $|\{i \mid \hat{c} \in S_i\}| \geq \max_{c \in C} |\{i \mid c \in S_i\}| - t$.*

| **Algorithm 1** |
| --- |
| 1: **procedure** DENSESTBALL $(x_1, \ldots, x_n; r, \alpha)$ |
| 2:      $C_{\alpha r} \leftarrow (\alpha r)$-cover from Lemma 9 |
| 3:      **for** $i \in [n]$ **do** |
| 4:          $S_i \leftarrow$ decoded list of $x$ at distance $(1 + \alpha)r$ with respect to $C_{\alpha r}$ |
| 5:      **return** SparseSelection$(S_1, \ldots, S_n)$ |

The crux of our SparseSelection algorithm is the following. Since $score[c^*] = 0$ for all $c^* \notin S_1 \cup \cdots \cup S_n$, to implement the Exponential Mechanism it suffices to first randomly select (with appropriate probability) whether we should sample from $S_1 \cup \cdots \cup S_n$ or uniformly from $C$. For the former, the sampling is efficient since $S_1 \cup \cdots \cup S_n$ is small. This gives the following for pure-DP:

**Lemma 11.** *Suppose there is a $\operatorname{poly} \log |C|$-time algorithm $\mathcal{O}$ that samples a random element of $C$ where each element of $C$ is output with probability at least $0.1/|C|$. Then, there is a $\operatorname{poly}(n, \ell, \log |C|)$-time $\epsilon$-DP algorithm that, with probability $0.99$, solves $\ell$-SparseSelection with additive error $O\left(\frac{1}{\epsilon} \cdot \log |C|\right)$.*

We remark that, in Lemma 11, we only require $\mathcal{O}$ to sample *approximately* uniformly from $C$. This is due to a technical reason that we only have such a sampler for the lattice covers we use. Nonetheless, the outline of the algorithm is still exactly the same as before.

For approximate-DP, it turns out that we can get rid of the dependency of $|C|$ in the additive error entirely, by adjusting the probability assigned to each of the two cases. In fact, for the second case, it even suffices to just output some symbol $\perp$ instead of sampling (approximately) uniformly from $C$. Hence, there is no need for a sampler for $C$ at all, and this gives us the following guarantees:

**Lemma 12.** *There is a $\operatorname{poly}(n, \ell, \log |C|)$-time $(\epsilon, \delta)$-DP algorithm that, with probability $0.99$, solves $\ell$-SparseSelection with additive error $O\left(\frac{1}{\epsilon} \log \left(\frac{n\ell}{\epsilon\delta}\right)\right)$.*

### 3.1.3  Putting Things Together

With the ingredients ready, the DensestBall algorithm is given in Algorithm 1. The pure- and approximate-DP algorithms for SparseSelection in Lemmas 11 and 12 lead to Theorem 7.

# 4   Private $k$-means and $k$-median

We next describe how we use our DensestBall algorithm along with additional ingredients adapted from previous studies of coresets to obtain DP approximation algorithms for $k$-means and $k$-median with nearly tight approximation ratios and small additive errors as stated next:

**Theorem 13.** *Assume there is a polynomial-time (not necessarily DP) algorithm for $k$-means (resp., $k$-median) in $\mathbb{R}^d$ with approximation ratio $w$. Then, there is an $\epsilon$-DP algorithm that runs in time $k^{O_\alpha(1)}\mathrm{poly}(nd)$ and, with probability $0.99$, produces a $\left(w(1+\alpha), O_{w,\alpha}\left(\left(\frac{kd+k^{O_\alpha(1)}}{\epsilon}\right)\mathrm{poly}\log n\right)\right)$-approximation for $k$-means (resp., $k$-median). Moreover, there is an $(\epsilon, \delta)$-DP algorithm with the same runtime and approximation ratio but with additive error $O_{w,\alpha}\left(\left(\frac{k\sqrt{d}}{\epsilon}\cdot\mathrm{poly}\log\left(\frac{k}{\delta}\right)\right)+\left(\frac{k^{O_\alpha(1)}}{\epsilon}\cdot\mathrm{poly}\log n\right)\right)$.*

To prove Theorem 13, as for DensestBall, we first reduce the dimension of the clustering instance from $d$ to $d' = O_\alpha(\log k)$, which can be done using the recent result of Makarychev et al. [MMR19]. Our task thus boils down to proving the following low-dimensional analogue of Theorem 13:

**Theorem 14.** *Under the same assumption as in Theorem 13, there is an $\epsilon$-DP algorithm that runs in time $2^{O_\alpha(d')}\mathrm{poly}(n)$ and, with probability $0.99$, produces a $\left(w(1+\alpha), O_{\alpha,w}\left(\frac{k^2\cdot2^{O_\alpha(d')}}{\epsilon}\mathrm{poly}\log n\right)\right)$-approximation for $k$-means (resp., $k$-median).*

We point out that it is crucial for us that the reduced dimension $d'$ is $O_\alpha(\log k)$ as opposed to $O_\alpha(\log n)$ (which is the bound from a generic application of the JL lemma), as otherwise the additive error in Theorem 14 would be $\mathrm{poly}(n)$, which is vacuous, instead of $\mathrm{poly}(k)$. We next proceed by (i) finding a "coarse" centroid set (satisfying Definition 4 with $w = O(1)$), (ii) turning the centroid set into a DP coreset (satisfying Definition 5 with $w = 1 + \alpha$), and (iii) running the non-private approximation algorithm as a black box. We describe these steps in more detail below.

## 4.1   Finding a Coarse Centroid Set via DensestBall

We consider geometrically increasing radii $r = 1/n, 2/n, 4/n, \ldots$. For each such $r$, we iteratively run our DensestBall algorithm $2k$ times, and for each returned center, remove all points within a distance of $8r$ from it. This yields $2k\log n$ candidate centers. We prove that they form a centroid set with a constant approximation ratio and a small additive error:

**Lemma 15.** *There is a polynomial time $\epsilon$-DP algorithm that, with probability $0.99$, outputs an $\left(O(1), O\left(\frac{k^2d'}{\epsilon}\mathrm{poly}\log n\right)\right)$-centroid set of size $2k\log n$ for $k$-means (resp., $k$-median).*

We point out that the solution to this step is not unique. For example, it is possible to run the DP $k$-means algorithm from [SK18] instead of Lemma 15. However, we choose to use our algorithm since its analysis works almost verbatim for both $k$-median and $k$-means, and it is simple.

## 4.2   Turning a Coarse Centroid Set into a Coreset

Once we have a coarse centroid set from the previous step, we follow the approach of Feldman et al. [FFKN09], which can turn the coarse centroid and eventually produce a DP coreset:

**Lemma 16.** *There is an $2^{O_\alpha(d')}\mathrm{poly}(n)$-time $\epsilon$-DP algorithm that, with probability $0.99$, produces an $\left(\alpha, O_\alpha\left(\frac{k^2\cdot2^{O_\alpha(d')}}{\epsilon}\mathrm{poly}\log n\right)\right)$-coreset for $k$-means (and $k$-median).*

Roughly speaking, the idea is to first "refine" the coarse centroid by constructing an *exponential cover* around each center $c$ from Lemma 15. Specifically, for each radius $r = 1/n, 2/n, 4/n, \ldots$, we consider all points in the $(\alpha r)$-cover of the ball of radius $r$ around $c$. Notice that the number of points in such a cover can be bounded by $2^{O_\alpha(d')}$. Taking the union over all such $c, r$, this result in a new *fine* centroid set of size $2^{O_\alpha(d')}\cdot\mathrm{poly}(k, \log n)$. Each input point is then snapped to the closet point in this set; these snapped points form a good coreset [HM04]. To make this coreset private, we add an appropriately calibrated noise to the number of input points snapped to each point in the fine centroid set. The additive error resulting from this step scales linearly with the size of the fine centroid set, which is $2^{O_\alpha(d')}\cdot\mathrm{poly}(k, \log n)$ as desired.

We note that, although our approach in this step is essentially the same as Feldman et al. [FFKN09], they only fully analyzed the algorithm for $k$-median and $d \leq 2$. Thus, we cannot use their result as a black box and hence, we provide a full proof that also works for $k$-means and for any $d > 0$ in Appendix C.

## 4.3 Finishing Steps

Finally, we can simply run the (not necessarily DP) approximation algorithm on the DP coreset from Lemma 16, which immediately yields Theorem 14.

## 5 Applications

Our DensestBall algorithms imply new results for other well-studied tasks, which we now describe.

### 5.1 1-Cluster

Recall the 1-Cluster problem from Section 1. As shown by [NSV16], a discretization of the inputs is necessary to guarantee a finite error with DP, so we assume that they lie in $\mathbb{B}_\kappa^d$. For this problem, they obtained an $O(\sqrt{\log n})$ approximation ratio, which was subsequently improved to some large constant by [NS18] albeit with an additive error that grows polynomially in $n$. Using our DensestBall algorithms we get a $1 + \alpha$ approximation ratio with additive error polylogarithmic in $n$:

**Theorem 17.** *For $0 < \kappa < 1$, there is an $\epsilon$-DP algorithm that runs in $(nd)^{O_\alpha(1)}\mathrm{poly}\log(\frac{1}{\kappa})$ time and with probability $0.99$, outputs a $\left(1 + \alpha, O_\alpha\left(\frac{d}{\epsilon}\mathrm{poly}\log\left(\frac{n}{\epsilon\kappa}\right)\right)\right)$-approximation for 1-Cluster. For any $\delta > 0$, there is an $(\epsilon, \delta)$-DP algorithm with the same runtime and approximation ratio but with additive error $O_\alpha\left(\frac{\sqrt{d}}{\epsilon} \cdot \mathrm{poly}\log\left(\frac{nd}{\epsilon\delta}\right)\right) + O\left(\frac{1}{\epsilon} \cdot \log(\frac{1}{\delta}) \cdot 9^{\log^*(d/\kappa)}\right)$.*

### 5.2 Sample and Aggregate

Consider functions $f : \mathcal{U}^* \to \mathbb{B}_\kappa^d$ mapping databases to the discretized unit ball. A basic technique in DP is *Sample and Aggregate* [NRS07], whose premise is that for large databases $S \in U^*$, evaluating $f$ on a random subsample of $S$ can give a good approximation to $f(S)$. This method enables bypassing worst-case sensitivity bounds in DP (see, e.g., [DR14]) and it captures basic machine learning primitives such as bagging [JYvdS19]. Concretely, a point $c \in \mathbb{B}_\kappa^d$ is an $(m, r, \zeta)$-*stable point* of $f$ on $S$ if $\Pr[\|f(S') - c\|_2 \leq r] \geq \zeta$ for $S'$ a database of $m$ i.i.d. samples from $S$. If such a point $c$ exists, $f$ is $(m, r, \zeta)$-stable on $S$, and $r$ is a radius of $c$. Via a reduction to 1-Cluster, [NSV16] find a stable point of radius within an $O(\sqrt{\log n})$ factor from the smallest possible while [NRS07] got an $O(\sqrt{d})$ approximation, and a constant factor is subsequently implied by [NS18]. Our 1-Cluster algorithm yields a $1 + \alpha$ approximation:

**Theorem 18.** *Let $d, m, n \in \mathbb{N}$ and $0 < \epsilon, \zeta, \alpha, \delta, \kappa < 1$ with $m \leq n$, $\epsilon \leq \frac{\zeta}{72}$ and $\delta \leq \frac{\epsilon}{300}$. There is an $(\epsilon, \delta)$-DP algorithm that takes $f : U^n \to \mathbb{B}_\kappa^d$ and parameters $m$, $\zeta$, $\epsilon$, $\delta$, runs in time $(\frac{nd}{m})^{O_\alpha(1)}\mathrm{poly}\log(\frac{1}{\kappa})$ plus the time for $O(\frac{n}{m})$ evaluations of $f$ on a dataset of size $m$, and whenever $f$ is $(m, r, \zeta)$-stable on $S$, with probability $0.99$, the algorithm outputs an $(m, (1 + \alpha)r, \frac{\zeta}{8})$-stable point of $f$ on $S$, provided that $n \geq m \cdot O_\alpha\left(\frac{\sqrt{d}}{\epsilon} \cdot \mathrm{poly}\log\left(\frac{nd}{\epsilon\delta}\right) + \frac{1}{\epsilon} \cdot \log(\frac{1}{\delta}) \cdot 9^{\log^*(d/\kappa)}\right)$.*

### 5.3 Agnostic Learning of Halfspaces with a Margin

We next apply our algorithms to the well-studied problem of agnostic learning of halfspaces with a margin (see, e.g., [BS00, BM02, McA03, SSS09, BS12, DKM19, DKM20]). Denote the error rate of a hypothesis $h$ on a distribution $D$ on labeled samples by $\mathrm{err}^D(h)$, and the $\mu$-margin error rate of halfspace $h_u(x) = \mathrm{sgn}(u \cdot x)$ on $D$ by $\mathrm{err}_\mu^D(u)$. (See Appendix G for precise definitions.) Furthermore, let $\mathrm{OPT}_\mu^D := \min_{u \in \mathbb{R}^d} \mathrm{err}_\mu^D(u)$. The problem of learning halfspaces with a margin in the agnostic PAC model [Hau92, KSS94] can be defined as follows.

**Definition 19.** *Let $d \in \mathbb{N}$ and $\mu, t \in \mathbb{R}^+$. An algorithm properly agnostically PAC learns halfspaces with margin $\mu$, error $t$ and sample complexity $m$, if given as input a training set*

$S = \{(x^{(i)}, y^{(i)})\}_{i=1}^{m}$ *of i.i.d. samples drawn from an unknown distribution $D$ on $\mathcal{B}(0,1) \times \{\pm 1\}$,* *it outputs a halfspace $h_u : \mathbb{R}^d \to \{\pm 1\}$ satisfying $\mathrm{err}^D(h_u) \leq \mathrm{OPT}_\mu^D + t$ with probability 0.99.*

Via a reduction of [BS00, BES02] from agnostic learning of halfspaces with a margin to DensestBall, we can use our DensestBall algorithm to derive the following:

**Theorem 20.** *For $0 < \mu, t < 1$, there is an $\epsilon$-DP algorithm that runs in time $\left(\frac{1}{\epsilon t}\right)^{O_\mu(1)} +$ $\mathrm{poly}\left(O_\mu\left(\frac{d}{\epsilon t}\right)\right)$, and with probability 0.99, properly agnostically learns halfspaces with margin $\mu$, error $t$, and sample complexity $O_\mu\left(\frac{1}{\epsilon t^2} \cdot \mathrm{poly} \log\left(\frac{1}{\epsilon t}\right)\right)$.*

We reiterate that this result can also be derived by an algorithm of Nguyen et al. [NUZ20]; we prove Theorem 20 here as it is a simple blackbox application of the DensestBall algorithm.

### 5.4 ClosestPair

Finally, we depart from the notion of DP and instead give an application of efficiently list-decodable covers to the ClosestPair problem:

**Definition 21** (ClosestPair). *Given points $x_1, \ldots, x_n \in \mathbb{Z}^d$, where each coordinate of $x_i$ is represented as an $L$-bit integer, and an integer $\xi \in \mathbb{Z}$, determine whether there exists $1 \leq i < j \leq n$ such that $\|x_i - x_j\|_2^2 \leq \xi$.*

In the dynamic setting of ClosestPair, we start with an empty set $S$ of points. At each step, a point maybe added to and removed[11] from $S$, and we have to answer whether there are two distinct points in $S$ whose squared Euclidean distance is at most $\xi$. Our main contribution is a faster history-independent data structure for dynamic ClosestPair. Recall that a deterministic data structure is said to be *history-independent* if, for any two sequences of updates that result in the same set of points, the states of the data structure must be the same in both cases. For a randomized data structure, we say that it is history-independent if, for any two sequences of updates that result in the same set of points, the distribution of the state of the data structure must be the same.

**Theorem 22.** *There is a history-independent randomized data structure for dynamic ClosestPair that supports up to $n$ updates, with each update takes $2^{O(d)}\mathrm{poly}(\log n, L)$ time, and uses $O(nd \cdot \mathrm{poly}(\log n, L))$ memory.*

We remark that the data structure is only randomized in terms of the layout of the memory (i.e., state), and that the correctness always holds. Our data structure improves that of Aaronson et al. [ACL+20], in which the running time per update operation is $d^{O(d)}\mathrm{poly}(\log n, L)$.

Aaronson et al. [ACL+20] show how to use their data structure together with quantum random walks from [MNRS11] (see also [Amb07, Sze04]) to provide a fast quantum algorithm for ClosestPair in low dimensions which runs in time $d^{O(d)}n^{2/3}\mathrm{poly}(\log n, L)$. With our improvement above, we immediately obtain a speed up in terms of the dependency on $d$ under the same model[12]:

**Corollary 23.** *There exists a quantum algorithm that solves (offline) ClosestPair with probability 0.99 in time $2^{O(d)}n^{2/3}\mathrm{poly}(\log n, L)$.*

## 6 Conclusion and Open Questions

In this work, we obtained tight approximation ratios for several fundamental DP clustering tasks. An interesting research direction is to study the smallest possible additive error for DP clustering while preserving the tight non-private approximation ratios that we achieve. Another important direction is to obtain practical implementations of DP clustering algorithms that could scale to large datasets with many clusters. We focused in this work on the Euclidean metric; it would also be interesting to extend our results to other metric spaces.

## Broader Impact

Our work lies in the active area of privacy and its broader impact should be interpreted in light of ongoing debates in academia and industry. The primary goal of our work is to develop efficient differentially private algorithms for clustering data, with quality approaching that of clustering algorithms that are indifferent to privacy.

Being able to cluster data without compromising privacy but with quality almost as good as without privacy considerations, we believe, has a few societal benefits. Firstly, it could compel applications that deal with sensitive data and that already use off-the-shelf clustering algorithms to switch to using private clustering since the quality losses of our algorithm are guaranteed to be minimal and our algorithms are only modestly more expensive to run. Secondly, since clustering is a fundamental primitive in machine learning and data analysis, our work can enable privacy in more intricate applications that depend on clustering. Thirdly, we believe our work can spur further research into making other private machine learning algorithms attain quality comparable to non-private ones. In other words, it can lead to the following state: preserving privacy does not entail a compromise in quality. This will have far-reaching effects on how researchers develop new methods.

On the other hand, there are possible negative consequences of our work. Since our work has not been tested in practice, it is conceivable that practitioners might be dissuaded from using it on their own. Further, there might be unintended or malicious applications of private clustering, where privacy might be used in a negative way; our work might become a latent enablers of such activity.

Overall we believe that protecting privacy is a net positive for the society and our work contribute towards this larger goal in a positive way.

## Acknowledgments and Disclosure of Funding

We are grateful to Noah Golowich for providing helpful comments on a previous draft. We also thank Nai-Hui Chia for useful discussions on the quantum ClosestPair problem.

## Footnotes

[1] For the formal definitions of $k$-means and $k$-median, see Definition 3 and the paragraph following it.

[2]To reduce from 1-Cluster to DensestBall, one can binary-search on the target radius. In this case, the number of iterations needed for the binary search depends logarithmically on the ratio between the maximum possible distance between two input points and the minimum possible distance between two (distinct) input points. In the other direction (i.e., reducing from DensestBall to 1-Cluster), one can binary-search on the number of points inside the optimal ball, and here the number of iterations will be logarithmic in the number of input points.

[3]The notation $\tilde{O}_x(\cdot)$ ignores factors involving $x$ and factors polylogarithmic in $n, d, \epsilon, \delta$.

[4] Recall that an algorithm is said to be *fixed parameter tractable* in $k$ if its running time is of the form $f(k) \cdot \mathrm{poly}(n)$ for some function $f$, and where $n$ is the input size [DF13].

[5]Whenever we assume that the inputs lie in $\mathbb{B}_\kappa^d$, our results will hold for any discretization as long as the minimum distance between two points as at least $\kappa$.

[6]This definition of DP is sometimes referred to as *removal DP*. Some works in the field consider the alternative notion of *replacement DP* where two datasets are considered neighbors if one results from modifying (instead of removing) a single data point of the other. We remark that $(\epsilon, \delta)$-removal DP implies $(2\epsilon, 2\delta)$-replacement DP. Thus, our results also hold (with the same asymptotic bounds) for the replacement DP notion.

[7]The cost is sometimes defined as the $(1/p)$th power.

[8]In the main body of the paper, we state error bounds that hold with probability 0.99. In the appendix, we extend all our bounds to hold with probability $1 - \beta$ for any $\beta > 0$, with a mild dependency on $\beta$ in the error.

[9]A $\zeta$-*cover* $C$ of $\mathcal{B}(0, 1)$ is a set of points such that for any $y \in \mathcal{B}(0, 1)$, there is $c \in C$ with $\|c - y\| \leq \zeta$.

[10]A *$\zeta$-packing* is a set of points such that each pairwise distance is at least $\zeta$.

[11]Throughout, we assume without loss of generality that $x$ must belong to $S$ before "remove $x$" can be invoked. To make the algorithm work when this assumption does not hold, we simply keep a history-independent data structure that can quickly answer whether $x$ belongs to $S$ [Amb07, BJLM13].

[12]The model assumes the presence of gates for random access to an $m$-qubit quantum memory that takes time only $\mathrm{poly}(\log m)$. As discussed in [Amb07], such an assumption is necessary even for element distinctness, which is an easier problem than ClosestPair.

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
