[Supplementary Material]

# Appendix

We give some further preliminaries in Section A. Our algorithms for DensestBall in low dimensions are given and analyzed in Section B, and those for $k$-means and $k$-median are presented in Section C. The resulting algorithms in high dimensions are obtained in Section D. Our results for 1-Cluster, Sample and Aggregate, agnostic learning of halfspaces with a margin, and ClosestPair are presented in Sections E, F, G, and H respectively.

## A    Additional Preliminaries

For any vector $v \in \mathbb{R}^d$, we denote by $\|v\|_2$ its $\ell_2$-norm, which is defined by $\|v\|_2 := \sqrt{\sum_{i=1}^d v_i^2}$; most of the times we simply use $\|v\|$ as a shorthand for $\|v\|_2$. For any positive real number $\lambda$, the Discrete Laplace distribution $\mathrm{DLap}(\lambda)$ is defined as $\mathrm{DLap}(k; \lambda) := \frac{1}{C(\lambda)} \cdot e^{-\frac{|k|}{\lambda}}$ for any $k \in \mathbb{Z}$, where $C(\lambda) := \sum_{k=-\infty}^{\infty} e^{-\frac{|k|}{\lambda}}$ is the normalization constant.

### A.1    Composition Theorems

We recall the "composition theorems" that allow us to easily keep track of privacy losses when running multiple algorithms on the same dataset.

**Theorem 24** (Basic Composition [DKM$^+$06])**.** *For any $\epsilon, \delta \geq 0$ and $k \in \mathbb{N}$, an algorithm that runs $k$ many $(\epsilon, \delta)$-DP algorithms (possibly adaptively) is $(k\epsilon, k\delta)$-DP.*

It is possible to get better bounds using the following theorem (albeit at the cost of adding a *positive* $\delta'$ parameter).

**Theorem 25** (Advanced Composition [DRV10])**.** *For any $\epsilon, \delta \geq 0, \delta' > 0$ and $k \in \mathbb{N}$, an algorithm that runs $k$ many $(\epsilon, \delta)$-DP algorithms (possibly adaptively) is $(2k\epsilon(e^\epsilon - 1) + \epsilon\sqrt{2k \ln \frac{1}{\delta'}}, k\delta + \delta')$-DP.*

For an extensive overview of DP, we refer the reader to [DR14, Vad17].

## B    DensestBall in Low Dimensions

In this section, we provide our algorithms for DensestBall in low dimensions, stated formally below. We start by stating our pure-DP algorithm.

**Theorem 26.** *For every $\epsilon > 0$ and $0 < \alpha \leq 1$, there is an $\epsilon$-DP algorithm that runs in time $(1 + 1/\alpha)^{O(d)}\mathrm{poly}\log(1/r)$ and, with probability $1 - \beta$, returns a $\left(1 + \alpha, O_\alpha\left(\frac{d}{\epsilon}\log\left(\frac{1}{\beta r}\right)\right)\right)$-approximation for DensestBall, for every $\beta > 0$.*

We next state our approximate-DP algorithm.

**Theorem 27.** *For every $\epsilon > 0$ and $0 < \delta, \alpha \leq 1$, there is an $(\epsilon, \delta)$-DP algorithm that runs in time $(1 + 1/\alpha)^{O(d)}\mathrm{poly}\log(1/r)$ and, with probability at least $1 - \beta$, returns a $\left(1 + \alpha, O_\alpha\left(\frac{d}{\epsilon}\log\left(\frac{n}{\min\{\epsilon,1\}\cdot\beta\delta}\right)\right)\right)$-approximation for DensestBall, for every $\beta > 0$.*

Notice that Theorems 26 and 27 imply Theorem 7 in Section 3.1. As discussed there, the main components of our algorithm are efficiently list-decodable covers and algorithms for the SparseSelection problem, which will be dealt with in the upcoming two subsections. Finally, in Section B.3, we put the ingredients together to obtain the DensestBall algorithms as stated in Theorems 26 and 27.

### B.1    List-Decodable Covers of the Unit Ball

We start by defining the notion of a $\Delta$-cover and its "list-decodable" variant.

**Definition 28.** *A $\Delta$-cover of the $d$-dimensional unit ball is a set $C \subseteq \mathbb{R}^d$ such that for every point $x$ in the unit ball, there exists $c \in C$ such that $\|c - x\| \leq \Delta$.*

*Furthermore, we say that a $\Delta$-cover is* list-decodable *at distance $\Delta' \geq \Delta$ with list size $\ell$ if, for any $x$ in the unit ball, we have that $|\{c \in C \mid \|c - x\| \leq \Delta'\}| \leq \ell$. Finally, if there is an algorithm that returns such a list in time $\mathrm{poly}(\ell, d, \log(1/\Delta))$, then we say that the cover is* efficiently list-decodable.

We will derive the existence of a certain family of efficiently list-decodable covers, which, as we argue next, can be done by combining tools from the literature on packings, coverings, and lattice algorithms. The properties of the family are stated below.

**Lemma 29.** *For every $0 < \Delta < 1$, there exists a $\Delta$-cover $C_\Delta$ that is efficiently list-decodable at any distance $\Delta' \geq \Delta$ with list size $O(1 + \Delta'/\Delta)^{O(d)}$.*

Furthermore, we will need to be able to quickly sample points from the cover, as stated next:

**Lemma 30.** *For every $0 < \Delta < 1$, there exists a $\mathrm{poly}(1/\Delta, 2^d)$-time algorithm $\mathcal{O}_\Delta$ that samples a random element from the cover $C_\Delta$ (given in Lemma 29) such that the probability that each element is output is at least $\frac{0.99}{|C_\Delta|}$.*

We prove Lemmas 29 and 30 in Subsections B.1.2 and B.1.3 respectively. Before doing so, we provide some additional preliminaries in Subsection B.1.1.

### B.1.1 Additional Preliminaries on Lattices

We start by defining lattices and related quantities that will be useful in our proofs. Interested readers may refer to surveys and books on the topic such as [MG12] for more background.

A *basis* is a set of linearly independent vectors. A *lattice* generated by a basis $B = \{b_1, \ldots, b_m\}$, denoted by $\mathcal{L}(\mathcal{B})$, is defined as the set $\{\sum_{i=1}^m a_i b_i \mid a_1, \ldots, a_m \in \mathbb{Z}\}$. The length of the shortest non-zero vector of a lattice $\mathcal{L}$ is denoted by $\lambda(\mathcal{L})$, i.e.,

$$\lambda(\mathcal{L}) := \min_{v \in \mathcal{L}, v \neq 0} \|v\|.$$

The *covering radius* of the lattice $\mathcal{L}(\mathcal{B})$ is defined as the smallest $r \in \mathbb{R}^+$ such that every point in $\mathbb{R}^d$ is within a distance of $r$ from some lattice point; more formally, the covering radius is

$$\mu(\mathcal{L}) := \inf \left\{ r \in \mathbb{R}^+ \mid \bigcup_{v \in \mathcal{L}} \mathcal{B}(v, r) = \mathbb{R}^d \right\}.$$

The *Voronoi cell* of a lattice $\mathcal{L}$ is denoted by $\mathcal{V}(\mathcal{L})$ and is defined as the set of points closer to 0 than to other points of the lattice, i.e.,

$$\mathcal{V}(\mathcal{L}) = \{y \in \mathbb{R}^d \mid \|y\| \leq \min_{v \in \mathcal{L}, v \neq 0} \|v - y\|\}.$$

It is known (see, e.g., [MV13]) that the Voronoi cell can also be defined as the intersection of at most $2(2^d - 1)$ halfspaces of the form $\{y \in \mathbb{R}^d \mid \|y\| \leq \|v - y\|\}$ for $v \in \mathcal{L}$. These vectors $v$ are said to be the *Voronoi relevant* vectors; we denote the set of Voronoi relevant vectors by $VR(\mathcal{L})$.

We will also use the following simple property of Voronoi relevant vectors. This fact is well-known but we include its proof for completeness.

**Observation 31.** *Let $v \in \mathcal{L}$ be a non-zero vector in the lattice. There exists a Voronoi relevant vector $v^* \in VR(\mathcal{L})$ such that $\|v - v^*\| < \|v\|$.*

*Proof.* Let $\eta > 0$ be the largest real number such that $\eta v \in \mathcal{V}(\mathcal{L})$. Notice that $\eta \leq 1/2$, as otherwise $\eta v$ is closer to $v$ than to 0. Moreover, $\eta v$ must lie on a facet of $\mathcal{V}(\mathcal{L})$; let $v^*$ be the Voronoi relevant vector corresponding to this facet. It is obvious that if $v^*$ is a multiple of $v$, then the claimed statement holds. Otherwise, we have

$$
\begin{aligned}
\|v - v^*\| &= \|(1 - \eta)v - (v^* - \eta v)\| \\
&\leq (1 - \eta)\|v\| + \|v^* - \eta v\| && \text{(triangle inequality)} \\
&\leq (1 - \eta)\|v\| + \|0 - \eta v\| && \text{(from definition of } v^*\text{)} \\
&= \|v\|.
\end{aligned}
$$

Moreover, since we assume that $v^*$ is not a multiple of $v$, the triangle inequality above must be a strict inequality. As a result, we must have $\|v - v^*\| < \|v\|$ as desired. $\qquad\square$

When $\mathcal{L}$ is clear from the context, we may drop it from the notations and simply write $\lambda, \mu, \mathcal{V}, VR$ instead of $\lambda(\mathcal{L}), \mu(\mathcal{L}), \mathcal{V}(\mathcal{L}), VR(\mathcal{L})$ respectively.

In the *Closest Vector Problem (CVP)*, we are given a target vector $v'$, and the goal is to find a vector $v \in \mathcal{L}(\mathcal{B})$ that is closest to $v'$ in the Euclidean metric (i.e., minimizes $\|v - v'\|$). It is known that this problem can be solved in time $2^{O(d)}$, as stated more precisely next.

**Theorem 32** ([MV13]). *There is a deterministic algorithm that takes a basis $B = \{b_1, \ldots, b_m\} \subseteq \mathbb{R}^d$ and a target vector $v' \in \mathbb{R}^d$ where each coordinate of these vectors has bit complexity $M$, and finds the closest vector to $v'$ in $\mathcal{L}(B)$ in time $\mathrm{poly}(M, 2^d)$. Furthermore, the set of Voronoi relevant vectors can be computed in the same time complexity.*

Note that there are faster *randomized* CVP algorithms [ADS15, AS18] that run in $2^{d+o(d)}\mathrm{poly}(M)$ time; we chose to employ the above algorithm, which is *deterministic*, for simplicity.

### B.1.2 Almost Perfect Lattices and Proof of Lemma 29

For completeness, we will prove Lemma 29 in this subsection. Many of the proof components are from [Mic04, Rog59]; in addition, we observe the efficient list-decodability. First, we have to define the notion of almost perfect lattices [Mic04], which are the lattices that are simultaneously good packings and coverings:

**Definition 33.** *Let $\tau \geq 1$. A lattice $\mathcal{L}$ is said to be $\tau$-perfect if $\mu(\mathcal{L})/\lambda(\mathcal{L}) \leq \tau/2$.*

It is known that $O(1)$-perfect lattices can be computed in $2^{O(d)}$-time[11].

**Theorem 34** ([Rog59, Mic04]). *There is an algorithm that, given $d \in \mathbb{N}$, runs in $2^{O(d)}$ time and outputs a basis $\mathcal{B} = \{b_1, \ldots, b_d\}$ such that $\mathcal{L}(\mathcal{B})$ is 3-perfect.*

With all the previous results stated, we can now easily prove Lemma 29.

*Proof of Lemma 29.* We use the algorithm from Theorem 34 to construct a basis $\mathcal{B} = \{b_1, \ldots, b_d\}$ that is 3-perfect. By scaling, we may assume that $\mu(\mathcal{L}(\mathcal{B})) \leq \Delta$ and $\lambda(\mathcal{L}(\mathcal{B})) \geq 2\Delta/3$. Our $\Delta$-cover is defined as $C_\Delta := \{v \in \mathcal{L}(\mathcal{B}) \mid \|v\| \leq 1 + \Delta\}$.

To list-decode at distance $\Delta'$, we first compute the set $R := \{v \in \mathcal{L}(\mathcal{B}) \mid \|v\| \leq \Delta' + \Delta\}$, as follows. We start from $R = \{0\}$. At each iteration, we go through all vectors $w$ in the current set $S$ and all Voronoi relevant vectors $v$; if $\|w + v\| \leq \Delta' + \Delta$, we add $w + v$ to $S$. We repeat this until no additional vectors are added to $S$. The correctness of the algorithm to construct $S$ follows from Observation 31. Furthermore, since the list of Voronoi relevant vectors can be computed in time $2^{O(d)}$ (Theorem 32), it is obvious that the algorithm runs in $\mathrm{poly}(|S|, 2^d)$. Now, from $\lambda(\mathcal{L}(\mathcal{B})) \geq 2\Delta/3$, $S$ is a $\Delta/3$-packing. As a result, by a standard volume argument, we have $|S| \leq O(1 + \Delta'/\Delta)^{O(d)}$. In other words, the running time of constructing $S$ is at most $O(1 + \Delta'/\Delta)^{O(d)}$ as desired.

Once we have constructed $S$, we can list-decode $x$ at distance $\Delta$ as follows. First, we use the CVP algorithm from Theorem 32 to find the closest vector $v \in \mathcal{L}(\mathcal{B})$ to it. Then, we consider $v + w$ for each $w \in S$; if $\|v + w - x\| \leq \Delta'$, we add $v + w$ into the list. Clearly, this step of the algorithm runs in time $2^{O(d)} + \mathrm{poly}(|S|) = O(1 + \Delta'/\Delta)^{O(d)}$, and this also constitutes the list size bound. Finally, the correctness of this step is also straightforward: for any vector $z \in \mathcal{L}(\mathcal{B})$ such that $\|z - x\| \leq \Delta'$, we must have $\|z - v\| \leq \|z - x\| + \|v - x\| \leq \Delta' + \Delta$, which means that it must be added to the list by our algorithm. $\square$

### B.1.3 Near-Uniform Sampler: Proof of Lemma 30

Finally, we give a proof of Lemma 30

*Proof of Lemma 30.* The algorithm repeats the following for $W = 100 \left(1 + 2\Delta\right)^d$ times: it samples a point $x$ uniformly at random from $\mathcal{B}(0, 1 + 2\Delta)$, uses the CVP algorithm from Theorem 32 to find

the closest lattice vector $v \in \mathcal{L}(\mathcal{B})$ to $x$ and, if $\|v\| \leq 1 + \Delta$, it returns $v$ and terminates. Otherwise, it returns 0.

First of all, notice that when the algorithm terminates within $W$ steps, it returns a point uniformly at random from the cover $C_\Delta$. Hence, we only have to show that the probability that it does not terminate within the first $W$ steps is at most 0.01. To see that this is the case, note that the algorithm always terminates if $\|x\| \leq 1$; in each iteration, this happens with probability $100/W$. Hence, the probability that this does not happen in the $W$ iterations is only $(1 - 100/W)^W \leq 0.01$. □

We remark that, if we never stop after $W$ iterations, then we would get an algorithm that has an expected running time of $O(W)$ and for which the output distribution is exactly uniform over $C_\Delta$. While the exact uniformity seems neat, it turns out that we do not need it anyway in the next section, which leads us to cut off after $W$ iterations so as to get a fixed upper bound on the running time.

## B.2 SparseSelection

In the Selection problem, each user $i$ receives a subset $S_i$ of some universe $\mathcal{U}$. The goal is to output an element $u \in \mathcal{U}$ that appears in a maximum number of the $S_i$'s. This problem is very well-studied in the DP literature, and tight bounds are known in a large regime of parameters both in the central [SU17, BU17] and the local [Ull18] models.

However, known algorithms [MT07, DNR+09][12] for Selection run in time $\Omega(|\mathcal{U}|)$ which can be large; specifically, this will be insufficient for our application to private clustering where $|\mathcal{U}|$ is super-polynomial. Instead, we will consider a restriction of the problem where we have an upper bound $\ell$ on the sizes of the $S_i$'s, and show that, under certain assumptions, we can solve Selection in this case with running time polynomial in $\ell$ and $\log |\mathcal{U}|$.

**Definition 35** (SparseSelection). *For a positive integer $\ell$, the input to the $\ell$-SparseSelection problem is a list $\mathbf{S} = (S_1, \ldots, S_n)$ of subsets, where $S_1, \ldots, S_n \in \binom{\mathcal{U}}{\leq \ell}$ for some finite universe $\mathcal{U}$. We say that an algorithm solves the $\ell$-SparseSelection problem with additive error $t$ if it outputs a universe element $\hat{u} \in U$ such that*

$$|\{i \mid \hat{u} \in S_i\}| \geq \max_{u \in \mathcal{U}} |\{i \mid u \in S_i\}| - t.$$

Throughout this section, we assume that each universe element of $\mathcal{U}$ can be represented by a poly $\log |\mathcal{U}|$-bit string, but that $\mathcal{U}$ itself is not explicitly known. (This is the case for lattice covers from the previous subsection, where each element of the covers can be represented by the coefficients.) We will give two simple $\mathrm{poly}(n, \ell, \log |\mathcal{U}|)$-time algorithms for the problem, both of which are variants of the Exponential Mechanism of McSherry and Talwar [MT07].

Our first algorithm is an approximate-DP algorithm with an additive error independent of the universe size $|\mathcal{U}|$; furthermore, this algorithm does not require any additional assumption.

**Lemma 36** (Approximate-DP Algorithm for SparseSelection). *For every $\epsilon > 0$ and $0 < \delta \leq 1$, there is a $\mathrm{poly}(n, \ell, \log |\mathcal{U}|)$-time $(\epsilon, \delta)$-DP algorithm that, with probability at least $1 - \beta$, outputs a universe element that solves the $\ell$-SparseSelection problem with additive error $O\left(\frac{1}{\epsilon} \log\left(\frac{n\ell}{\min\{\epsilon, 1\} \cdot \delta\beta}\right)\right)$, for every $\beta \in (0, 1)$.*

Next, we give a pure-DP algorithm for the problem. This algorithm is nearly identical to the original Exponential Mechanism of McSherry and Talwar [MT07] except that, instead of going over all elements of $\mathcal{U}$ in the algorithm itself, we assume that there is an oracle $\mathcal{O}$ that can sample an approximately uniformly random element from $\mathcal{U}$.

**Lemma 37** (Pure-DP Algorithm for SparseSelection). *Suppose there is an oracle $\mathcal{O}$ that runs in time $\mathrm{poly} \log |\mathcal{U}|$ and outputs a sample from $\mathcal{U}$ such that the probability of outputting each element $u \in \mathcal{U}$ is at least $p > 0$. Then, for every $\epsilon > 0$, there is a $\mathrm{poly}(n, \ell, \log |\mathcal{U}|)$-time $\epsilon$-DP algorithm that, with probability at least $1 - \beta$, outputs a universe element that solves the $\ell$-SparseSelection problem with additive error $O\left(\frac{1}{\epsilon} \ln\left(\frac{1}{\beta p}\right)\right)$, for every $\beta \in (0, 1)$.*

We remark that the approximate-DP algorithm in Lemma 36 has an additive error that does not grow with $|\mathcal{U}|$, whereas the pure-DP algorithm in Lemma 37 incurs an additive error that depends (at least) logarithmically on $|\mathcal{U}|$ because $p$ can be at most $\frac{1}{|\mathcal{U}|}$. It is simple to see that this $\log(|\mathcal{U}|)$ dependency of the pure-DP algorithm is necessary even when $\ell = 1$. Finally, note that Lemmas 37 and 36 imply Lemmas 11 and 12 in Section 3.1.2, respectively.

We next prove Lemma 36 in Section B.2.1 and Lemma 37 in Section B.2.2.

### B.2.1 Approximate-DP Algorithm

This section is devoted to the proof of Lemma 36. On a high level, the algorithm runs the Exponential Mechanism on the union $S_1 \cup \cdots \cup S_n$, with a small modification: we have an additional candidate $\perp$ whose score is fixed. We prove below that, when the score of $\perp$ is set to be sufficiently large (i.e., $O\left(\frac{1}{\epsilon} \log\left(\frac{\ell}{\epsilon\delta}\right)\right)$), the resulting algorithm is $(\epsilon, \delta)$-DP.

---

**Algorithm 2** Approximate-DP Algorithm for SparseSelection.

1: **procedure** APXSPARSESELECTION($\mathbf{S} = (S_1, \ldots, S_n)$)
2: $\quad \mathcal{U}(\mathbf{S}) \leftarrow S_1 \cup \cdots \cup S_n$.
3: $\quad$ **for** $u \in \mathcal{U}(\mathbf{S})$ **do**
4: $\quad\quad score_{\mathbf{S}}[u] \leftarrow |\{i \mid u \in S_i\}|$
5: $\quad score_{\mathbf{S}}[\perp] \leftarrow \frac{2}{\epsilon}\left(1 + \ln\left(\frac{\ell}{\delta(1 - e^{-\epsilon/2})}\right)\right)$
6: $\quad$ **return** a value drawn from $\mathcal{U}(\mathbf{S}) \cup \{\perp\}$ where $u$ has probability $\frac{e^{(\epsilon/2) \cdot score_{\mathbf{S}}[u]}}{\sum_{u \in \mathcal{U}(\mathbf{S}) \cup \{\perp\}} e^{(\epsilon/2) \cdot score_{\mathbf{S}}[u]}}$

---

*Proof of Lemma 36.* We now prove that Algorithm 2 satisfies the desired privacy and accuracy guarantees. For brevity, we use $\mathcal{M}$ as a shorthand for the mechanism APXSPARSESELECTION. It is immediate that the algorithm runs in time $\text{poly}(n, \ell, \log|\mathcal{U}|)$, as desired.

**Privacy.** Consider any pair of neighboring input datasets $\mathbf{S}$ and $\mathbf{S}'$. Recall that to show that the algorithm is $(\epsilon, \delta)$-DP, it suffices to show that

$$\Pr_{o \sim \mathcal{M}(\mathbf{S})}\left[\frac{\Pr[o = \mathcal{M}(\mathbf{S})]}{\Pr[o = \mathcal{M}(\mathbf{S}')]} > e^{\epsilon}\right] \leq \delta. \tag{1}$$

To prove the inequality in (1), let $score_{\perp}$ be the (fixed) score of $\perp$. Additionally, we denote

$$Z_{\mathbf{S}} := \sum_{u \in \mathcal{U}(\mathbf{S}) \cup \{\perp\}} e^{(\epsilon/2) \cdot score_{\mathbf{S}}[u]},$$

$$Z_{\mathbf{S}'} := \sum_{u \in \mathcal{U}(\mathbf{S}') \cup \{\perp\}} e^{(\epsilon/2) \cdot score_{\mathbf{S}'}[u]}.$$

First, we will argue that $Z_{\mathbf{S}} \geq e^{-\epsilon/2} \cdot Z_{\mathbf{S}'}$. This holds because

$$Z_{\mathbf{S}} = \sum_{u \in \mathcal{U}(\mathbf{S}) \cup \{\perp\}} e^{(\epsilon/2) \cdot score_{\mathbf{S}}[u]}$$

$$\geq \left(\sum_{u \in \mathcal{U}(\mathbf{S}) \cap \mathcal{U}(\mathbf{S}')} e^{(\epsilon/2) \cdot score_{\mathbf{S}}[u]}\right) + e^{(\epsilon/2) \cdot score_{\perp}}$$

$$\geq \left(\sum_{u \in \mathcal{U}(\mathbf{S}) \cap \mathcal{U}(\mathbf{S}')} e^{(\epsilon/2) \cdot (score_{\mathbf{S}'}[u] - 1)}\right) + e^{(\epsilon/2) \cdot score_{\perp}}$$

$$= e^{-\epsilon/2} \cdot Z_{\mathbf{S}'} - \left(\sum_{u \in \mathcal{U}(\mathbf{S}') \setminus \mathcal{U}(\mathbf{S})} e^{(\epsilon/2) \cdot (score_{\mathbf{S}'}[u] - 1)}\right) + e^{(\epsilon/2) \cdot score_{\perp}} \cdot \left(1 - e^{-\epsilon/2}\right). \tag{2}$$

Now observe that if $u$ belongs to $\mathcal{U}(\mathbf{S}') \setminus \mathcal{U}(\mathbf{S})$, it must belong to a single set in $\mathbf{S}'$ or equivalently $score_{\mathbf{S}'}[u] = 1$. Furthermore, since each set has size at most $\ell$, we have $|\mathcal{U}(\mathbf{S}') \setminus \mathcal{U}(\mathbf{S})| \leq \ell$. Plugging this into (2), we get

$$Z_{\mathbf{S}} \geq e^{-\epsilon/2} \cdot Z_{\mathbf{S}'} - \ell + e^{(\epsilon/2) \cdot score_\perp} \cdot \left(1 - e^{-\epsilon/2}\right)$$

$$\geq e^{-\epsilon/2} \cdot Z_{\mathbf{S}'}, \tag{3}$$

where the last inequality holds from our setting of $score_\perp$ in Algorithm 2.

For every $u \in (\mathcal{U}(S) \cap \mathcal{U}(S')) \cup \{\perp\}$, we thus get that

$$\frac{\Pr[u = \mathcal{M}(\mathbf{S})]}{\Pr[u = \mathcal{M}(\mathbf{S}')]} = \frac{e^{\epsilon/2 \cdot score_{\mathbf{S}}[u]}/Z_{\mathbf{S}}}{e^{\epsilon/2 \cdot score_{\mathbf{S}'}[u]}/Z_{\mathbf{S}'}}$$

$$\leq \frac{e^{\epsilon/2 \cdot (score_{\mathbf{S}'}[u]+1)}/Z_{\mathbf{S}}}{e^{\epsilon/2 \cdot score_{\mathbf{S}'}[u]}/Z_{\mathbf{S}'}}$$

$$\leq e^\epsilon,$$

where the last inequality follows from (3) above. As a result, we obtain

$$\Pr_{u \sim \mathcal{M}(\mathbf{S})}\left[\frac{\Pr[u = \mathcal{M}(\mathbf{S})]}{\Pr[u = \mathcal{M}(\mathbf{S}')]} > e^\epsilon\right] \leq \Pr_{u \sim \mathcal{M}(\mathbf{S})}[u \in \mathcal{U}(\mathbf{S}) \setminus \mathcal{U}(\mathbf{S}')]$$

$$= \sum_{u \in \mathcal{U}(\mathbf{S}) \setminus \mathcal{U}(\mathbf{S}')} \frac{e^{\epsilon/2 \cdot score_{\mathbf{S}}[u]}}{Z_{\mathbf{S}}}$$

$$= \sum_{u \in \mathcal{U}(\mathbf{S}) \setminus \mathcal{U}(\mathbf{S}')} \frac{e^{\epsilon/2}}{Z_{\mathbf{S}}}$$

$$\leq \frac{\ell \cdot e^{\epsilon/2}}{Z_{\mathbf{S}}}$$

$$\leq \frac{\ell \cdot e^{\epsilon/2}}{e^{\epsilon/2 \cdot score_\perp}}$$

$$\leq \delta,$$

where the second equality uses the fact that $score_{\mathbf{S}}[u] = 1$ whenever $u \in \mathcal{U}(\mathbf{S}) \setminus \mathcal{U}(\mathbf{S}')$, and the last inequality follows from our setting of $score_\perp$ in Algorithm 2. Thus, Algorithm 2 is $(\epsilon, \delta)$-DP as claimed.

**Accuracy.** We will now show that, with probability at least $1 - \beta$, Algorithm 2 outputs a universe element that solves the SparseSelection problem with additive error[13] $t = score_\perp + \frac{2}{\epsilon} \ln\left(\frac{2n\ell}{\beta}\right)$. To do so, we let OPT $:= \max_{u \in \mathcal{U}} |\{i \mid u \in S_i\}|$. If OPT $\leq t$, the statement trivially holds. If OPT $> t$, we let $\mathcal{U}_{good} := \{u \in \mathcal{U} \mid |\{i \mid u \in S_i\}| \geq \text{OPT} - t\}$. Let $Z_{good} := \sum_{u \in \mathcal{U}_{good}} e^{(\epsilon/2) \cdot score_{\mathbf{S}}[u]}$. Note that $Z_{good} \geq e^{(\epsilon/2) \cdot \text{OPT}}$. We therefore have that

$$\Pr_{u \sim \mathcal{M}(\mathbf{S})}[u \notin \mathcal{U}_{good}] = 1 - \frac{Z_{good}}{Z_{\mathbf{S}}}$$

$$= \frac{e^{(\epsilon/2) \cdot score_\perp} + \sum_{u \in \mathcal{U}(S) \setminus \mathcal{U}_{good}} e^{(\epsilon/2) \cdot score_{\mathbf{S}}[u]}}{Z_{good} + e^{(\epsilon/2) \cdot score_\perp} + \sum_{u \in \mathcal{U}(S) \setminus \mathcal{U}_{good}} e^{(\epsilon/2) \cdot score_{\mathbf{S}}[u]}}$$

$$\leq \frac{e^{(\epsilon/2) \cdot score_\perp} + n\ell \cdot e^{(\epsilon/2) \cdot (\text{OPT} - t)}}{Z_{good}}$$

$$\leq e^{(\epsilon/2) \cdot (score_\perp - \text{OPT})} + n\ell \cdot e^{(\epsilon/2) \cdot (-t)}$$

$$\leq \beta,$$

where the first inequality follows from the fact that $|\mathcal{U}(\mathbf{S})| \leq |S_1| + \cdots + |S_n| \leq n\ell$, and the last inequality follows from our setting of $t$ and from the assumption that OPT $> t$. We thus conclude that the output of Algorithm 2, with probability at least $1 - \beta$, solves SparseSelection with additive error $t$ as desired. $\square$

### B.2.2 Pure-DP Algorithm

We next prove Lemma 37. It relies on Algorithm 3, which is very similar to Algorithm 2 for approximate-DP, except that (i) instead of returning $\perp$, we draw from the oracle $\mathcal{O}$ and return its output, and (2) for each $u \in \mathcal{U}(S)$, we adjust the probability of sampling it directly to offset the probability that it is returned by $\mathcal{O}$. (Below the "adjusted multiplier" is $q_{\mathbf{S}}[u]$, which serves similar purpose to $e^{score_{\mathbf{S}}[u]}$ in the vanilla Exponential Mechanism.)

---

**Algorithm 3** Pure-DP Algorithm for SparseSelection.

1: **procedure** $\textsc{PureSparseSelection}_{\mathcal{O}}(\mathbf{S} = (S_1, \ldots, S_n))$
2:      $\mathcal{U}(\mathbf{S}) \leftarrow S_1 \cup \cdots \cup S_n$.
3:      **for** $u \in \mathcal{U}(\mathbf{S})$ **do**
4:          $score_{\mathbf{S}}[u] \leftarrow |\{i \mid u \in S_i\}|$
5:          $q_{\mathbf{S}}[u] \leftarrow e^{(\epsilon/2) \cdot score_{\mathbf{S}}[u]} - 1$
6:      $score_{\mathbf{S}}[\perp] \leftarrow \frac{2}{\epsilon} \ln\left(\frac{1}{p}\right)$
7:      $q_{\mathbf{S}}[\perp] \leftarrow e^{(\epsilon/2) \cdot score_{\mathbf{S}}[\perp]}$
8:      $\hat{u} \leftarrow$ a value drawn from $\mathcal{U}(\mathbf{S}) \cup \{\perp\}$ where $u$ has probability $\frac{q_{\mathbf{S}}[u]}{\sum_{u \in \mathcal{U}(\mathbf{S}) \cup \{\perp\}} q_{\mathbf{S}}[u]}$
9:      **if** $\hat{u} = \perp$ **then**
10:          **return** an output from a call to $\mathcal{O}$
11:      **else**
12:          **return** $\hat{u}$

---

*Proof of Lemma 37.* We now prove that Algorithm 3 yields the desired privacy and accuracy guarantees. For brevity, we use $\mathcal{M}$ as a shorthand for the mechanism $\textsc{PureSparseSelection}$. It is immediate that Algorithm 3 runs in time $\text{poly}(n, \ell, \log|\mathcal{U}|)$, as desired.

**Privacy.** For every $u \in \mathcal{U}$, we let $p_{\mathcal{O}}(u) \geq p$ denote the probability that the oracle $\mathcal{O}$ outputs $u$. For convenience, when $u \notin \mathcal{U}(\mathbf{S})$, we set $score_{\mathbf{S}}[u]$ to 0. We define

$$\widetilde{score}_{\mathbf{S}}[u] := \frac{2}{\epsilon} \cdot \ln\left(e^{(\epsilon/2) \cdot score_{\perp}} \cdot p_{\mathcal{O}}(u) + \mathbf{1}[u \in \mathcal{U}(\mathcal{S})] \cdot (e^{(\epsilon/2) \cdot score_{\mathbf{S}}[u]} - 1)\right)$$

$$= \frac{2}{\epsilon} \cdot \ln\left(e^{(\epsilon/2) \cdot score_{\perp}} \cdot p_{\mathcal{O}}(u) + (e^{(\epsilon/2) \cdot score_{\mathbf{S}}[u]} - 1)\right).$$

We observe that for an input $\mathbf{S} = (S_1, \cdots, S_n)$, the probability that each $u^* \in \mathcal{U}$ is selected is exactly $\frac{e^{(\epsilon/2) \cdot \widetilde{score}_{\mathbf{S}}(u^*)}}{\sum_{u \in \mathcal{U}} e^{(\epsilon/2) \cdot \widetilde{score}_{\mathbf{S}}(u)}}$. Thus, Algorithm 3 is equivalent to running the exponential mechanism of [MT07] with the scoring function $\widetilde{score}_{\mathbf{S}}$. Hence, to prove that Algorithm 3 is $\epsilon$-DP, it suffices to show that the sensitivity of $\widetilde{score}_{\mathbf{S}}[u]$ is at most 1. Consider any two neighboring datasets $\mathbf{S}$ and $\mathbf{S}'$. Due to symmetry, it suffices to show that

$$\widetilde{score}_{\mathbf{S}}[u] - \widetilde{score}_{\mathbf{S}'}[u] \leq 1,$$

which is equivalent to

$$\frac{e^{(\epsilon/2) \cdot score_{\perp}} \cdot p_{\mathcal{O}}(u) + (e^{(\epsilon/2) \cdot score_{\mathbf{S}}[u]} - 1)}{e^{(\epsilon/2) \cdot score_{\perp}} \cdot p_{\mathcal{O}}(u) + (e^{(\epsilon/2) \cdot score_{\mathbf{S}'}[u]} - 1)} \leq e^{\epsilon/2}. \tag{4}$$

To prove (4), notice that $e^{(\epsilon/2) \cdot score_{\perp}} = 1/p$. As a result, we have

$$\frac{e^{(\epsilon/2) \cdot score_{\perp}} \cdot p + (e^{(\epsilon/2) \cdot score_{\mathbf{S}}[u]} - 1)}{e^{(\epsilon/2) \cdot score_{\perp}} \cdot p + (e^{(\epsilon/2) \cdot score_{\mathbf{S}'}[u]} - 1)} = \frac{e^{(\epsilon/2) \cdot score_{\mathbf{S}}[u]}}{e^{(\epsilon/2) \cdot score_{\mathbf{S}'}[u]}} \leq e^{\epsilon/2}.$$

This, together with $p_{\mathcal{O}}(u) \geq p$, implies that (4) holds, and hence our algorithm is $\epsilon$-DP as desired.

**Accuracy.** The accuracy analysis is very similar to the proof of Lemma 36. Specifically, we will now show that, with probability at least $1 - \beta$, Algorithm 3 outputs a universe element that solves the SparseSelection problem with additive error[14] $t = score_\perp + \frac{2}{\epsilon} \ln \left( \frac{2|\mathcal{U}|}{\beta} \right)$. To do so, we let $\mathrm{OPT} := \max_{u \in \mathcal{U}} |\{i \mid u \in S_i\}|$. If $\mathrm{OPT} \leq t$, the statement trivially holds. If $\mathrm{OPT} > t$, we let $\mathcal{U}_{good} := \{u \in \mathcal{U} \mid |\{i \mid u \in S_i\}| \geq \mathrm{OPT} - t\}$. Let $Z_{good} := \sum_{u \in \mathcal{U}_{good}} e^{(\epsilon/2) \cdot \widetilde{score}_\mathbf{S}[u]}$. Note that $Z_{good} \geq e^{(\epsilon/2) \cdot \mathrm{OPT}}$. Also, let $Z_\mathbf{S} := \sum_{u \in \mathcal{U}} e^{(\epsilon/2) \cdot \widetilde{score}_\mathbf{S}[u]}$. We therefore have that

$$
\begin{aligned}
\Pr_{u \sim \mathcal{M}(\mathbf{S})}[u \notin \mathcal{U}_{good}] &= 1 - \frac{Z_{good}}{Z_\mathbf{S}} \\
&\leq \frac{e^{(\epsilon/2) \cdot score_\perp} + \sum_{u \in \mathcal{U} \setminus \mathcal{U}_{good}}(e^{(\epsilon/2) \cdot score_\mathbf{S}[u]} - 1)}{Z_\mathbf{S}} \\
&\leq \frac{e^{(\epsilon/2) \cdot score_\perp} + |\mathcal{U}| \cdot e^{(\epsilon/2) \cdot (\mathrm{OPT} - t)}}{Z_{good}} \\
&\leq e^{(\epsilon/2) \cdot (score_\perp - \mathrm{OPT})} + |\mathcal{U}| \cdot e^{(\epsilon/2) \cdot (-t)} \\
&\leq \beta,
\end{aligned}
$$

where the last inequality follows from our setting of $t$ and from the assumption that $\mathrm{OPT} > t$. We thus conclude that the output of Algorithm 3, with probability at least $1 - \beta$, solves SparseSelection with additive error $t$ as desired. $\qquad\square$

### B.3 Putting Things Together

Having set up all the ingredients in Sections B.2 and B.1, we now put them together to derive our DP algorithm for DensestBall in low dimensions. The idea is to run Algorithm 4, where the algorithm for SparseSelection is either from Lemma 36 or Lemma 37.

---

**Algorithm 4** DensestBall Algorithm.

---

1: **procedure** DENSESTBALLLOWDIMENSION$(x_1, \ldots, x_n; r, \alpha)$
2:     $C_{\alpha r} \leftarrow \alpha r$-cover from Lemma 29
3:     **for** $i \in \{1, \ldots, n\}$ **do**
4:         $S_i \leftarrow$ decoded list of $x$ at distance $(1 + \alpha)r$ with respect to $C_{\alpha r}$
        **return** SparseSelection $(S_1, \ldots, S_n)$

---

When we set SparseSelection on Line 4 to be the pure-DP algorithm for SparseSelection from Lemma 37, we obtain the pure-DP algorithm for DensestBall in low dimensions (Theorem 26).

*Proof of Theorem 26.* We run Algorithm 4 with SparseSelection being the $\epsilon$-DP algorithm from Lemma 37 using the oracle $\mathcal{O}$ from Lemma 30 for $C_{\alpha r}$. Recall that the list size $\ell$ guarantee from Lemma 29 is $((1 + \alpha)/\alpha)^{O(d)} = (1 + 1/\alpha)^{O(d)}$. Hence, the running time of the algorithm is $\mathrm{poly}(\ell, d, \log(1/r)) = (1 + 1/\alpha)^{O(d)} \mathrm{poly} \log(1/r)$ as desired.

The privacy of the algorithm follows immediately from the $\epsilon$-DP of the SparseSelection algorithm. Finally, to argue about its accuracy, assume that there exists a ball $\mathcal{B}(c^*, r)$ that contains at least $T$ of the input points. Since $C_{\alpha r}$ is an $\alpha r$-cover of the unit ball, there exists $c \in C_{\alpha r}$ such that $\|c - c^*\| \leq \alpha r$. As a result, $\mathcal{B}(c, (1 + \alpha)r)$ contains at least $T$ of the input points, which means that $c$ belongs to the decoded list $S_i$ of these points. By Lemma 37, the algorithm SparseSelection outputs, with probability at least $1 - \beta$, a center $c'$ that belongs to at least $T - O\left( \frac{1}{\epsilon} \log \left( \frac{1}{\beta p} \right) \right) = T - O\left( \frac{1}{\epsilon} \log \left( \frac{|C_{\alpha r}|}{\beta} \right) \right) = T - O_\alpha \left( \frac{d}{\epsilon} \log \left( \frac{1}{\beta \alpha r} \right) \right)$ decoded lists $S_i$'s. This indeed means that $c'$ is a $\left( 1 + \alpha, O_\alpha \left( \frac{d}{\epsilon} \log \left( \frac{1}{\beta r} \right) \right) \right)$-approximate solution, as desired. $\qquad\square$

We similarly obtain an approximate-DP algorithm for DensestBall with possibly smaller additive error than in Theorem 26 by setting SparseSelection to be the approximate-DP algorithm for SparseSelection from Lemma 36:

*Proof of Theorem 27.* The proof of this theorem is exactly the same as that of Theorem 26, except that SparseSelection is chosen as the $(\epsilon, \delta)$-DP algorithm from Lemma 36. □

## C $k$-means and $k$-median in Low Dimensions

In this section, we use our algorithm for DensestBall in low dimensions from Section B to obtain DP approximation algorithms for $k$-means and $k$-median, culminating in the proofs of the following theorems, which essentially matches the approximation ratios in the non-private case:

**Theorem 38.** *For any $p \geq 1$, suppose that there is a polynomial-time (not necessarily private) $w$-approximation algorithm for $(k, p)$-Clustering. Then, for every $\epsilon > 0$ and $0 < \alpha \leq 1$, there is an $\epsilon$-DP algorithm that runs in time $2^{O_{p,\alpha}(d)} \cdot \text{poly}(n)$ and, with probability $1 - \beta$, outputs a $\left(w(1 + \alpha), O_{p,\alpha,w}\left(\frac{k^2 \log^2 n \cdot 2^{O_{p,\alpha}(d)}}{\epsilon} \log\left(\frac{n}{\beta}\right) + 1\right)\right)$-approximation for $(k, p)$-Clustering, for every $\beta \in (0, 1)$.*

**Theorem 39.** *For every $\epsilon > 0$, $0 < \alpha \leq 1$ and $p \geq 1$, there is an $\epsilon$-DP algorithm that runs in time $2^{O_{\alpha,p}(dk + k \log k)} \cdot \text{poly}(n)$ and, with probability $1 - \beta$, outputs an $\left(1 + \alpha, O_{\alpha,p}\left(\frac{dk^2 \log n}{\epsilon} \log\left(\frac{n}{\beta}\right) + 1\right)\right)$-approximation for $(k, p)$-Clustering, for every $\beta \in (0, 1)$.*

Note here that Theorem 38 implies Theorem 14 in Section 4.

The structure of the proof of Theorem 38 closely follows the outline in Section 4. First, in Section C.1, we construct a centroid set with $w = O(1)$ by repeated applications of DensestBall. From that point on, we roughly follow the approach of [FFKN09, HM04]. Specifically, in Section C.2, we refine our centroid set to get $w = 1 + \alpha$ using exponential covers. Then, in Section C.3.1, we argue that the noisy snapped points form a private coreset with $\gamma$ arbitrarily close to zero. Finally, in Section C.3.2, we put things together and obtain a proof of Theorem 38.

While this approach also yields an FPT algorithm with approximation ratio $1 + \alpha$, the additive errors will depend exponentially on $d$ (as in Theorem 38). In this case, the error can be reduced to $\text{poly}(d, k, \log n, 1/\epsilon)$ as stated in Theorem 39. Roughly speaking, we can directly run the Exponential Mechanism on the refined coreset. This is formalized in Section C.4.

### C.1 Coarse Centroid Set via Repeated Invocations of DensestBall

The first step in our approximation algorithm is to construct a "coarse" centroid set (with $w = O(1)$) by repeatedly applying our DensestBall algorithm[15], while geometrically increasing the radius $r$ with each call. Each time a center is found, we also remove points that are close to it. The procedure is described more precisely below as Algorithm 5. (Here we use 0 to denote the origin in $\mathbb{R}^d$.)

---

**Algorithm 5** Finding Coarse Centroid Set.

1: **procedure** COARSECENTROIDSET$^\epsilon(x_1, \dots, x_n)$
2:      $\mathbf{X}_{uncovered} \leftarrow (x_1, \dots, x_n)$
3:      $\mathcal{C} \leftarrow \{0\}$
4:      **for** $i \in \{1, \dots, \lceil \log n \rceil\}$ **do**
5:          $r \leftarrow 2^i / n$
6:          **for** $j = 1, \dots, 2k$ **do**
7:              $c_{i,j} \leftarrow$ DENSESTBALLLOWDIMENSION$(\mathbf{X}_{uncovered}; r, 1)$
8:              $\mathcal{C} \leftarrow \mathcal{C} \cup \{c\}$
9:              $\mathbf{X}_{uncovered} \leftarrow \mathbf{X}_{uncovered} \setminus \mathcal{B}(c, 8r)$
     **return** $\mathcal{C}$

We can show that the produced set $\mathcal{C}$ is a centroid set with approximation ratio $w = O(1)$. In fact, below we state an even stronger property that for every $c$ and $r$ where the ball $\mathcal{B}(c, r)$ contains many points, at least one of the point in $\mathcal{C}$ is close to $c$. Throughout this section, we write OPT as a shorthand for $\mathrm{OPT}_{\mathbf{X}}^{p,k}$.

**Lemma 40.** *For any $d \in \mathbb{N}, \epsilon > 0$, and $0 < r, \alpha, \beta \leq 1$, let $T_{d,\epsilon,\beta,r,\alpha} = O_\alpha \left( \frac{d}{\epsilon} \log \left( \frac{1}{\beta r} \right) \right)$ be the additive error guarantee from Theorem 26. Furthermore, let $T^*$ be a shorthand for $T_{d, \frac{\epsilon}{2k\lceil \log n \rceil}, \frac{\beta}{2k\lceil \log n \rceil}, \frac{1}{n}, 1} = O \left( \frac{dk \log n}{\epsilon} \log \left( \frac{n}{\beta} \right) \right)$.*

*For every $\epsilon > 0$, there is a $2^{O(d)} \mathrm{poly}(n)$-time $\epsilon$-DP algorithm that outputs a set $\mathcal{C} \subseteq \mathbb{R}^d$ of size $O(k \log n)$ which, for every $\beta \in (0, 1)$, satisfies the following with probability at least $1 - \beta$: for all $c \in \mathbb{R}^d$ and $r \in \left[ \frac{1}{n}, 1 \right]$ such that $n_{c,r} := |\mathbf{X} \cap \mathcal{B}(c, r)|$ is at least $2T^*$, there exists $c' \in \mathcal{C}$ such that*
$$\|c - c'\| \leq 18 \cdot \max \left\{ r, \left( \frac{2\,\mathrm{OPT}}{n_{c,r}k} \right)^{1/p} \right\}.$$

Before we prove Lemma 40, let us note that it immediately implies that the output set is an $\left( O_p(1), O_p \left( \frac{dk^2 \log n}{\epsilon} \log \left( \frac{n}{\beta} \right) + 1 \right) \right)$-centroid set, as stated below. Nonetheless, we will not use this fact directly in subsequent steps since the properties in Lemma 40 are stronger and more convenient to use.

**Corollary 41.** *For every $\epsilon > 0$ and $p \geq 1$, there is an $2^{O(d)} \mathrm{poly}(n)$-time $\epsilon$-DP algorithm that, with probability $1 - \beta$, outputs an $\left( O_p(1), O_p \left( \frac{dk^2 \log n}{\epsilon} \log \left( \frac{n}{\beta} \right) + 1 \right) \right)$-centroid set for $(k, p)$-Clustering of size $O(k \log n)$, for every $\beta \in (0, 1)$.*

Note that Corollary 41 implies Lemma 15 in Section 4.

*Proof of Corollary 41.* We claim that the set of points $\mathcal{C}$ guaranteed by Lemma 40 forms the desired centroid set. To prove this, let us fix an optimal solution $c_1^*, \ldots, c_k^*$ of $(k, p)$-Clustering on the input $\mathbf{X}$. where ties are broken arbitrarily. For such a solution, let the map $\psi : [n] \to [k]$ be such that $c_{\psi(i)}^* \in \arg\min_{j \in [k]} \|x_i - c_j^*\|$ (with ties broken arbitrarily). For every $j \in [k]$, let[16] $n_j^* := |\psi^{-1}(j)|$ be the number of input points closest to center $c_j^*$ and let $r_j^* := \left( \frac{1}{n_j^*} \sum_{i \in \psi^{-1}(j)} \|x_i - c_j^*\|^p \right)^{1/p}$. Finally, we use $\tilde{r}_j$ to denote $\max \left\{ 2r_j^*, \frac{1}{n}, 2 \left( \frac{4\,\mathrm{OPT}}{n_j^* k} \right)^{1/p} \right\}$.

Let $T^*$ be as in Lemma 40. Let $J \subseteq [k]$ be the set $\{ j \in [k] \mid n_j^* \geq 4T^* \}$. Due to Markov's inequality and $p \geq 1$, we have that $|\mathbf{X} \cap \mathcal{B}(c_j, 2r_j^*)| \geq 0.5n_j^*$, which is at least $2T^*$ for all $j \in J$.

Thus, Lemma 40 ensures that, with probability $1 - \beta$, the following holds for all $j \in J$: there exists $c_j' \in \mathcal{C}$ such that $\|c_j' - c_j^*\| \leq 18\tilde{r}_j$. Henceforth, we will assume that this event holds and show that $\mathcal{C}$ must be an $\left( O_p(1), O_p \left( \frac{dk^2 \log n}{\epsilon} \log \left( \frac{n}{\beta} \right) + 1 \right) \right)$-centroid set of $\mathbf{X}$.

For convenience, we let $c_j' = 0$ for all $j \notin J$. From the discussion in the previous paragraph, we can derive

$$\mathrm{cost}_{\mathbf{X}}^p(c_1', \ldots, c_k') \leq \sum_{i \in [n]} \|x_i' - c_{\psi(i)}'\|^p$$
$$= \sum_{j \in [k]} \sum_{i \in \psi^{-1}(j)} \|x_i' - c_j'\|^p$$
$$= \sum_{j \in J} \sum_{i \in \psi^{-1}(j)} \|x_i' - c_j'\|^p + \sum_{j \in J \setminus [k]} \sum_{i \in \psi^{-1}(j)} \|x_i' - c_j'\|^p$$
$$\leq \sum_{j \in J} \sum_{i \in \psi^{-1}(j)} (\|x_i' - c_j^*\| + \|c_j^* - c_j'\|)^p + \sum_{j \in J \setminus [k]} \sum_{i \in \psi^{-1}(j)} 1$$

$$\leq \sum_{j \in J} \sum_{i \in \psi^{-1}(j)} \left(2^p \|x_i' - c_j^*\|^p + 2^p \|c_j^* - c_j'\|^p\right) + \sum_{j \in J \setminus [k]} 4T^*$$

$$\leq \sum_{j \in J} \sum_{i \in \psi^{-1}(j)} \left(2^p \|x_i' - c_j^*\|^p + 2^p \|c_j^* - c_j'\|^p\right) + 4kT^*$$

$$\leq 2^p \cdot \text{OPT} + 2^p \left(\sum_{j \in J} n_j^* \|c_j^* - c_j'\|^p\right) + O\left(\frac{dk^2 \log n}{\epsilon} \log\left(\frac{n}{\beta}\right)\right). \quad (5)$$

Now, since $\|c_j' - c_j^*\| \leq 18\tilde{r}_j$, we have

$$\sum_{j \in J} n_j^* \|c_j^* - c_j'\|^p \leq \sum_{j \in J} n_j^* (18\tilde{r}_j)^p$$

$$= 18^p \sum_{j \in J} n_j^* \left(\max\left\{2r_j^*, \frac{1}{n}, 2\left(\frac{4\,\text{OPT}}{n_j^* k}\right)^{1/p}\right\}\right)^p$$

$$\leq 18^p \sum_{j \in J} n_j^* \left((2r_j^*)^p + \left(\frac{1}{n}\right)^p + \frac{4\,\text{OPT}}{n_j^* k}\right)$$

$$\leq 36^p \,\text{OPT} + 18^p + 4 \cdot 18^p \cdot \text{OPT}.$$

Plugging this back into 5, we have

$$\text{cost}_{\mathbf{X}}^p(c_1', \ldots, c_k') \leq O_p(1) \cdot \text{OPT} + O_p\left(\frac{dk^2 \log n}{\epsilon} \log\left(\frac{n}{\beta}\right) + 1\right),$$

which concludes our proof. $\qquad\qquad\square$

We will now turn our attention back to the proof of Lemma 40.

*Proof of Lemma 40.* We claim that Algorithm 5, where DensestBall on Line 7 is the $\left(\frac{\epsilon}{2k\lceil \log n\rceil}\right)$-DP algorithm from Theorem 26 (with $\alpha = 1$), satisfies the properties. It is clear that the runtime of the algorithm is as claimed. We will next argue the privacy and security guarantees of our algorithm.

**Privacy.** We will now argue that the algorithm is $\epsilon$-DP. To do so, consider any pair of datasets $\mathbf{X}, \mathbf{X}'$ and any possible output $\tilde{\mathbf{c}} = (\tilde{c}_{i,j})_{i \in [\lceil \log n\rceil], j \in [2k]}$. Furthermore, let $\mathcal{M}$ be the shorthand for our algorithm COARSECANDIDATES, and for every $(i,j) \in [\lceil \log n\rceil] \times [2k]$, let $R_{<(i,j)} = \{(i', j') \in [\lceil \log n\rceil] \times [2k] \mid i' < i \text{ or } i' = i, j' < j\}$. We have

$$\frac{\Pr[\mathcal{M}(\mathbf{X}) = \mathbf{c}]}{\Pr[\mathcal{M}(\mathbf{X}') = \mathbf{c}]} \quad\quad\quad\quad\quad\quad\quad\quad\quad\quad\quad (6)$$

$$= \Pi_{(i,j) \in [\lceil \log n\rceil] \times [2k]} \frac{\Pr\left[\mathcal{M}(\mathbf{X})_{(i,j)} = \tilde{c}_{i,j} \mid \forall (i', j') \in R_{<(i,j)} \mathcal{M}(\mathbf{X})_{(i',j')} = \tilde{c}_{i',j'}\right]}{\Pr\left[\mathcal{M}(\mathbf{X}')_{(i,j)} = \tilde{c}_{i,j} \mid \forall (i', j') \in R_{<(i,j)} \mathcal{M}(\mathbf{X}')_{(i',j')} = \tilde{c}_{i',j'}\right]}. \quad (7)$$

Now note that when $\mathcal{M}(\mathbf{X})_{(i',j')} = \mathcal{M}(\mathbf{X}')_{(i',j')}$ for all $(i', j') < R_{<(i,j)}$, the sets $\mathbf{X}_{uncovered}$ at step $(i,j)$ of the two runs are neighboring datasets. Thus, the $\left(\frac{\epsilon}{2k\lceil \log n\rceil}\right)$-DP guarantee of the call to DensestBall on line 7 implies that

$$\frac{\Pr\left[\mathcal{M}(\mathbf{X})_{(i,j)} = \tilde{c}_{i,j} \mid \forall (i', j') \in R_{<(i,j)} \mathcal{M}(\mathbf{X})_{(i',j')} = \tilde{c}_{i',j'}\right]}{\Pr\left[\mathcal{M}(\mathbf{X}')_{(i,j)} = \tilde{c}_{i,j} \mid \forall (i', j') \in R_{<(i,j)} \mathcal{M}(\mathbf{X}')_{(i',j')} = \tilde{c}_{i',j'}\right]} \leq e^{\frac{\epsilon}{2k\lceil \log n\rceil}}.$$

Plugging this back into (6), we get

$$\frac{\Pr[\mathcal{M}(\mathbf{X}) = \mathbf{c}]}{\Pr[\mathcal{M}(\mathbf{X}') = \mathbf{c}]} \leq \left(e^{\frac{\epsilon}{2k\lceil \log n\rceil}}\right)^{2k\lceil \log n\rceil} = e^\epsilon,$$

which means that our algorithm is $\epsilon$-DP as desired.

**Accuracy.** The rest of this proof is devoted to proving the accuracy guarantee of Algorithm 5. To do so, we first note that the accuracy guarantee in Theorem 26 implies that each call to the DensestBall algorithm in line 7 solves the DensestBall problem with approximation ratio 2 and additive error $T^*$, with probability at least $1 - \frac{\beta}{2k\lceil \log n \rceil}$. By a union bound, this holds for *all* calls to DensestBall with probability at least $1 - \beta$. Henceforth, we assume that this event, which we denote by $E_{\mathsf{DensestBall}}$ for brevity, occurs.

Now, let us fix $c \in \mathbb{R}^d$ and $r \in [1/n, 1]$ such that $n_{c,r} := |\mathbf{X} \cap \mathcal{B}(c, r)|$ is at least $2T^*$. We will next argue that, with probability at least $1 - \beta$, there exists $c' \in \mathcal{C}$ such that $\|c - c'\| \leq 18 \cdot \max\left\{ r, \left( \frac{2\,\mathrm{OPT}}{n_{c,r}k} \right)^{1/p} \right\}$. We will prove this by contradiction.

Suppose for the sake of contradiction that for all $c' \in \mathcal{C}$, we have $\|c - c'\| > 18 \cdot \max\left\{ r, \left( \frac{2\,\mathrm{OPT}}{n_{c,r}k} \right)^{1/p} \right\}$. Let $\tilde{i} = \left\lceil \log \left( n \cdot \max\left\{ r, \left( \frac{2\,\mathrm{OPT}}{n_{c,r}k} \right)^{1/p} \right\} \right) \right\rceil$ and $\tilde{r} = 2^{\tilde{i}}/n$. Our assumption implies that

$$\|c - c'\| \geq 9\tilde{r} \tag{8}$$

for all $c' \in \mathcal{C}$.

Now, let us consider the centers selected on line 7 when $i = \tilde{i}$; let these centers be $c'_1, \ldots, c'_{2k}$. Using (8) and the fact that $\tilde{r} \geq r$, we get that all the $n_{c,r}$ points in $\mathbf{X} \cap \mathcal{B}(c, r)$ still remain in $\mathbf{X}_{uncovered}$. As a result, from our assumption that $E_{\mathsf{DensestBall}}$ occurs, when $c'_j$ is selected (in line 7) we must have that

$$|\mathcal{B}(c'_j, 2\tilde{r}) \cap \mathbf{X}_{uncovered}| \geq n_{c,r} - T^* \geq 0.5 n_{c,r}, \tag{9}$$

for all $j \in [2k]$. Note that this also implies that

$$\|c'_j - c'_{j'}\| > 6\tilde{r}, \tag{10}$$

for $j < j'$; otherwise, $\mathcal{B}(c'_{j'}, 2\tilde{r})$ would have been completely contained in $\mathcal{B}(c'_j, 8\tilde{r})$ and line 9 would have already removed all elements of $\mathcal{B}(c'_{j'}, 2\tilde{r})$ from $\mathbf{X}_{uncovered}$.

Now, consider any optimal solution $C^* = \{c^*_1, \ldots, c^*_k\}$ to the $(k, p)$-Clustering problem with cost OPT. Notice that (10) implies that the balls $\mathcal{B}(c'_1, 3\tilde{r}), \ldots, \mathcal{B}(c'_{2k}, 3\tilde{r})$ are disjoint. As a result, there must be (at least) $k$ selected centers $c'_{j_1}, \ldots, c'_{j_k}$ such that $\mathcal{B}(c'_{j_1}, 3\tilde{r}), \ldots, \mathcal{B}(c'_{j_k}, 3\tilde{r})$ do not contain any optimal centers from $C^*$. This implies that every point in $\mathcal{B}(c'_{j_1}, 2\tilde{r}), \ldots, \mathcal{B}(c'_{j_k}, 2\tilde{r})$ is at distance more than $\tilde{r}$ from any centers in $C^*$. Furthermore, from (10) and (9), the balls $\mathcal{B}(c'_{j_1}, 2\tilde{r}), \ldots, \mathcal{B}(c'_{j_k}, 2\tilde{r})$ are all pairwise disjoint and each contains at least $0.5 n_{c,r}$ points. This means that

$$
\begin{aligned}
\mathrm{cost}^p_{\mathbf{X}}(c^*_1, \ldots, c^*_k) &> k \cdot 0.5 \cdot n_{c,r} \cdot \tilde{r}^p \\
&\geq k \cdot 0.5 \cdot n_{c,r} \left( \left( \frac{2\,\mathrm{OPT}}{n_{c,r}k} \right)^{1/p} \right)^p \qquad \text{(from our choice of } \tilde{r}) \\
&= \mathrm{OPT}.
\end{aligned}
$$

This contradicts our assumption that $\mathrm{cost}^p_{\mathbf{X}}(c^*_1, \ldots, c^*_k) = \mathrm{OPT}$.

As a result, the accuracy guarantee holds conditioned on $E_{\mathsf{DensestBall}}$. Since we argued earlier that $\Pr[E_{\mathsf{DensestBall}}] \geq 1 - \beta$, we have completed our proof. □

## C.2 Centroid Set Refinement via Exponential Covers

As stated earlier, we will now follow the approach of [FFKN09], which is in turn based on a (non-private) coreset construction of [HM04]. Specifically, we refine our centroid set by placing exponential covers over each of the point in the coarse centroid set from Section C.1. This is described formally in Algorithm 6 below. We note that [HM04] orginally uses *exponential grids*, where covers are replaced by grids; this does not work for us because grids will lead to an additive error bound of $O(d)^d$ (instead of $O(1)^d$ for covers) which is super-polynomial for our regime of parameter $d = O(\log k)$. We also remark that exponential covers are implicitly taken in [FFKN09] where

---

**Algorithm 6** Centroid Set Refinement.

---

1: **procedure** REFINEDCENTROIDSET$^\epsilon(x_1, \ldots, x_n; \zeta)$
2:   $\mathcal{C} \leftarrow$ COARSECENTROIDSET$^\epsilon(x_1, \ldots, x_n)$
3:   $\mathcal{C}' \leftarrow \{0\}$
4:   **for** $c \in \mathcal{C}$ **do**
5:     **for** $i \in \{1, \ldots, \lceil \log n \rceil\}$ **do**
6:       $r \leftarrow 2^i/n$
7:       $C_{r,j} \leftarrow (\zeta r)$-cover of the ball $\mathcal{B}(c, 40r)$
8:       $\mathcal{C}' \leftarrow \mathcal{C}' \cup C_{r,j}$
      **return** $\mathcal{C}'$

---

the authors take equally space lines through each center and place points at exponentially increasing distance on each such line.

At this point, we take two separate paths. First, in Section C.3, we will continue following the approach of [FFKN09] and eventually prove Theorem 38. In the second path, we use a different approach to prove Theorem 39 in Section C.4.

While the REFINEDCANDIDATES algorithm will be used in both paths, the needed guarantees are different, and thus we will state them separately in each subsequent section.

### C.3 Approximation Algorithm I: Achieving Non-Private Approximation Ratio via Private Coresets

This section is devoted to the proof of Theorem 38. The bulk of the proof is in providing a good private coreset for the problem, which is done in Section C.3.1. As stated earlier, this part closely follows Feldman et al. [FFKN09], except that our proof is more general in that it works for every $p \geq 1$ and that we give a full analysis for all dimension $d$. Once the private coreset is constructed, we may simply run the non-private approximation algorithm on the coreset to get the desired result; this is formalized in Section C.3.2.

#### C.3.1 Private Coreset Construction

We first show that we can construct a private coreset efficiently when the dimension $d$ is small:

**Lemma 42.** *For every $\epsilon > 0$, $p \geq 1$ and $0 < \alpha < 1$, there is an $2^{O_{\alpha,p}(d)}\mathrm{poly}(n)$-time $\epsilon$-DP algorithm that, with probability $1 - \beta$, outputs an $\left(\alpha, O_{p,\alpha}\left(\frac{k^2 \log^2 n \cdot 2^{O_{p,\alpha}(d)}}{\epsilon} \log\left(\frac{n}{\beta}\right) + 1\right)\right)$-coreset for $(k, p)$-Clustering, for every $\beta \in (0, 1)$.*

Notice that Lemma 42 implies Lemma 16 in Section 4. The algorithm is presented below in Algorithm 7; here $\zeta$ is a parameter to be specified in the proof of Lemma 42.

---

**Algorithm 7** Private Coreset Construction.

---

1: **procedure** PRIVATECORESET$^\epsilon(x_1, \ldots, x_n; \zeta)$
2:   $\mathcal{C}' \leftarrow$ REFINEDCENTROIDSET$^{\epsilon/2}(x_1, \ldots, x_n; \zeta)$.
3:   **for** $c \in \mathcal{C}'$ **do**
4:     $count[c] = 0$
5:   **for** $i \in [n]$ **do**
6:     $x'_i \leftarrow$ closest point in $\mathcal{C}'$ to $x_i$
7:     $count[x'_i] \leftarrow count[x'_i] + 1$
8:   $\mathbf{X}' \leftarrow \emptyset$
9:   **for** $c \in \mathcal{C}'$ **do**
10:     $\widetilde{count}[c] \leftarrow count[c] + \mathrm{DLap}(2/\epsilon)$
11:     Add $\max\{\widetilde{count}[c], 0\}$ copies of $c$ to $\mathbf{X}'$
12:   **return** $\mathbf{X}'$

---

To prove Lemma 42, we will use the following simple fact:

**Fact 43.** *For any $p \geq 1$ and $\gamma > 0$, define $\lambda_{p,\gamma} := \left( \frac{1+\gamma}{((1+\gamma)^{1/p}-1)^p} \right)$. Then, for all $a, b \geq 0$, we have*

$$(a+b)^p \leq (1+\gamma)a^p + \lambda_{p,\gamma} \cdot b^p.$$

*Proof.* It is obvious to see that the inequality holds when $a = 0$ or $b = 0$. Hence, we may assume that $a, b > 0$. Now, consider two cases, based on whether $b \leq ((1+\gamma)^{1/p} - 1)\, a$.

If $b \leq ((1+\gamma)^{1/p} - 1)\, a$, we have $(a+b)^p \leq ((1+\gamma)^{1/p}a)^p = (1+\gamma)a^p$.

On the other hand, if $b > ((1+\gamma)^{1/p} - 1)\, a$, we have $a < \frac{b}{(1+\gamma)^{1/p}-1}$. This implies that

$$(a+b)^p \leq \left( \frac{(1+\gamma)^{1/p}}{(1+\gamma)^{1/p}-1} \cdot b \right)^p = \lambda_{p,\gamma} \cdot b^p. \qquad \square$$

We run Algorithm 6 with $\zeta = 0.01 \cdot \left( \frac{\alpha}{10\lambda_{p,\alpha/2}} \right)^{1/p}$. It is obvious that the algorithm is $\epsilon$-DP. Furthermore, the running time of the algorithm is polynomial in $n, k$ and the size of the cover used in Line 7 of Algorithm 6. We can pick such a cover so that the size[17] is $O(1/\zeta)^d = 2^{O_{\alpha,p}(d)}$ as desired. Thus, we are only left to prove that $\mathbf{X}'$ is (with high probability) a good coreset of $\mathbf{X}$.

To prove this, let $\mathbf{X}_{snapped}$ denote the multiset of points that contain $count[c]$ copies of every $c \in \mathcal{C}$. (In other words, for every input point $x_i \in \mathbf{X}$, we add its closest point $c_i$ from $\mathcal{C}$ to $\mathbf{X}_{snapped}$.) The correctness proof of Lemma 42 is then divided into two parts. First, we will show that $\mathbf{X}_{snapped}$ is a good coreset of $\mathbf{X}$:

**Lemma 44.** *For every $\beta > 0$, with probability $1 - \frac{\beta}{2}$, $\mathbf{X}_{snapped}$ is an $\left( \alpha, O_{p,\alpha}\left( \frac{dk^2 \log n}{\epsilon} \cdot \log\left( \frac{n}{\beta} \right) + 1 \right) \right)$-coreset of $\mathbf{X}$.*

Then, we show that the final set $\mathbf{X}'$ is a good coreset of $\mathbf{X}$.

**Lemma 45.** *For every $\beta > 0$, with probability $1 - \frac{\beta}{2}$, $\mathbf{X}'$ is a $\left( 0, O\left( \frac{(k \log^2 n) \cdot 2^{O_{p,\alpha}(d)}}{\epsilon} \cdot \log\left( \frac{n}{\beta} \right) \right) \right)$-coreset of $\mathbf{X}_{snapped}$.*

It is simple to see that Lemma 42 is an immediate consequence of Lemmas 44 and 45. Hence, we are left to prove these two lemmas.

**Snapped Points are a Coreset: Proof of Lemma 44.** The proof of Lemma 44 share some similar components as that in Corollary 41, but the $(\zeta r)$-covers employed in Algorithm 6 allow one to get a sharped bound, leading to the better ratio.

*Proof of Lemma 44.* Let us fix an optimal solution $c_1^*, \ldots, c_k^*$ of $(k, p)$-Clustering on the input $\mathbf{X}$. where ties are broken arbitrarily. For such a solution, let the map $\psi : [n] \to [k]$ be such that $c_{\psi(i)}^* \in \arg\min_{j \in [k]} \|x_i - c_j^*\|$ (with ties broken arbitrarily). For every $j \in [k]$, let $n_j^* := |\psi^{-1}(j)|$ be the number of input points closest to center $c_j^*$ and let $r_j^* := \left( \frac{1}{n_j^*} \sum_{i \in \psi^{-1}(j)} \|x_i - c_j^*\|^p \right)^{1/p}$. Finally, we let $\tilde{r}_j$ to denote $\max\left\{ 2r_j^*, \frac{1}{n}, 2\left( \frac{4\,\mathrm{OPT}}{n_j^* k} \right)^{1/p} \right\}$.

Let $T^*$ be as in Lemma 40, but with failure probability $\beta/2$ instead of $\beta$. Let $J \subseteq [k]$ be the set $\{ j \in [k] \mid n_j^* \geq 4T^* \}$. Due to Markov's inequality and $p \geq 1$, we have that $|\mathbf{X} \cap \mathcal{B}(c_j, 2r_j^*)| \geq 0.5 n_j^*$, which is at least $2T^*$ for all $j \in J$.

Thus, Lemma 40 ensures that, with probability $1 - \beta/2$, the following holds for all $j \in J$: there exists $c_j' \in \mathcal{C}$ such that $\|c_j' - c_j^*\| \leq 18\tilde{r}_j$. Henceforth, we will assume that this event holds and show that $\mathbf{X}_{snapped}$ must be an $\left( \alpha, O_{p,\alpha}\left( \frac{dk^2 \log n}{\epsilon} \log\left( \frac{n}{\beta} \right) \right) \right)$-coreset of $\mathbf{X}$.

Consider any input point $i \in \psi^{-1}(J)$. Let $\hat{r}_i = \|x_i - c^*_{\psi(i)}\| + 18\tilde{r}_{\psi(i)}$. From the previous paragraph, we have $\|x_i - c'_{\psi(i)}\| \leq \hat{r}_i$. Hence, from Line 7 of Algorithm 6,

$$\|x_i - x'_i\| \leq 2\zeta\hat{r}_i. \tag{11}$$

Now, consider any $c_1, \ldots, c_k \in \mathbb{R}^d$. We have

$$
\begin{aligned}
\mathrm{cost}^p_{\mathbf{X}_{snapped}}(c_1, \ldots, c_k) &= \sum_{i \in [n]} \left( \min_{j' \in [k]} \|x'_i - c_{j'}\| \right)^p \\
&\leq \sum_{i \in [n]} \left( \left( \min_{j' \in [k]} \|x_i - c_{j'}\| \right) + \|x_i - x'_i\| \right)^p \\
&\leq \sum_{i \in [n]} \left( (1 + \alpha/2) \cdot \left( \min_{j' \in [k]} \|x_i - c_{j'}\| \right)^p + \lambda_{p,\alpha/2} \cdot \|x_i - x'_i\|^p \right) \\
&\qquad\qquad\qquad\qquad\qquad\qquad\qquad\qquad\qquad\qquad \text{(by Fact 43)} \\
&= (1 + \alpha/2) \cdot \mathrm{cost}^p_{\mathbf{X}}(c_1, \ldots, c_k) + \lambda_{p,\alpha/2} \cdot \sum_{i \in [n]} \|x_i - x'_i\|^p. \tag{12}
\end{aligned}
$$

Now, we can separate the term $\sum_{i \in [n]} \|x_i - x'_i\|^p$ as follows.

$$
\begin{aligned}
\sum_{i \in [n]} \|x_i - x'_i\|^p &= \sum_{j \in k} \sum_{i \in \psi^{-1}(j)} \|x_i - x'_i\|^p \\
&= \sum_{j \in J} \sum_{i \in \psi^{-1}(j)} \|x_i - x'_i\|^p + \sum_{j \notin J} \sum_{i \in \psi^{-1}(j)} \|x_i - x'_i\|^p \\
&\overset{(11)}{\leq} \sum_{j \in J} \sum_{i \in \psi^{-1}(j)} (2\zeta\hat{r}_i)^p + \sum_{j \in [k] \setminus J} \sum_{i \in \psi^{-1}(j)} 1 \\
&\leq (2\zeta)^p \cdot \left( \sum_{j \in J} \sum_{i \in \psi^{-1}(j)} \hat{r}_i^p \right) + k \cdot 4T^* \\
&= (2\zeta)^p \cdot \left( \sum_{j \in J} \sum_{i \in \psi^{-1}(j)} \hat{r}_i^p \right) + O\left( \frac{dk^2 \log n}{\epsilon} \log \left( \frac{n}{\beta} \right) \right), \tag{13}
\end{aligned}
$$

where in the last inequality we recall from the definition that $|\psi^{-1}(j)| \leq 4T^*$ for all $j \notin J$.

From the definition of $\hat{r}_i$, we can now bound the term $\sum_{j \in J} \sum_{i \in \psi^{-1}(j)} \hat{r}_i^p$ by

$$
\begin{aligned}
\sum_{j \in J} \sum_{i \in \psi^{-1}(j)} \hat{r}_i^p &= \sum_{j \in J} \sum_{i \in \psi^{-1}(j)} \left( \|x_i - c^*_j\| + 18\tilde{r}_j \right)^p \\
&\leq 19^p \cdot \sum_{j \in J} \sum_{i \in \psi^{-1}(j)} \max\{\|x_i - c^*_{\psi(i)}\|, \tilde{r}_j\}^p \\
&= 19^p \cdot \sum_{j \in J} \sum_{i \in \psi^{-1}(j)} \left( \|x_i - c^*_{\psi(i)}\|^p + \tilde{r}_j^p \right) \\
&\leq 19^p \left( \mathrm{OPT} + \sum_{j \in J} n^*_j \tilde{r}_j^p \right). \tag{14}
\end{aligned}
$$

where the first inequality follows from the fact that $(a + b)^p \leq (2a)^p + (2b)^p$.

From the definition of $\tilde{r}_j$, we may now bound the term $\sum_{j \in J} n^*_j \tilde{r}_j^p$ by

$$
\sum_{j \in J} n^*_j \tilde{r}_j^p = \sum_{j \in J} n^*_j \cdot \max \left\{ 2r^*_j, \frac{1}{n}, 2 \left( \frac{4\,\mathrm{OPT}}{n^*_j k} \right)^{1/p} \right\}^p
$$

$$\begin{aligned}
&= 2^p \sum_{j \in J} n_j^* \cdot \left( (r_j^*)^p + \frac{1}{n} + \frac{4\,\mathrm{OPT}}{n_j^* k} \right) \\
&\le 2^p \left( \mathrm{OPT} + 1 + 4\,\mathrm{OPT} \right) \\
&= 5 \cdot 2^p \cdot \mathrm{OPT} + O_p(1). \qquad\qquad (15)
\end{aligned}$$

Plugging (13), (14), and (15) back into (12), we get

$$\mathrm{cost}^p_{\mathbf{X}_{snapped}}(c_1, \ldots, c_k)$$

$$\le (1 + \alpha/2) \cdot \mathrm{cost}^p_{\mathbf{X}}(c_1, \ldots, c_k) + \lambda_{p, \alpha/2} \cdot (100\zeta)^p \,\mathrm{OPT} + O_{p, \alpha}\left( \frac{dk^2 \log n}{\epsilon} \log\left( \frac{n}{\beta} \right) + 1 \right)$$

$$\le (1 + \alpha/2) \cdot \mathrm{cost}^p_{\mathbf{X}}(c_1, \ldots, c_k) + (\alpha/2) \cdot \mathrm{OPT} + O_{p, \alpha}\left( \frac{dk^2 \log n}{\epsilon} \log\left( \frac{n}{\beta} \right) + 1 \right)$$

$$\le (1 + \alpha) \cdot \mathrm{cost}^p_{\mathbf{X}}(c_1, \ldots, c_k) + O_{p, \alpha}\left( \frac{dk^2 \log n}{\epsilon} \log\left( \frac{n}{\beta} \right) + 1 \right),$$

where the second inequality follows from our choice of $\zeta$.

Using an analogous argument, we get that

$$\mathrm{cost}_{\mathbf{X}}(c_1, \ldots, c_k) \le (1 + \alpha) \cdot \mathrm{cost}^p_{\mathbf{X}_{snapped}}(c_1, \ldots, c_k) + O_{p, \alpha}\left( \frac{dk^2 \log n}{\epsilon} \log\left( \frac{n}{\beta} \right) + 1 \right).$$

Dividing both sides by $1 + \alpha$ yields

$$(1 - \alpha) \cdot \mathrm{cost}^p_{\mathbf{X}}(c_1, \ldots, c_k) \le \mathrm{cost}^p_{\mathbf{X}_{snapped}}(c_1, \ldots, c_k) + O_{p, \alpha}\left( \frac{dk^2 \log n}{\epsilon} \log\left( \frac{n}{\beta} \right) + 1 \right).$$

Thus, $\mathbf{X}_{snapped}$ is a $\left( 1 + \alpha, O_{p, \alpha}\left( \frac{dk^2 \log n}{\epsilon} \log\left( \frac{n}{\beta} \right) \right) + 1 \right)$-coreset of $\mathbf{X}$ as desired. $\qquad\square$

**Handling Noisy Counts: Proof of Lemma 45.** We next give a straightforward proof of Lemma 45. Similar statements were shown before in [FFKN09, Ste20]; we include the proof here for completeness.

*Proof of Lemma 45.* For each $c \in \mathcal{C}'$, recall that $|\widetilde{count}[c] - count[c]|$ is just distributed as the absolute value of the discrete Laplace distribution with parameter $2/\epsilon$. It is simple to see that, with probability $0.5\beta/|\mathcal{C}'|$, we have $|\widetilde{count}[c] - count[c]| \le \frac{\log(2|\mathcal{C}'|/\beta)}{\epsilon}$. As a result, by a union bound, we get that $\sum_{c \in \mathcal{C}'} |\widetilde{count}[c] - count[c]| \le |\mathcal{C}'| \cdot \frac{\log(|\mathcal{C}'|/\beta)}{\epsilon}$ with probability at least $1 - \beta/2$.

Finally, we observe that for any centers $c_1, \ldots, c_k \in \mathbb{R}^d$, it holds that

$$\begin{aligned}
&\left| \mathrm{cost}^p_{\mathbf{X}_{snapped}}(c_1, \ldots, c_k) - \mathrm{cost}^p_{\mathbf{X}'}(c_1, \ldots, c_k) \right| \\
&\le \sum_{c \in \mathcal{C}'} \left| \max\{\widetilde{count}[c], 0\} - count[c] \right| \cdot \left( \min_{i \in [k]} \| c_i - c \| \right). \\
&\le \sum_{c \in \mathcal{C}'} \left| \widetilde{count}[c] - count[c] \right| \\
&\le |\mathcal{C}'| \cdot \frac{\log(|\mathcal{C}'|/\beta)}{\epsilon}.
\end{aligned}$$

Finally, recall that $|\mathcal{C}'| \le |\mathcal{C}| \cdot \lceil \log n \rceil \cdot O(1/\zeta)^d = O\left( k \log^2 n \cdot 2^{O_{p, \alpha}(d)} \right)$. Plugging this to the above yields the desired bound. $\qquad\square$

### C.3.2 From Coreset to Approximation Algorithm

Finally, we give our DP approximation algorithm. This is extremely simple: first find a private coreset using Algorithm 7 and then run a (possibly non-private) approximation algorithm on this coreset.

---

**Algorithm 8** Algorithm for $(k, p)$-Clustering in Low Dimension.

---

1: **procedure** CLUSTERINGLOWDIMENSION$^\epsilon(x_1, \ldots, x_n, k; \zeta)$
2:      $\mathbf{X}' \leftarrow$ PRIVATECORESET$^\epsilon(x_1, \ldots, x_n; \zeta)$
3:      **return** NONPRIVATEAPPROXIMATION$(\mathbf{X}', k)$

---

As alluded to earlier, the above algorithm can give us an approximation ratio that is arbitrarily close to that of the non-private approximation algorithm, while the error remains small (when the dimension is small). This is formalized below.

*Proof of Theorem 38.* We run Algorithm 8 with $\zeta$ being the same as in the proof of Lemma 42, except that with approximation guarantee $0.1\alpha$ instead of $\alpha$, and NONPRIVATEAPPROXIMATION being the (not necessarily DP) $w$-approximation algorithm. The privacy and running time of the algorithm follow from Lemma 42. We will now argue its approximation guarantee.

By Lemma 42, with probability at least $1 - \beta$, $\mathbf{X}'$ is a $(0.1\alpha, t)$-coreset of $\mathbf{X}$, where $t = O_{p,\alpha}\left(\frac{k^2 \log^2 n \cdot 2^{O_{p,\alpha}(d)}}{\epsilon} \log\left(\frac{n}{\beta}\right) + 1\right)$. Let $c_1^*, \ldots, c_k^*$ be the optimal solution of $\mathbf{X}$. Since NONPRI-VATEAPPROXIMATION is a $w$-approximation algorithm, it must return a set $c_1, \ldots, c_k$ of centers such that

$$
\begin{aligned}
\text{cost}_{\mathbf{X}'}(c_1, \ldots, c_k) &\leq w \cdot \text{OPT}_{\mathbf{X}'}^{p,k} \\
&\leq w \cdot \text{cost}_{\mathbf{X}'}^p(c_1^*, \ldots, c_k^*) \\
&\leq w(1 + 0.1\alpha) \cdot \text{cost}_{\mathbf{X}}^p(c_1^*, \ldots, c_k^*) + wt \\
&\qquad\qquad\qquad \text{(since } \mathbf{X}' \text{ is a } (0.1\alpha, t)\text{-coreset of } \mathbf{X}) \\
&= w(1 + 0.1\alpha) \cdot \text{OPT}_{\mathbf{X}}^{p,k} + wt. \qquad\qquad (16)
\end{aligned}
$$

Using once again the fact that $\mathbf{X}'$ is a $(0.1\alpha, t)$-coreset of $\mathbf{X}$, we get

$$
\begin{aligned}
\text{cost}_{\mathbf{X}}^p(c_1, \ldots, c_k) &\leq \frac{1}{1 - 0.1\alpha} \cdot (\text{cost}_{\mathbf{X}'}^p(c_1, \ldots, c_k) + t) \\
&\overset{(16)}{\leq} \frac{1}{1 - 0.1\alpha} \cdot \left(w(1 + 0.1\alpha) \cdot \text{OPT}_{\mathbf{X}}^{p,k} + wt + t\right) \\
&\leq w(1 + \alpha) \text{OPT}_{\mathbf{X}}^{p,k} + O_w(t),
\end{aligned}
$$

which completes our proof. $\qquad\qquad\qquad\qquad\qquad\qquad\qquad\qquad\qquad\qquad\qquad$ $\square$

## C.4 Approximation Algorithms II: Private Discrete $(k, p)$-Clustering Algorithm

In this section, we show how to reduce the additive error in some cases, by using a DP algorithm for *Discrete* $(k, p)$-Clustering. Recall the definition of discrete $(k, p)$-Clustering from Section 2: in addition to $\mathbf{X} = (x_1, \ldots, x_n) \in (\mathbb{R}^d)^n$ and $k \in \mathbb{N}$, we are also given a set $\mathcal{C} \subseteq \mathbb{R}^d$ and the goal is to find $c_1, \ldots, c_k \in \mathcal{C}$ that minimizes $\text{cost}_{\mathbf{X}}^p(c_1, \ldots, c_k)$.

The overview is very simple: we will first show (in Section C.4.1) that REFINEDCENTROIDSET can produce a centroid set with an approximation ratio arbitrarily close to one. Then, we explain in Section C.4.2 that by running the natural Exponential Mechanism for Discrete $(k, p)$-Clustering with the candidate set being the output from REFINEDCENTROIDSET, we arrive at a solution for $(k, p)$-Clustering with an approximation ratio arbitrarily close to one, thereby proving Theorem 39.

We remark that previous works [BDL$^+$17, SK18, Ste20] also take the approach of producing a centroid set and then run DP approximation for Discrete $(k, p)$-Clustering from [GLM$^+$10]. However, the centroid sets produced in previous works do not achieve ratio arbitrarily close to one and thus cannot be used to derive such a result as our Theorem 39.

### C.4.1 Centroid Set Guarantee of REFINEDCENTROIDSET

The centroid set guarantee for the candidates output by REFINEDCENTROIDSET is stated below. The crucial point is that the approximation ratio can be $1 + \alpha$ for any $\alpha > 0$.

**Lemma 46.** *For every $\epsilon > 0, p \geq 1$ and $0 < \alpha \leq 1$, there is an $2^{O_{\alpha,p}(d)}\mathrm{poly}(n)$-time $\epsilon$-DP algorithm that, with probability $1 - \beta$, outputs an $\left(1 + \alpha, O_{\alpha,p}\left(\frac{dk^2 \log n}{\epsilon} \log\left(\frac{n}{\beta}\right) + 1\right)\right)$-centroid set for $(k, p)$-Clustering of size $O\left(k \log^2 n \cdot 2^{O_{\alpha,p}(d)}\right)$, for every $\beta \in (0, 1)$.*

The proof of Lemma 46 below follows similar blueprint as that of Lemma 44.

*Proof of Lemma 46.* We simply run Algorithm 6 with $\zeta = 0.01 \cdot \left(\frac{\alpha}{10\lambda_{p,\alpha/2}}\right)^{1/p}$ (where $\lambda_{\cdot,\cdot}$ is as defined in Fact 43). It follows immediately from Lemma 40 that the algorithm is $\epsilon$-DP. To bound the size of $\mathcal{C}$, note that we may pick the cover on Line 7 so that its size is $O(1/\zeta)^d = 2^{O_{\alpha,p}(d)}$. Hence, the size of the output set $\mathcal{C}'$ is at most $O\left(k \log^2 n \cdot 2^{O_{\alpha,p}(d)}\right)$ as desired.

We let $c_1^*, \ldots, c_k^*, \psi, n_1^*, \ldots, n_k^*, r_1^*, \ldots, r_k^*, \tilde{r}_1, \ldots, \tilde{r}_k, T^*, J$ be defined similarly as in the proof of Lemma 44.

Recall from the proof of Lemma 44 that, with probability at least $1 - \beta$, the following holds for all $j \in J$: there exists $c_j' \in \mathcal{C}$ such that $\|c_j' - c_j\| \leq 18\tilde{r}_j$. We henceforth assume that this event occurs. From line 7, this implies that for all $j \in J$ there exists $c_j \in \mathcal{C}'$ such that

$$\|c_j - c_j^*\| \leq 2\zeta\tilde{r}_j. \tag{17}$$

For all $j \notin J$, let $c_j = 0$ for notational convenience.

We will now bound $\mathrm{OPT}_{\mathbf{X}}^{p,k}(\mathcal{C}')$ as follows.

$$\begin{aligned}
\mathrm{OPT}_{\mathbf{X}}^{p,k}(\mathcal{C}') &\leq \mathrm{cost}_{\mathbf{X}}^p(c_1, \ldots, c_k) \\
&= \sum_{i \in [n]} \left(\min_{j' \in [k]} \|x_i - c_{j'}\|\right)^p \\
&= \sum_{j \in [k]} \sum_{i \in \psi^{-1}(j)} \left(\min_{j' \in [k]} \|x_i - c_{j'}\|\right)^p \\
&\leq \sum_{j \in [k]} \sum_{i \in \psi^{-1}(j)} \|x_i - c_j\|^p \\
&= \left(\sum_{j \in J} \sum_{i \in \psi^{-1}(j)} \|x_i - c_j\|^p\right) + \left(\sum_{j \in [k] \setminus J} \sum_{i \in \psi^{-1}(j)} \|x_i - c_j\|^p\right). \tag{18}
\end{aligned}$$

We will bound the two terms in (18) separately. First, we bound the second term. Recall that since $j \notin J$, we have that $|\psi^{-1}(j)| \leq n_j^* \leq 4T^* = O\left(\frac{dk \log n}{\epsilon} \log\left(\frac{n}{\epsilon\beta}\right)\right)$. Hence, we get

$$\begin{aligned}
\left(\sum_{j \in [k] \setminus J} \sum_{i \in \psi^{-1}(j)} \|x_i - c_j\|^p\right) &= \left(\sum_{j \in [k] \setminus J} \sum_{i \in \psi^{-1}(j)} \|x_i\|^p\right) \\
&\leq k \cdot 4T^* \\
&= O\left(\frac{dk^2 \log n}{\epsilon} \log\left(\frac{n}{\beta}\right)\right). \tag{19}
\end{aligned}$$

Next, we can bound the first term in (18) as follows.

$$\begin{aligned}
\left(\sum_{j \in J} \sum_{i \in \psi^{-1}(j)} \|x_i - c_j\|^p\right) &\leq \left(\sum_{j \in J} \sum_{i \in \psi^{-1}(j)} (\|x_i - c_j^*\| + \|c_j - c_j^*\|)^p\right) \\
&\leq \left(\sum_{j \in J} \sum_{i \in \psi^{-1}(j)} (1 + \alpha/2) \cdot \|x_i - c_j^*\|^p + \lambda_{p,\alpha/2} \cdot \|c_j - c_j^*\|^p\right) \\
&\qquad\qquad\qquad\qquad\qquad\qquad\qquad\qquad\qquad\qquad\qquad\text{(Fact 43)}
\end{aligned}$$

$$\leq (1 + \alpha/2) \cdot \text{OPT} + \left( \sum_{j \in J} n_j^* \cdot \lambda_{p,\alpha/2} \cdot \|c_j - c_j^*\|^p \right)$$

$$\overset{(17)}{\leq} (1 + \alpha/2) \cdot \text{OPT} + \left( \sum_{j \in J} n_j^* \cdot \lambda_{p,\alpha/2} \cdot (2\zeta\tilde{r}_j)^p \right)$$

$$= (1 + \alpha/2) \cdot \text{OPT} + \lambda_{p,\alpha/2} \cdot (2\zeta)^p \cdot \left( \sum_{j \in J} n_j^* \tilde{r}_j \right)$$

$$\leq (1 + \alpha/2) \cdot \text{OPT} + \lambda_{p,\alpha/2} \cdot (2\zeta)^p \cdot (5 \cdot 2^p \cdot \text{OPT} + O_p(1))$$

$$\leq (1 + \alpha) \cdot \text{OPT} + O_{\alpha,p}(1), \qquad \text{(from our choice of } \zeta)$$

where the second-to-last inequality holds via a similar argument to (15). Plugging (19) and (from our choice of $\zeta$) back into (18), we conclude that $\mathcal{C}'$ is a $\left(1 + \alpha, O_{\alpha,p}\left(\frac{dk^2 \log n}{\epsilon} \log\left(\frac{n}{\beta}\right) + 1\right)\right)$-centroid set of $\mathbf{X}$ as desired. $\qquad \square$

### C.4.2 Approximation Algorithm from Private Discrete $(k, p)$-Cluster

It was observed by Gupta et al. [GLM+10][18] that the straightforward application of the Exponential Mechanism [MT07] gives an algorithm with approximation ratio 1 and additive error $O\left(\frac{k \log |\mathcal{C}|}{\epsilon}\right)$, albeit with running time $|\mathcal{C}|^k \cdot \text{poly}(n)$:

**Theorem 47** ([GLM+10, Theorem 4.1]). *For any $\epsilon > 0$ and $p \geq 1$, there is an $|\mathcal{C}|^k \cdot \text{poly}(n)$-time $\epsilon$-DP algorithm that, with probability $1 - \beta$, outputs an $\left(1, O\left(\frac{k}{\epsilon} \log\left(\frac{|\mathcal{C}|}{\beta}\right)\right)\right)$-approximation for $(k, p)$-Clustering, for every $\beta \in (0, 1)$.*

Our algorithm is simply to run the above algorithm on $(\mathbf{X}, \text{REFINEDCENTROIDSET}(\mathbf{X}))$:

---
**Algorithm 9** Approximation Algorithm for $(k, p)$-Clustering.
---
1: **procedure** $\text{APXCLUSTERING}^\epsilon(x_1, \ldots, x_n; \zeta)$
2: $\quad \mathcal{C} \leftarrow \text{REFINEDCENTROIDSET}^{\epsilon/2}(x_1, \ldots, x_n; \zeta)$.
3: $\quad$ **return** $\text{DISCRETECLUSTERINGAPPROX}^{\epsilon/2}(x_1, \ldots, x_n, \mathcal{C}, k)$
---

*Proof of Theorem 39.* We run Algorithm 9, where $\zeta$ is as in the proof of Lemma 46 and the algorithm on Line 3 is an $(\epsilon/2)$-DP algorithm from Theorem 47. To see that the algorithm is $\epsilon$-DP, recall from Lemma 46 that the algorithm on Line 2 is $(\epsilon/2)$-DP. Since DISCRETECLUSTERINGAPPROX is $(\epsilon/2)$-DP, Basic Composition (Theorem 24) implies that the entire algorithm is $\epsilon$-DP as desired. The bottleneck in terms of running time comes from DISCRETECLUSTERINGAPPROX. From Theorem 47, the running time bound is

$$|\mathcal{C}|^k \cdot \text{poly}(n) \leq O(k \log^2 n \cdot 2^{O_{\alpha,p}(d)})^k \cdot \text{poly}(n) = 2^{O_{\alpha,p}(kd + k \log k)} \cdot \text{poly}(n)$$

where the bound on $|\mathcal{C}|$ comes from Lemma 46, and the second inequality comes from the fact that[19] $(k \log n)^k \leq 2^{O(k \log k)} \cdot \text{poly}(n)$.

Finally, we argue the approximation guarantee of the algorithm. Recall from Lemma 46 that, with probability $1 - \beta/2$, $\mathcal{C}$ is a $\left(1 + \alpha, O_{\alpha,p}\left(\frac{dk^2 \log n}{\epsilon} \log\left(\frac{n}{\beta}\right) + 1\right)\right)$-centroid set of $\mathbf{X}$. Furthermore, from the approximation guarantee of Theorem 47, DISCRETECLUSTERINGAPPROX outputs

$c_1, \ldots, c_k$ such that $\mathrm{cost}_{\mathbf{X}}^p(c_1, \ldots, c_k) \leq \mathrm{OPT}_{\mathbf{X}}^{p,k}(\mathcal{C}) + O\left(\frac{k}{\epsilon} \log\left(\frac{|\mathcal{C}|}{\beta}\right)\right)$. Combining these two, the following holds with probability $1 - \beta$:

$$\mathrm{cost}_{\mathbf{X}}^p(c_1, \ldots, c_k)$$
$$\leq \mathrm{OPT}_{\mathbf{X}}^{p,k}(\mathcal{C}) + O_p\left(\frac{k}{\epsilon} \log\left(\frac{|\mathcal{C}|}{\beta}\right)\right)$$
$$\leq \left((1 + \alpha) \cdot \mathrm{OPT} + O_{\alpha,p}\left(\frac{dk^2 \log n}{\epsilon} \log\left(\frac{n}{\beta}\right) + 1\right)\right) + O\left(\frac{k}{\epsilon} \log\left(\frac{k \log^2 n \cdot 2^{O_{\alpha,p}(d)}}{\beta}\right)\right)$$
$$\leq (1 + \alpha) \cdot \mathrm{OPT} + O_{\alpha,p}\left(\frac{dk^2 \log n}{\epsilon} \log\left(\frac{n}{\beta}\right) + 1\right),$$

which completes our proof. □

## D   Dimensional Reduction: There and Back Again

In this section, we will extend our algorithm to work in high dimension. The overall idea is quite simple: we will use well-known random dimensionality reduction techniques, and use our formerly described algorithms to solve the problem in this low-dimensional space. While the *centers* found in low-dimensional space may not immediately give us the information about the *centers* in the high-dimensional space, it does give us an important information: the *clusters*. For $(k, p)$-Clustering, these clusters mean the partition of the points into $k$ parts (each consisting of the points closest to each center). For DensestBall, the cluster is simply the set of points in the desired ball. As we will elaborate below, known techniques imply that it suffices to only consider these clusters in high dimension without too much additional error. Given these clusters, we only have to find the center in high-dimension. It turns out that this is an easier task, compared to determining the partitions themselves. In fact, without privacy constraints, finding the optimal center of a given cluster is a simple convex program. Indeed, for $(k, p)$-Clustering, finding a center privately can be done using known tools in private convex optimization [CMS11, KST12, JKT12, DJW13, BST14, WYX17]. On the other hand, the case of DensestBall is slightly more complicated, as applying these exisiting tools directly result in a large error; as we will see below, it turns out that we will apply another dimensional reduction one more time to overcome this issue.

We will now formalize the intuition outlined above. It will be convenient to use the following notation throughout this section: For any $\theta \geq 0$, we write $a \approx_{1+\theta} b$ to denote $\frac{1}{1+\theta} \leq \frac{a}{b} \leq 1 + \theta$.

### D.1   $(k, p)$-Clustering

We will start with $(k, p)$-Clustering. The formal statements of our results are stated below:

**Theorem 48.** *For any $p \geq 1$, suppose that there exists a polynomial time (not necessarily private) $w$-approximation algorithm for $(k, p)$-Clustering. Then, for every $0 < \epsilon \leq O(1)$ and $0 < \alpha, \beta \leq 1$, there exists an $\epsilon$-DP algorithm that runs in $(k/\beta)^{O_{p,\alpha}(1)}\mathrm{poly}(nd)$ time and, with probability $1 - \beta$, outputs an $\left(w(1 + \alpha), O_{p,\alpha,w}\left(\left(\frac{kd+(k/\beta)^{O_{p,\alpha}(1)}}{\epsilon}\right) \cdot \mathrm{poly}\log\left(\frac{n}{\beta}\right)\right)\right)$-approximation $(k, p)$-Clustering.*

**Theorem 49.** *For any $p \geq 1$, suppose that there exists a polynomial time (not necessarily private) $w$-approximation algorithm for $(k, p)$-Clustering. Then, for every $0 < \epsilon \leq O(1)$ and $0 < \delta, \alpha, \beta \leq 1$, there exists an $\epsilon$-DP algorithm that runs in $(k/\beta)^{O_{p,\alpha}(1)}\mathrm{poly}(nd)$ time and, with probability $1 - \beta$, outputs an $\left(w(1 + \alpha), O_{p,\alpha,w}\left(\left(\frac{k\sqrt{d}}{\epsilon} \cdot \mathrm{poly}\log\left(\frac{k}{\delta\beta}\right)\right) + \left(\frac{(k/\beta)^{O_{p,\alpha}(1)}}{\epsilon} \cdot \mathrm{poly}\log\left(\frac{n}{\beta}\right)\right)\right)\right)$-approximation for $(k, p)$-Clustering.*

We remark that, throughout this section, we will state our results under the assumption that $\epsilon \leq O(1)$. In all cases, our algorithms extend to the case $\epsilon = \omega(1)$, but with more complicated additve error expressions; thus, we choose not state them here.

To do so, we will need the following definition of the cost of a $k$-partition, as stated below. Roughly speaking, this means that we already fix the points assigned to each of the $k$ clusters, and we can only select the center of each cluster.

**Definition 50** (Partition Cost). *Given a partition $\mathcal{X} = (\mathbf{X}_1, \ldots, \mathbf{X}_k)$ of $\mathbf{X}$, its cost is defined as*

$$\text{cost}^p(\mathcal{X}) := \sum_{i=1}^{k} \min_{c_i \in \mathbb{R}^d} \|x_i - c_j\|^p.$$

For $(k, p)$-Clustering, we need the following recent breakthrough result due to Makarychev et al. [MMR19], which roughly stating that reducing to $O(\log k)$ dimension suffices to preserve the cost of $(k, p)$-Clustering for all paritions.

**Theorem 51** (Dimensionality Reduction for $(k, p)$-Cluster [MMR19]). *For every $0 < \beta, \tilde{\alpha} < 1, p \geq 1$ and $k \in \mathbb{N}$, there exists $d' = O_{\tilde{\alpha}}\left(p^4 \log(k/\beta)\right)$. Let $S$ be a random $d$-dimensional subspace of $\mathbb{R}^d$ and $\Pi_S$ denote the projection from $\mathbb{R}^d$ to $S$. Then, with probability $1 - \beta$, the following holds for every partition $\mathcal{X} = (\mathbf{X}_1, \ldots, \mathbf{X}_k)$ of $\mathbf{X}$:*

$$\text{cost}^p(\mathcal{X}) \approx_{1+\tilde{\alpha}} (d/d')^{p/2} \cdot \text{cost}^p(\Pi_S(\mathcal{X})),$$

*where $\Pi_S(\mathcal{X})$ denote the partition $(\Pi_S(\mathbf{X}_1), \ldots, \Pi_S(\mathbf{X}_k))$.*

Another ingredient we need is the algorithms for private empirical risk minimization (ERM). Recall that, in ERM, there is a convex loss function $\ell$ and we are given data points $x_1, \ldots, x_n$. The goal to find $\theta$ in the unit ball in $p$ dimension that minimizes $\sum_{i=1}^{n} \ell(\theta; x_i)$. When $\ell$ is $L$-Lipschitz, Bassily et al. [BST14] give an algorithm with small errors, both for pure- and approximate-DP. These are stated formally below.

**Theorem 52** ([BST14]). *Suppose that $\ell(\cdot; x)$ is convex and $L$-Lipschitz for some constant $L$. For every $\epsilon > 0$, there exists an $\epsilon$-DP polynomial time algorithm for ERM with loss function $\ell$ such that, with probability $1 - \beta$, the additive error is at most $O_L\left(\frac{d}{\epsilon} \cdot \text{poly} \log\left(\frac{1}{\beta}\right)\right)$, for every $\beta \in (0, 1)$.*

**Theorem 53** ([BST14]). *Suppose that $\ell(\cdot; x)$ is convex and $L$-Lipschitz for some constant $L$. For every $0 < \epsilon < O(1)$ and $0 < \delta < 1$, there exists an $\epsilon$-DP polynomial time algorithm for ERM with loss function $\ell$ such that, with probability $1 - \beta$, the additive error is at most $O_L\left(\frac{\sqrt{d}}{\epsilon} \cdot \text{poly} \log\left(\frac{n}{\delta\beta}\right)\right)$, for every $\beta \in (0, 1)$.*

We remark here that the "high probability" versions we use above are not described in the main body of [BST14], but they are included in Appendix D of the arXiv version of [BST14].

Notice that the $(1, p)$-Clustering is exactly the ERM problem, but with $\ell(\theta, x) = \|\theta - x\|^p$ where $\theta$ is the center. Note that since both $\theta, x \in \mathcal{B}(0, 1)$, $\ell(\cdot; x)$ is $O_p(1)$-Lipschitz for $p \geq 1$. It is also simple to see that $\ell(\cdot; x)$ is convex. Thus, results of [BST14] immediately yield the following corollaries.

**Corollary 54.** *For every $\epsilon > 0$ and $p \geq 1$, there exists an $\epsilon$-DP polynomial time algorithm for $(1, p)$-Clustering such that, with probability $1 - \beta$, the additive error is at most $O_p\left(\frac{d}{\epsilon} \cdot \text{poly} \log\left(\frac{1}{\beta}\right)\right)$, for every $\beta \in (0, 1)$.*

**Corollary 55.** *For every $0 < \epsilon < O(1), 0 < \delta < 1$ and $p \geq 1$, there exists an $(\epsilon, \delta)$-DP polynomial time algorithm for $(1, p)$-Clustering such that, with probability $1 - \beta$, the additive error is at most $O_p\left(\frac{\sqrt{d}}{\epsilon} \cdot \text{poly} \log\left(\frac{n}{\delta\beta}\right)\right)$, for every $\beta \in (0, 1)$.*

We are now ready to state the algorithm. As outlined before, we start by projecting to a random low-dimensional space and use our low-dimensional algorithm (Theorem 38) to determine the clusters (i.e., partition). Then, for each of the cluster, we use the algorithms above (Corollaries 54 and 55) to find the center. The full pseudo-code of the algorithm is given in Algorithm 10. There is actually one deviation from our rough outline here: we scale the points after projection by a factor of $\Lambda$ (and zero them out if the norm is larger than one). The reason is: if we do not implement this step, the additive error from our low dimensional algorithm will get multiplied by a factor of $(d/d')^{p/2} = \tilde{\Omega}(d^{p/2})$, which is too large for our purpose. By picking an appropriate scaling factor $\Lambda$, we only incur a polylogarithmic multiplicative factor in the additive error.

We will now prove the guarantee of the algorithm, starting with the pure-DP case:

*Proof of Theorem 48.* We simply run Algorithm 11 where $d'$ be as in Theorem 51 with failure probability $\beta/4$ and $\tilde{\alpha} = 0.1\alpha$, $\Lambda = \sqrt{\frac{0.01}{\log(n/\beta)} \cdot \frac{d'}{d}}$, CLUSTERINGLOWDIMENSION is the algorithm

---

**Algorithm 10** Algorithm for $(k, p)$-Clustering.

---

1: **procedure** CLUSTERINGHIGHDIMENSION$^\epsilon(x_1, \ldots, x_n; r, \alpha; d', \Lambda)$
2:     $S \leftarrow$ Random $d'$-dimension subspace of $\mathbb{R}^d$
3:     **for** $i \in \{1, \ldots, n\}$ **do**
4:         $\tilde{x}_i \leftarrow \Pi_S(x_i)$
5:         **if** $\|\tilde{x}_i\| \leq 1/\Lambda$ **then**
6:             $x_i' = \Lambda \tilde{x}_i$
7:         **else**
8:             $x_i' = 0$
9:     $(c_1', \ldots, c_k') \leftarrow$ CLUSTERINGLOWDIMENSION$^{\epsilon/2}(x_1', \ldots, x_n')$
10:    $(\mathbf{X}_1, \ldots, \mathbf{X}_k) \leftarrow$ the partition induced by $(c_1', \ldots, c_k')$ on $(x_1', \ldots, x_n')$
11:    **for** $j \in \{1, \ldots, k\}$ **do**
12:       $c_j \leftarrow$ FINDCENTER$^{\epsilon/2}(\mathbf{X}_j)$
13:    **return** $(c_1, \ldots, c_k)$

---

from Theorem 38 that is $(\epsilon/2)$-DP, has with $\alpha = 0.1\alpha$ and the failure probability $\frac{\beta}{4k}$, and FIND-CENTER is the algorithm from Corollary 54 that is $(\epsilon/2)$-DP and the failure probability $\beta/4$. Since algorithm CLUSTERINGLOWDIMENSION is $(\epsilon/2)$-DP and each parition $\mathbf{X}_j$ is applied FINDCENTER only once, the trivial composition implies that the entire algorithm is $\epsilon$-DP. Furthermore, it is obvious that every step except the application of CLUSTERINGLOWDIMENSION runs in polynomial time. From Theorem 38, the application of CLUSTERINGLOWDIMENSION takes

$$(1 + 10/\alpha)^{O_{p,\alpha}(d')}\text{poly}(n) = (1 + 10/\alpha)^{O_{p,\alpha}(\log(k/\beta))}\text{poly}(n) = (k/\beta)^{O_{p,\alpha}(1)}\text{poly}(n)$$

time. As a result, the entire algorithm runs in $(k/\beta)^{O_\alpha(1)}\text{poly}(nd)$ time as desired.

We will now prove the accuracy of the algorithm. Let $\tilde{\mathbf{X}} = (\tilde{x}_1, \ldots, \tilde{x}_n)$ and $\mathbf{X} = (x_1', \ldots, x_n')$. By applying Theorem 51, the following holds with probability $1 - \beta/4$:

$$\text{OPT}_{\tilde{\mathbf{X}}}^{p,k} \leq \left(\frac{d'}{d}\right)^{p/2} \cdot (1 + 0.1\alpha) \cdot \text{OPT}_{\mathbf{X}}^{p,k}. \tag{20}$$

Furthermore, standard concentration implies that $\|\tilde{x}_i\| \leq 1/\Lambda$ with probability $0.1\beta/n$. By union bound, this means that the following simultaneously holds for all $i \in \{1, \ldots, n\}$ with probability $1 - 0.1\beta$:

$$x_i' = \Lambda \tilde{x}_i. \tag{21}$$

When (20) and (21) both hold, we may apply Theorem 38, which implies that, with probability $1 - \beta/2$, we have

$$
\begin{aligned}
&\text{cost}_{\mathbf{X}'}^p(c_1, \ldots, c_k) \\
&\leq w(1 + 0.1\alpha)\text{OPT}_{\mathbf{X}'}^{p,k} + O_{p,\alpha,w}\left(\frac{k^2 \log^2 n \cdot 2^{O_{p,\alpha}(d)}}{\epsilon}\log\left(\frac{n}{\beta}\right)\right) \\
&= w(1 + 0.1\alpha)\text{OPT}_{\mathbf{X}'}^{p,k} + O_{p,\alpha,w}\left(\frac{(k/\beta)^{O_{p,\alpha}(1)}}{\epsilon} \cdot \text{poly}\log\left(\frac{n}{\beta}\right)\right) \\
&\overset{(21)}{=} \Lambda^p \cdot w(1 + 0.1\alpha)\text{OPT}_{\tilde{\mathbf{X}}}^{p,k} + O_{p,\alpha,w}\left(\frac{(k/\beta)^{O_{p,\alpha}(1)}}{\epsilon} \cdot \text{poly}\log\left(\frac{n}{\beta}\right)\right) \\
&\overset{(20)}{\leq} \Lambda^p \cdot w(1 + 0.3\alpha)\text{OPT}_{\tilde{\mathbf{X}}}^{p,k} + O_{p,\alpha,w}\left(\frac{(k/\beta)^{O_{p,\alpha}(1)}}{\epsilon} \cdot \text{poly}\log\left(\frac{n}{\beta}\right)\right),
\end{aligned} \tag{22}
$$

where the first equality follows from $d' = O_{p,\alpha}\left(\log\left(\frac{k}{\beta}\right)\right)$.

Let $\mathbf{X}_1', \ldots, \mathbf{X}_k'$ partition of $\mathbf{X}'$ induced by $c_1, \ldots, c_k$, and let $\tilde{\mathbf{X}}_1, \ldots, \tilde{\mathbf{X}}_k$ denote the corresponding partition of $\tilde{\mathbf{X}}$. From Theorem 51, the following holds with probability $1 - \beta/4$:

$$\text{cost}_{(\mathbf{X}_1, \ldots, \mathbf{X}_k)}^p \leq \left(\frac{d}{d'}\right)^{p/2} \cdot (1 + 0.1\alpha) \cdot \text{cost}_{(\tilde{\mathbf{X}}_1, \ldots, \tilde{\mathbf{X}}_k)}^p. \tag{23}$$

By union bound (20), (21), (22) and (23) together occur with probability $1 - 3\beta/4$. When this is the case, we have

$$\text{cost}^p_{(\mathbf{X}_1,\ldots,\mathbf{X}_k)}$$

$$\overset{(23)}{\leq} \left(\frac{d}{d'}\right)^{p/2} \cdot (1 + 0.1\alpha) \cdot \text{cost}^p_{(\tilde{\mathbf{X}}_1,\ldots,\tilde{\mathbf{X}}_k)}$$

$$\overset{(21)}{=} \frac{1}{\Lambda^p} \cdot \left(\frac{d}{d'}\right)^{p/2} \cdot (1 + 0.1\alpha) \cdot \text{cost}^p_{(\mathbf{X}'_1,\ldots,\mathbf{X}'_k)}$$

$$= \frac{1}{\Lambda^p} \cdot \left(\frac{d}{d'}\right)^{p/2} \cdot (1 + 0.1\alpha) \cdot \text{cost}^p_{\mathbf{X}'}(c_1,\ldots,c_k)$$

$$\overset{(22)}{\leq} \left(\frac{d}{d'}\right)^{p/2} \cdot w(1 + 0.5\alpha) \cdot \text{OPT}^{p,k}_{\tilde{\mathbf{X}}} + O_{p,\alpha,w}\left(\frac{1}{\Lambda^p} \cdot \left(\frac{d}{d'}\right)^{p/2} \cdot \frac{(k/\beta)^{O_{p,\alpha}(1)}}{\epsilon} \cdot \text{poly}\log\left(\frac{n}{\beta}\right)\right)$$

$$\overset{(20)}{\leq} w(1 + \alpha) \cdot \text{OPT}^{p,k}_{\mathbf{X}} + O_{p,\alpha,w}\left(\frac{1}{\Lambda^p} \cdot \left(\frac{d}{d'}\right)^{p/2} \cdot \left(\frac{(k/\beta)^{O_{p,\alpha}(1)}}{\epsilon} \cdot \text{poly}\log\left(\frac{n}{\beta}\right)\right)\right)$$

$$= w(1 + \alpha) \cdot \text{OPT}^{p,k}_{\mathbf{X}} + O_{p,\alpha,w}\left(\frac{(k/\beta)^{O_{p,\alpha}(1)}}{\epsilon} \cdot \text{poly}\log\left(\frac{n}{\beta}\right)\right), \tag{24}$$

where in the last inequality we use the fact that, by our choice of parameters, $\frac{1}{\Lambda^2} \cdot \frac{d}{d'} = O(\log(1/\beta))$.

Now, using the guarantee from Corollary (54) and the union bound over all $j = 1,\ldots,k$, the following holds simultaneously for all $j = 1,\ldots,k$ with probability $1 - \beta/4$:

$$\text{cost}^p_{\mathbf{X}_j}(c_j) \leq \text{OPT}^{p,1}_{\mathbf{X}_j} + O_p\left(\frac{d}{\epsilon} \cdot \log\left(\frac{k}{\beta}\right)\right). \tag{25}$$

When (24) and (25) both occur (with probability at least $1 - \beta$), we have

$$\text{cost}^p_{\mathbf{X}}(c_1,\ldots,c_k) \leq \sum_{j=1}^{k} \text{cost}^p_{\mathbf{X}_j}(c_j)$$

$$\overset{(25)}{\leq} \sum_{j=1}^{k} \left(\text{OPT}^{p,1}_{\mathbf{X}_j} + O_p\left(\frac{d}{\epsilon} \cdot \log\left(\frac{k}{\beta}\right)\right)\right)$$

$$= \text{cost}^p_{(\mathbf{X}_1,\ldots,\mathbf{X}_k)} + O_p\left(\frac{kd}{\epsilon} \cdot \log\left(\frac{k}{\beta}\right)\right)$$

$$\overset{(24)}{\leq} w(1 + \alpha) \cdot \text{OPT}^{p,k}_{\mathbf{X}} + O_{p,\alpha,w}\left(\left(\frac{kd + (k/\beta)^{O_{p,\alpha}(1)}}{\epsilon}\right) \cdot \text{poly}\log\left(\frac{n}{\beta}\right)\right),$$

which concludes our proof. $\qquad\square$

We will next state the proof for approximate-DP case, which is almost the same as that of the pure-DP case.

*Proof of Theorem 49.* This proof is exactly the same as that of Theorem 48, except that we use the $(1, p)$-Clustering algorithm from Corollary 55 instead of Corollary 54. Everything in the proof remains the same except that the additive error on the right handside of (25) becomes $O_p\left(\frac{\sqrt{d}}{\epsilon} \cdot \log\left(\frac{k}{\delta\beta}\right)\right)$ (instead of $O_p\left(\frac{d}{\epsilon} \cdot \log\left(\frac{k}{\beta}\right)\right)$ as in Theorem 48), resulting in the new additive error bound. $\qquad\square$

We remark that Theorems 48 and 49 imply Theorem 13 in Section 4.

**FPT Approximation Schemes.** Finally, we state the results for FPT algorithms below. These are almost exactly the same as above, except that we use the FPT algorithm from Theorem 39 to solve the low-dimensional $(k, p)$-Clustering, leading to approximation ratio arbitrarily close to one.

**Theorem 56.** *For every* $0 < \epsilon \leq O(1)$, $0 < \alpha, \beta \leq 1$ *and* $p \geq 1$, *there exists an* $\epsilon$-*DP algorithm that runs in* $(1/\beta)^{O_{p,\alpha}(k \log k)} \text{poly}(nd)$ *time and, w.p.* $1 - \beta$, *outputs an* $\left(1 + \alpha, O_{p,\alpha} \left(\left(\frac{kd + k^2}{\epsilon}\right) \cdot \text{poly} \log \left(\frac{n}{\beta}\right)\right)\right)$-*approximation for* $(k, p)$-*Clustering.*

*Proof.* This proof is the same as the proof of Theorem 48, except that we use the algorithm from Theorem 39 instead of that from Theorem 38. Note here that the bottleneck in the running time is from the application of Theorem 39, which takes $2^{O_{p,\alpha}(d'k + k \log k)} \cdot \text{poly}(n) = (1/\beta)^{O_{p,\alpha}(k \log k)} \cdot \text{poly}(n)$ time because $d = O_{p,\alpha}(\log(k/\beta))$. $\square$

**Theorem 57.** *For every* $0 < \epsilon \leq O(1)$, $0 < \delta, \alpha, \beta \leq 1$ *and* $p \geq 1$, *there exists an* $(\epsilon, \delta)$-*DP algorithm that runs in* $(1/\beta)^{O_{p,\alpha}(k \log k)} \text{poly}(nd)$ *time and, with probability* $1 - \beta$, *outputs an* $\left(1 + \alpha, O_{p,\alpha} \left(\left(\frac{k\sqrt{d}}{\epsilon} \cdot \text{poly} \log \left(\frac{k}{\delta \beta}\right)\right) + \left(\frac{k^2}{\epsilon} \cdot \text{poly} \log \left(\frac{n}{\beta}\right)\right)\right)\right)$-*approximation for* $(k, p)$-*Clustering.*

*Proof.* This is exactly the same as the proof of Theorem 49, except that we use the algorithm from Theorem 39 instead of that from Theorem 38. $\square$

## D.2 DensestBall

We refer to the variant of the DensestBall problem where we are promised that *all* points are within a certain radius as the 1-Center problem:

**Definition 58** (1-Center). *The input of* 1-*Center consists of* $n$ *points in the* $d$-*dimensional unit ball and a positive real number* $r$. *It is also promised that* all *input points lie in some ball of radius* $r$. *A* $(w, t)$-*approximation for* 1-*Center is a ball* $B$ *of radius* $w \cdot r$ *that contains at least* $n - t$ *input points.*

### D.2.1 1-Center Algorithm in High Dimension

Once again, we will first show how to solve the 1-Center problem in high dimensions:

**Lemma 59.** *For every* $\epsilon > 0$ *and* $0 < \alpha, \beta \leq 1$, *there exists an* $\epsilon$-*DP algorithm that runs in time* $(nd)^{O_\alpha(1)} \text{poly} \log(1/r)$ *and, with probability* $1 - \beta$, *outputs an* $\left(1 + \alpha, O_\alpha \left(\frac{d}{\epsilon} \cdot \log \left(\frac{d}{\beta r}\right)\right)\right)$-*approximation for* 1-*Center.*

**Lemma 60.** *For every* $0 < \epsilon \leq O(1)$ *and* $0 < \alpha, \beta, \delta \leq 1$, *there exists an* $(\epsilon, \delta)$-*DP algorithm that runs in time* $(nd)^{O_\alpha(1)} \text{poly} \log(1/r)$ *and, w.p.* $1 - \beta$, *outputs an* $\left(1 + \alpha, O_\alpha \left(\frac{\sqrt{d}}{\epsilon} \cdot \text{poly} \log \left(\frac{nd}{\epsilon \delta \beta}\right)\right)\right)$-*approximation for* 1-*Center.*

A natural way to solve the 1-Center problem in high dimensions is to use differentially private ERM similarly to the case of $(k, p)$-Clustering, but with a *hinge loss* such as $\ell(c, x) = \frac{1}{r} \max\{0, r - \|c - x\|\}$. In other words, the loss is zero if $c$ is within the ball of radius $r$ aroun the center $c$, whereas the loss is at least one when it is say at a distance $2r$ from $c$. The main issue with this approach is that the Lipchitz constant of this function is as large as $1/r$. However, since the expected error in the loss has to grow linearly with the Lipchitz constant [BST14], this will give us an additive error that is linear in $1/r$, which is undesirable.

Due to this obstacle, we will instead take a different path: use a dimensionality reduction argument again! More specifically, we randomly rotate each vector and think of blocks each of roughly $O(\log(nd))$ coordinates as a single vector. We then run our low-dimensional DensestBall algorithm from Section B on each block. Combining these solutions together immediately gives us the desired solution in the high-dimensional space. The full pseudo-code of the procedure is given below; here $b$ is the parameter of the algorithm, DENSESTBALLLOWDIMENSION is the algorithm for solving DensestBall in low dimensions, and we use the notation $y|_{i,\dots,j}$ to denote a vector resulting from the restriction of $y$ to the coordinates $i, \dots, j$.

---

**Algorithm 11** 1-Center Algorithm.

---

1: **procedure** 1-CENTER$_b(x_1, \ldots, x_n; r, \alpha)$
2:     $R \leftarrow$ Random $(d \times d)$ rotation matrix
3:     **for** $i \in \{1, \ldots, n\}$ **do**
4:         **for** $j \in \{1, \ldots, b\}$ **do**
5:             $x_i^j \leftarrow (Rx_i)|_{1 + \lfloor \frac{(j-1)d}{b} \rfloor, \ldots, \lfloor \frac{jd}{b} \rfloor}$
6:     **for** $j \in \{1, \ldots, b\}$ **do**
7:         $d^j \leftarrow \lfloor \frac{jd}{b} \rfloor - \lfloor \frac{(j-1)d}{b} \rfloor$
8:         $r^j \leftarrow (1 + 0.1\alpha) \cdot \sqrt{d^j/d} \cdot r$
9:         $c^j \leftarrow$ DENSESTBALLLOWDIMENSION$(x_1^j, \ldots, x_n^j; r^j, 0.1\alpha)$.
10:     $\tilde{c} \leftarrow$ concatenation of $c^1, \ldots, c^t$
11:     **return** $R^{-1}(\tilde{c})$

---

To prove the correctness of our algorithm, we will need the Johnson–Lindenstrauss (JL) lemma [JL84]. The version we use below follows from the proof in [DG03].

**Theorem 61** ([DG03]). *Let $v$ be any $d$-dimensional vector. Let $S$ denote a random $d$-dimensional subspace of $\mathbb{R}^d$ and let $\Pi_S$ denote the projection from $\mathbb{R}^d$ onto $S$. Then, for any $\zeta \in (0, 1)$ we have*

$$\Pr\left[ \|v\|_2 \approx_{1+\zeta} \sqrt{d/d'} \cdot \|\Pi v\|_2 \right] \geq 1 - 2\exp\left( -\frac{d'\zeta^2}{100} \right).$$

We are now ready to prove our results for 1-Center, starting with the pure-DP algorithm (Lemma 59).

*Proof of Lemma 59.* We simply run Algorithm 11 with $b = \max\left\{ 1, \lfloor \frac{d}{10^8 \log(nd/\beta)/\alpha^2} \rfloor \right\}$ and with DENSESTBALLLOWDIMENSION on Line 9 being the algorithm $\mathcal{A}$ from Theorem 26 that is $(\epsilon/b)$-DP, has approximation ratio $w = 1 + 0.1\alpha$ and failure probability $\frac{\beta}{2d}$. Since algorithm $\mathcal{A}$ is $(\epsilon/b)$-DP and we apply the algorithm $b$ times, the trivial composition implies that the entire algorithm is $\epsilon$-DP. Furthermore, it is obvious that every step except the application of $\mathcal{A}$ runs in polynomial time. From Theorem 26, the $j$th application of $\mathcal{A}$ takes time

$$(1 + 1/\alpha)^{O_\alpha(d/b)} \operatorname{poly}\log(1/r') = (1 + 1/\alpha)^{O_\alpha(\log(nd\beta))} \operatorname{poly}\log(\sqrt{d/d^j} \cdot r)$$
$$= (nd)^{O_\alpha(1)} \operatorname{poly}\log(1/r).$$

As a result, the entire algorithm runs in time $(nd)^{O_\alpha(1)} \operatorname{poly}\log(1/r)$ as desired.

The remainder of this proof is dedicated to proving the accuracy of the algorithm. To do this, let $c_{\text{OPT}}$ denote the solution, i.e., the center such that $x_1, \ldots, x_n \in \mathcal{B}(c_{\text{OPT}}, r)$. Moreover, for every $j \in \{1, \ldots, b\}$, let $c_{\text{OPT}}^j$ be $R(c_{\text{OPT}})$ restricted to the coordinates $1 + \lfloor \frac{(j-1)d}{b} \rfloor, \ldots, \lfloor \frac{jd}{b} \rfloor$.

Notice that $d^j \geq \frac{d}{10^6 \log(nd\beta)/\alpha^2}$ for every $j \in \{1, \ldots, b\}$. As a result, by applying Theorem 61 and the union bound, the following bounds hold simultaneously for all $j \in \{1, \ldots, b\}$ and $i, i' \in \{1, \ldots, n\}$ with probability $1 - \beta/2$:

$$\|x_i^j - c_{\text{OPT}}^j\| \leq (1 + 0.1\alpha) \cdot \sqrt{\frac{d^j}{d}} \cdot \|x_i - c_{\text{OPT}}\| \leq r^j, \tag{26}$$

$$\|x_i^j - x_{i'}^j\| \leq (1 + 0.1\alpha) \cdot \sqrt{\frac{d^j}{d}} \cdot \|x_i - x_{i'}\| \leq 2r^j, \tag{27}$$

where the last inequality follows from the triangle inequality (through $c_{\text{OPT}}$).

Observe that, when (26) holds, $x_1^j, \ldots, x_n^j \in \mathcal{B}(c_{\text{OPT}}^j, r^j)$. As a result, the accuracy guarantee from Theorem 26 and the union bound implies that the following holds for all $j \in \{1, \ldots, b\}$, with probability $1 - \beta/2$, we have

$$|\{x_1^j, \ldots, x_n^j\} \setminus \mathcal{B}(c, (1 + 0.1\alpha)r^j)| \leq t^j, \tag{28}$$

where $t^j = O_\alpha\left(\frac{d^j}{(\epsilon/b)}\log\left(\frac{1}{(\beta/2b)r^j}\right)\right) = O_\alpha\left(\frac{d}{\epsilon}\cdot\log\left(\frac{d}{\beta r}\right)\right)$. For convenience, let $t^{\max} = \max_{j\in\{1,\ldots,b\}} t^j = O_\alpha\left(\frac{d}{\epsilon}\cdot\log\left(\frac{d}{\beta r}\right)\right)$.

We may assume that $n > t^{\max}$ as otherwise the desired accuracy guarantee holds trivially. When this is the case, we have that $\{x_1^j,\ldots,x_n^j\}\cap\mathcal{B}(c,(1+0.1\alpha)r^j)$ is not empty. From this and from (27), we have

$$\|x_i^j - c^j\| \leq (3+0.1\alpha)r^j \leq 3.1r^j, \tag{29}$$

for all $j\in\{1,\ldots,b\}$ and $i\in\{1,\ldots,n\}$.

To summarize, we have so far shown that (26), (27), (28), and (29) hold simultaneously for all $j\in\{1,\ldots,b\}$ and $i,i'\in\{1,\ldots,n\}$ with probability at least $1-\beta$. We will henceforth assume that this "good" event occurs and show that we have the desired additive error bound, i.e., $|\{x_1,\ldots,x_n\}\setminus\mathcal{B}(c,(1+\alpha)r)| \leq O_\alpha\left(\frac{d}{\epsilon}\cdot\log\left(\frac{d}{\beta r}\right)\right)$.

To prove such a bound, let $\mathbf{X}_{\text{far}} = \{x_1,\ldots,x_n\}\setminus\mathcal{B}(c,(1+\alpha)r)$ and, for every $j\in\{1,\ldots,b\}$, let $\mathbf{X}_{\text{far}}^j = \{x_1^j,\ldots,x_n^j\}\setminus\mathcal{B}(c,(1+0.1\alpha)r^j)$. Notice that, for every input point $x_i$, we have

$$\|x_i - c\|^2 = \|Rx_i - \tilde{c}\|^2$$
$$= \sum_{j\in\{1,\ldots,b\}} \|x_i^j - c^j\|^2$$
$$= \sum_{\substack{j\in\{1,\ldots,b\}\\ x_i\notin\mathbf{X}_{\text{far}}^j}} \|x_i^j - c^j\|^2 + \sum_{\substack{j\in\{1,\ldots,b\}\\ x_i\in\mathbf{X}_{\text{far}}^j}} \|x_i^j - c^j\|^2$$
$$\overset{(29)}{\leq} \sum_{\substack{j\in\{1,\ldots,b\}\\ x_i\notin\mathbf{X}_{\text{far}}^j}} (1+0.1\alpha)^2(r^j)^2 + \sum_{\substack{j\in\{1,\ldots,b\}\\ x_i\in\mathbf{X}_{\text{far}}^j}} (3.1r^j)^2$$
$$\leq (1+0.1\alpha)^4 r^2 + \sum_{\substack{j\in\{1,\ldots,b\}\\ x_i\in\mathbf{X}_{\text{far}}^j}} (3.1r^j)^2, \tag{30}$$

where the last inequality follows from the identity $(r^1)^2 + \cdots + (r^b)^2 = (1+0.1\alpha)^2 r^2$. Notice also that, since $d^j$ is within a factor of 2 of each other, this implies that $r^j \leq (4(1+0.1\alpha)^2 r^2)/b \leq \frac{16r^2}{b}$ for all $j\in\{1,\ldots,b\}$. Plugging this back to (30), we have

$$\|x_i - c\|^2 \leq (1+0.1\alpha)^4 r^2 + \frac{160r^2}{b}\cdot|\{j\in\{1,\ldots,b\} \mid x_i\in\mathbf{X}_{\text{far}}^j\}|$$

Recall that $x_i\in\mathbf{X}_{\text{far}}$ iff $\|x_i - \tilde{c}\| \geq (1+\alpha)r$. Hence, for such $x_i$, we must have

$$|\{j\in\{1,\ldots,b\} \mid x_i\in\mathbf{X}_{\text{far}}^j\}| \geq \frac{b}{160r^2}\cdot\left((1+\alpha)^2 r^2 - (1+0.1\alpha)^4 r^2\right)$$
$$\geq \frac{b\alpha}{160}.$$

Summing the above inequality over all $x_i\in\mathbf{X}_{\text{far}}$, we have

$$\sum_{j\in\{1,\ldots,b\}} |\mathbf{X}_{\text{far}}^j| \geq \frac{b\alpha}{160}\cdot|\mathbf{X}_{\text{far}}|.$$

Recall from (28) that $|\mathbf{X}_{\text{far}}^j| \leq t^{\max}$. Together with the above, we have

$$|\mathbf{X}_{\text{far}}| \leq \frac{160}{b\alpha}\cdot b\cdot t^{\max} = O_\alpha\left(\frac{d}{\epsilon}\cdot\log\left(\frac{d}{\beta r}\right)\right),$$

which concludes our proof. $\qquad\square$

The proof of Lemma 60 is similar, except we use the approximate-DP algorithm for DensestBall (from Theorem 27) as well as advanced composition (Theorem 25).

*Proof of Lemma 60.* We simply run Algorithm 11 with $b = \max\left\{1, \lfloor \frac{d}{10^6 \log(nd/\beta)/\alpha^2} \rfloor\right\}$, and with $\mathcal{A}$ being the algorithm from Theorem 27 that is $(\epsilon', \delta')$-DP with $\epsilon' = \min\left\{1, \frac{\epsilon}{100\sqrt{b\ln(2/\delta)}}\right\}$ and $\delta' = 0.5\delta/b$, has approximation ratio $w = 1 + 0.1\alpha$ and failure probability $\frac{\beta}{2d}$. Since algorithm $\mathcal{A}$ is $(\epsilon', \delta')$-DP and we apply the algorithm $b$ times, the advanced composition theorem (Theorem 25) implies[20] that the entire algorithm is $(\epsilon, \delta)$-DP. The running time analysis is exactly the same as that of Lemma 59.

Finally, the proof of the additive error bound is almost identical to that of Lemma 59, except that here, using Theorem 27 instead of Theorem 26, we have

$$t^j = O_\alpha\left(\frac{d^j}{\epsilon'}\log\left(\frac{n}{\epsilon'\delta' \cdot (0.5\beta/b)}\right)\right)$$

$$= O_\alpha\left(\frac{(d/b)}{\epsilon/\sqrt{b\log(1/\delta)}}\log\left(\frac{n}{\min\{(\epsilon/\sqrt{b\log(1/\delta)}), 1\} \cdot (\delta/b) \cdot (0.5\beta/b)}\right)\right)$$

$$\leq O_\alpha\left(\frac{d}{\sqrt{b}} \cdot \frac{\sqrt{\log(1/\delta)}}{\epsilon} \cdot \log\left(\frac{nd}{\epsilon\delta\beta}\right)\right)$$

$$= O_\alpha\left(\sqrt{d\log(nd/\beta)} \cdot \frac{\sqrt{\log(1/\delta)}}{\epsilon} \cdot \log\left(\frac{nd}{\epsilon\delta\beta}\right)\right)$$

$$= O_\alpha\left(\frac{\sqrt{d}}{\epsilon} \cdot \text{poly}\log\left(\frac{nd}{\epsilon\delta\beta}\right)\right),$$

which results in a similar bound on the additive error for the overall algorithm. $\square$

### D.2.2 From 1-**Center** to **DensestBall** via Dimensionality Reduction

We are now ready to prove the main theorems regarding **DensestBall** (Theorems 62 and 63).

**Theorem 62.** *For every $\epsilon > 0$ and $0 < \alpha, \beta \leq 1$, there exists an $\epsilon$-DP algorithm that runs in $(nd)^{O_\alpha(1)}\text{poly}\log(1/r)$ time and, with probability $1 - \beta$, outputs an $\left(1 + \alpha, O_\alpha\left(\frac{d}{\epsilon} \cdot \log\left(\frac{d}{\beta r}\right)\right)\right)$-approximation for* **DensestBall***.*

**Theorem 63.** *For every $0 < \epsilon \leq O(1)$ and $0 < \delta, \alpha, \beta \leq 1$, there exists an $(\epsilon, \delta)$-DP algorithm that runs in $(nd)^{O_\alpha(1)}\text{poly}\log(1/r)$ time and, with probability $1 - \beta$, solves the* **DensestBall** *problem with approximation ratio $1 + \alpha$ and additive error $O_\alpha\left(\frac{\sqrt{d}}{\epsilon} \cdot \text{poly}\log\left(\frac{nd}{\epsilon\delta\beta}\right)\right)$.*

Note here that Theorems 62 and 63 imply Theorems 6 in Section 3.

With the 1-**Center** algorithm in the previous subsection, the algorithm for **DensestBall** in high dimension follows the same footprint as its counterpart for $(k, p)$-Clustering. The pseudo-code is given below.

---

**Algorithm 12** DensestBall Algorithm (High Dimension).

---

1: **procedure** DENSESTBALLHIGHDIMENSION$_{d'}(x_1, \ldots, x_n; r, \alpha)$
2:      $S \leftarrow$ Random $d'$-dimensional subspace of $\mathbb{R}^d$
3:      **for** $i \in \{1, \ldots, n\}$ **do**
4:          $x_i' \leftarrow$ projection of $x_i$ onto $S$
5:          $r' \leftarrow (1 + 0.1\alpha) \cdot \sqrt{d'/d} \cdot r$
6:      $c' \leftarrow$ DENSESTBALLLOWDIMENSION$(x_1', \ldots, x_n'; r', 0.1\alpha)$
7:      $\mathbf{X}_{\text{cluster}} = \{x_i \mid x_i' \in \mathcal{B}(c', (1 + 0.1\alpha)r')\}$
8:      **return** 1-CENTER$(\mathbf{X}_{\text{cluster}}; (1 + 0.1\alpha)^3 r, 0.1\alpha)$

To prove Theorems 62 and 63, we will also need the following well-known theorem. Its use in our proof below has appeared before in similar context of clustering (see, e.g., [MMR19]).

**Theorem 64** (Kirszbraun Theorem [Kir34]). *Suppose that there exists an $L$-Lipchitz map $\psi$ from $X \subseteq \mathbb{R}^d$ to $\mathbb{R}^{d'}$. Then, there exists an $L$-Lipchitz extension[21] $\tilde{\psi}$ of $\psi$ from $\mathbb{R}^d$ to $\mathbb{R}^{d'}$.*

*Proof of Theorem 62.* We simply run Algorithm 12 where $d' = \min\left\{d, \lceil 10^6 \log(nd/\beta)/\alpha^2\rceil\right\}$, DENSESTBALLLOWDIMENSION is the algorithm from Theorem 26 that is $(\epsilon/2)$-DP, has approximation ratio $w = 1 + 0.1\alpha$ and the failure probability $\frac{\beta}{3}$, and the 1-Center algorithm on Line 8 is the algorithm from Lemma 59 that is $(\epsilon/2)$-DP, has approximation ratio $w = 1 + 0.1\alpha$ and the failure probability $\frac{\beta}{3}$. Basic composition immediately implies that the entire algorithm is $\epsilon$-DP. Furthermore, similar to the proof of Lemma 59, it is also simple to check that the entire algorithm runs in $(nd)^{O_\alpha(1)}\mathrm{poly}\log(1/r)$ time as desired.

We will now argue the accuracy of the algorithm. To do this, let $c_{\mathrm{OPT}}$ be the solution, i.e., the center such that $|\{x_1, \ldots, x_n\} \cap \mathcal{B}(c_{\mathrm{OPT}}, r)|$ is maximized; we let $T = |\{x_1, \ldots, x_n\} \cap \mathcal{B}(c_{\mathrm{OPT}}, r)|$. Moreover, let $c'_{\mathrm{OPT}}$ denote the projection of $c_{\mathrm{OPT}}$ onto $S$.

By applying Theorem 61 and the union bound, the following holds simultaneously for all $j \in \{1, \ldots, t\}$ and $i, i' \in \{1, \ldots, n\}$ with probability $1 - \beta/3$:

$$\|x_i^j - c'_{\mathrm{OPT}}\| \leq (1 + 0.1\alpha) \cdot \sqrt{\frac{d'}{d}} \cdot \|x_i - c_{\mathrm{OPT}}\| \leq r', \tag{31}$$

$$\|x_i' - x_{i'}'\| \approx_{1+0.1\alpha} \sqrt{\frac{d'}{d}} \cdot \|x_i - x_{i'}\|. \tag{32}$$

When (31) holds, $x_1', \ldots, x_n' \in \mathcal{B}(c'_{\mathrm{OPT}}, r')$. As a result, from the accuracy guarantee from Theorem 26, with probability $1 - \beta/3$, we have

$$|\mathbf{X}_{\mathrm{cluster}}| \geq T - O_\alpha\left(\frac{d'}{(\epsilon/2)} \log\left(\frac{1}{(\beta/2)r'}\right)\right) \geq T - O_\alpha\left(\frac{d}{\epsilon} \cdot \log\left(\frac{d}{\beta r}\right)\right). \tag{33}$$

Now, consider the map $\psi : \{x_1', \ldots, x_n'\} \to \mathbb{R}^d$ where $\psi(x_i') = x_i$. From (32), this map is $L$-Lipchitz for $L = (1 + 0.1\alpha)\sqrt{\frac{d}{d'}}$. Thus, from the Kirszbraun Theorem (Theorem 64), there exists an $L$-Lipchitz extension $\tilde{\psi}$ of $\psi$. Consider $\tilde{\psi}(c')$. By the $L$-Lipchitzness of $\tilde{\psi}$, we have

$$\|x_i - \tilde{\psi}(c')\| \leq L \cdot \|x_i' - c'\| \leq (1 + 0.1\alpha)\sqrt{\frac{d}{d'}} \cdot (1 + 0.1\alpha)r' = (1 + 0.1\alpha)^3 r. \tag{34}$$

for all $x_i \in \mathbf{X}_{\mathrm{cluster}}$.

When (34) holds, the accuracy guarantee of Lemma 59 implies that with probability $1 - \beta/3$ the output center $c$ from 1-CENTER, satisfies

$$|\mathcal{B}(c, (1 + 0.1\alpha)^4 r)| \geq |\mathbf{X}_{\mathrm{cluster}}| - O_\alpha\left(\frac{d}{\epsilon} \cdot \log\left(\frac{d}{\beta r}\right)\right). \tag{35}$$

Finally, observe that $(1+0.1\alpha)^4 \leq (1+\alpha)$. Hence, by combining (33) and (35), the algorithm solves the DensestBall problem with approximation ratio $1 + \alpha$ and size error $O_\alpha\left(\frac{d}{\epsilon} \cdot \log\left(\frac{d}{\beta r}\right)\right)$. □

*Proof of Theorem 63.* This proof is exactly the same as that of Theorem 62, except that we use $(\epsilon/2, \delta/2)$-DP algorithms as subroutines (instead of $\epsilon/2$-DP algorithms as before). The size error bounds from Theorem 27 and Lemma 60 can then be used in placed of those from Theorem 26 and Lemma 59, resulting in the new $O_\alpha\left(\frac{\sqrt{d}}{\epsilon} \cdot \mathrm{poly}\log\left(\frac{nd}{\epsilon\delta\beta}\right)\right)$ bound. □

# E From **DensestBall** to 1-**Cluster**

In this section, we prove Theorem 17. We start by formally defining the 1-**Cluster** problem.

**Definition 65** (1-Cluster, e.g., [NSV16]). *Let $n$, $T$ and $t$ be non-negative integers and let $w \geq 1$ be a real number. The input to 1-**Cluster** consists of a subset $S$ of $n \geq T$ points in $\mathbb{B}_\kappa^d$, the discretized d-dimensional unit ball with a minimum discretization step of $\kappa$ per dimension. An algorithm is said to solve the 1-**Cluster** problem with multiplicative approximation $w$, additive error $t$ and probability $1 - \beta$ if it outputs a center $c$ and a radius $r$ such that, with probability at least $1 - \beta$, the ball of radius $r$ centered at $c$ contains at least $T - t$ points in $S$ and $r \leq w \cdot r_{\text{opt}}$ where $r_{\text{opt}}$ is the radius of the smallest ball containing at least $T$ points in $S$.*

*Moreover, we denote by 1-**Cluster** $_{r_{\text{low}},r_{\text{high}}}$ the corresponding promise problem where $r_{\text{opt}}$ is guaranteed to be between $r_{\text{low}}$ and $r_{\text{high}}$ for given $0 < r_{\text{low}} < r_{\text{high}} < 1$.*

Note that for $r_{\text{low}} = \kappa$ and $r_{\text{high}} = 1$ in Definition 65, the 1-**Cluster** $_{r_{\text{low}},r_{\text{high}}}$ problem coincides with the 1-**Cluster** problem without promise.

The following lemma allows us to use our DP algorithm for **DensestBall** in order to obtain a DP algorithm for 1-**Cluster**.

**Lemma 66** (DP Reduction from 1-Cluster $_{r_{\text{low}},r_{\text{high}}}$ to DensestBall). *Let $\epsilon, \delta > 0$. If there is an $(\epsilon, \delta)$-DP algorithm for **DensestBall** with approximation ratio $w$, additive error $t(n, d, w, r, \epsilon, \delta, \beta)$ and running time $\tau(n, d, w, r, \epsilon, \delta, \beta)$, then there is an $(O(\epsilon \cdot \log_w(r_{\text{high}}/r_{\text{low}})), O(\delta \cdot \log_w(r_{\text{high}}/r_{\text{low}})))$-DP algorithm that, with probability at least $1 - O(\beta \log_w(r_{\text{high}}/r_{\text{low}}))$ solves 1-**Cluster** $_{r_{\text{low}},r_{\text{high}}}$ with approximation ratio $w^2$, additive error*

$$\max_{i=0,1,\ldots,\lfloor \log_w(r_{\text{high}}/r_{\text{low}}) \rfloor} t(n, d, w, r/w^i, \epsilon, \delta, \beta) + O\left( \frac{\log_w(r_{\text{high}}/r_{\text{low}}) \log(1/\beta)}{\epsilon} \right)$$

*and running time*

$$\max_{i=0,1,\ldots,\lfloor \log_w(r_{\text{high}}/r_{\text{low}}) \rfloor} \tau(n, d, w, r/w^i, \epsilon, \delta, \beta) \cdot O(\log_w(r_{\text{high}}/r_{\text{low}})) + O(\log(1/\epsilon)).$$

The following theorem follows directly by combining Lemma 66 (with $r_{\text{high}} = 1$ and $r_{\text{low}} = \kappa$) with our pure DP algorithm for **DensestBall** from Theorem 62.

**Theorem 67.** *For every $0 < \epsilon \leq O(1)$ and $0 < \alpha, \beta < 1$, there is an $\epsilon$-DP algorithm that runs in time $(nd)^{O_\alpha(1)} \text{poly} \log(1/\kappa)$ and with probability at least $1 - \beta$, solves 1-**Cluster** with approximation ratio $1 + \alpha$ and additive error $O_\alpha\left( \frac{d}{\epsilon} \log\left( \frac{d}{\beta\kappa} \right) \right)$.*

We now prove Lemma 66.

---

**Algorithm 13** 1-Cluster from DensestBall

---

1: **procedure** 1-CLUSTER($\mathbf{X}$) WITH PARAMETERS $\epsilon, \delta \geq 0$, $\kappa, \beta > 0$, $w > 1$, $\lambda, r_{\text{low}}, r_{\text{high}} > 0$
    AND $0 < t' \leq T$
2:      $r \leftarrow r_{\text{high}}$.
3:      **while** $r \geq r_{\text{low}}$ **do**
4:          $c_1 \leftarrow$ center output by DensestBall $^{\epsilon,\delta,\beta}(\mathbf{X}; r)$
5:          $s_1 \leftarrow |\mathbf{X} \cap \mathcal{B}(c_1, r)| + \text{DLap}(\lambda)$
6:          **if** $s_1 \leq T - t'$ **then**
7:              **return** $\perp$
8:          $c_2 \leftarrow$ center output by DensestBall $^{\epsilon,\delta,\beta}(\mathbf{X}; r/w)$
9:          $s_2 \leftarrow |\mathbf{X} \cap \mathcal{B}(c_2, r/w)| + \text{DLap}(\lambda)$
10:        **if** $s_2 \leq T - t'$ **then**
11:            **return** $(c_1, wr)$
12:        **else**
13:            $r \leftarrow r/w$
14:      **return** $(c_1, r)$

---

*Proof of Lemma 66.* We apply the reduction in Algorithm 13 with $r_{\text{low}}$, $r_{\text{high}}$, $w$, and $T$ set the the values given in the statement of Lemma 66. We also set $\lambda = \frac{1}{\epsilon}$ and $t' = t(n, d, w, \epsilon, \delta, \beta) + O(\frac{\log_w(r_{\text{high}}/r_{\text{low}}) \log(1/\beta)}{\epsilon})$. We now analyze the properties of the resulting algorithm for 1-Cluster. On a high level, this algorithm performs differentially private binary search on the possible values of the ball's radius. In fact, in every iteration of the **while** loop in Algorithm 13, we either return or decrease the radius $r$ by a factor of $w$. Thus, the total number of iterations executed is at most $\lfloor \log_w(r_{\text{high}}/r_{\text{low}}) \rfloor$.

**Privacy.** The DP property directly follows from the setting of $\lambda$, the privacy properties of the DensestBall algorithm, and Basic Composition (i.e., Theorem 24).

**Accuracy.** Denote $t := t(n, d, w, \epsilon, \delta, \beta)$. The standard tail bound for Discrete Laplace random variables implies that the probability that a $\text{DLap}(\lambda)$ random variable has absolute value larger than some $\eta > 0$ is at most $e^{-\Omega(\eta/\lambda)}$. By a union bound, we have that with probability at least $1 - O(\beta \log_w(r_{\text{high}}/r_{\text{low}}))$, all the runs of DensestBall succeed and each of the added $\text{DLap}(\lambda)$ random variables has absolute value at most $O\left( \frac{\log_w(r_{\text{high}}/r_{\text{low}}) \log(1/\beta)}{\epsilon} \right)$ in Algorithm 13. We henceforth condition on this event. In this case, the following holds in each iteration of the **while** loop:

- If there is a ball of radius $r$ that contains at least $T$ of the points in $\mathbf{X}$, then the ball centered at $c_1$ output in line 4 and of radius $wr$ would contain at least $T - t$ points in $\mathbf{X}$. Moreover, the setting of $s_1$ in line 5 will not pass the **if** statement in line 6.

- If there is a ball of radius $r/w$ that contains at least $T$ of the points in $\mathbf{X}$, then the ball centered at $c_2$ output in line 8 and of radius $r$ would contain at least $T - t$ points in $\mathbf{X}$. Moreover, the setting of $s_2$ in line 9 will not pass the **if** statement in line 10.

Put together, these properties imply that the radius output by Algorithm 13 line 14 is at most $w^2 \cdot r_{\text{opt}}$ where $r_{\text{opt}}$ is the radius of the smallest ball containing at least $T$ points in $S$. Moreover, the ball of the output radius around the output center is guaranteed to contain $T - t'$ points in $\mathbf{X}$.

**Running Time.** The running time bound stated in Lemma 66 directly follows from the bound on the number iterations and the facts that in each iteration at most 2 calls to the DensestBall algorithm are made (each with a radius parameter of the form $r/w^i$ for some $i = 0, 1, \ldots, \lfloor \log_w(r_{\text{high}}/r_{\text{low}}) \rfloor$), and that the running time for sampling a Discrete Laplace random variable with parameter $\lambda$ is $O(1 + \log(\lambda))$ [BF13]. $\qquad\square$

We next show that in the case of approximate DP, there is an algorithm with an additive error with better dependence on both the dimension $d$ and the discretization step $\kappa$ per dimension.

**Theorem 68.** *For every $\alpha, \epsilon, \delta, \beta > 0$, $\kappa \in (0, 1)$ and positive integers $n$ and $d$, there is an $(\epsilon, \delta)$-DP algorithm that runs in time $(nd)^{O_\alpha(1)}\text{poly}\log(1/\kappa)$ and solves the 1-**Cluster** problem with approximation ratio $1 + \alpha$ and additive error $O_\alpha\left( \frac{\sqrt{d}}{\epsilon} \cdot \text{poly}\log\left(\frac{nd}{\epsilon\delta\beta}\right) \right) + O\left( \frac{1}{\epsilon} \cdot \log(\frac{1}{\beta\delta}) \cdot 9^{\log^*(d/\kappa)} \right)$.*

On a high level, the improved dependence of the dimension $d$ will follow from the use of our approximate DP algorithm for DensestBall from Theorem 63 (instead of our pure DP algorithm for DensestBall from Theorem 62). On the other hand, the improved dependence of $\kappa$ will be obtained by applying the following algorithm of Nissim et al. [NSV16].

**Theorem 69** ([NSV16]). *For every $\epsilon, \delta, \beta > 0$, $\kappa \in (0, 1)$ and positive integers $n$ and $d$, there is an $(\epsilon, \delta)$-DP algorithm, GoodRadius, that runs in time $poly(n, d, \log(1/\kappa))$ and solves the 1-**Cluster** problem with approximation ratio $w = 4$ and additive error $t = O\left( \frac{1}{\epsilon} \cdot \log(\frac{1}{\beta\delta}) \cdot 9^{\log^*(d/\kappa)} \right)$.*

We are now ready to prove Theorem 68.

*Proof of Theorem 68.* We proceed by first running the GoodRadius algorithm from Theorem 69 to get a radius $r_{\text{approx}}$. If $r_{\text{approx}} = 0$, we run our approximate DP algorithm for DensestBall from

Theorem 63 with $r = 0$, round the resulting center to the closest point in $\mathbb{B}_\kappa^d$, which we then output along with a radius of 0. Otherwise, we apply Lemma 66 with $r_{\text{low}} = r_{\text{approx}}/4$ and $r_{\text{high}} = r_{\text{approx}}$ and with our approximate DP algorithm for DensestBall from Theorem 63.

The privacy of the combined algorithm can be guaranteed by dividing the $(\epsilon, \delta)$-DP budget, e.g., equally among the call to GOODRADIUS and that to Lemma 66 (and ultimately to Theorem 63), and applying Basic Composition (i.e., Theorem 24).

The accuracy follows from the approximation ratio and additive error guarantees of Theorem 69, Lemma 66 and Theorem 63, and by dividing the failure probability $\beta$, e.g., equally among the two algorithms, and then applying the union bound.

The running time is simply the sum of the running times of the two procedures, and can thus be directly bounded using the running time bounds in Theorem 69, Lemma 66 and Theorem 63. □

## F  Sample and Aggregate

This section is devoted to establishing Theorem 18. As mentioned in Section 5.2, one of the basic techniques in DP is the Sample and Aggregate framework of [NRS07]. Consider a universe $\mathcal{U}$ and functions $f : \mathcal{U}^* \to \mathbb{B}_\kappa^d$ mapping databases to points in $\mathbb{B}_\kappa^d$. Intuitively, the premise of the Sample and Aggregate framework is that, for sufficiently large databases $S \in \mathcal{U}^*$, evaluating the function $f$ on a random subsample of $S$ can yield a good approximation to the point $f(S)$. The following definition quantifies how good such approximations are.

**Definition 70** ([NSV16]). *Let $\kappa \in (0, 1)$. Consider a function $f : \mathcal{U}^* \to \mathbb{B}_\kappa^d$ and a database $S \in U^*$. A point $c \in \mathbb{B}_\kappa^d$ is said to be an $(m, r, \zeta)$-stable point of $f$ on $S$ if for $S'$ a database consisting of $m$ i.i.d. samples $S$, it holds that $\Pr[\|f(S') - c\|_2 \leq r] \geq \zeta$. If such a point $c$ exists, the function $f$ is said to be $(m, r, \zeta)$-stable on $S$, and $r$ is said to be a radius of the stable point $c$.*

Nissim et al. [NSV16] obtained the following DP reduction from the problem of finding a stable point of small radius to 1-Cluster.

**Lemma 71** ([NSV16]). *Let $d$ and $n \geq m$ be positive integers, and $\epsilon > 0$ and $0 < \zeta, \beta, \delta < 1$ be real numbers satisfying $\epsilon \leq \zeta/72$ and $\delta \leq \frac{\beta\epsilon}{3}$. If there is an $(\epsilon, \delta)$-DP algorithm for 1-Cluster on $k$ points in $d$ dimensions with approximation ratio $w$, additive error $t$, error probability $\beta/3$, and running time $\tau(k, d, w, \epsilon, \delta, \beta/3)$, then there is an $(\epsilon, \delta)$-DP algorithm that takes as input a function $f : \mathcal{U}^* \to \mathbb{B}_\kappa^d$ along with the parameters $m$, $\zeta$, $\epsilon$, and $\delta$, runs in time $\tau(n/(9m), d, w, \epsilon, \delta, \beta/3)$ plus $O(n/m)$ times the running time for evaluating $f$ on a dataset of size $m$, and whenever $f$ is $(m, r, \zeta)$-stable on $S$, with probability $1 - \beta$, the algorithm outputs an $(m, wr, \frac{\zeta}{8})$-stable point of $f$ on $S$, provided that $n \geq m \cdot O\left(\frac{t}{\zeta} + \frac{1}{\zeta^2} \log\left(\frac{12}{\beta}\right)\right)$.*

By combining Lemma 71 and our Theorem 68, we obtain the following algorithm.

**Theorem 72.** *Let $d$ and $n \geq m$ be positive integers, and $\epsilon > 0$ and $0 < \zeta, \alpha, \beta, \delta, \kappa < 1$ be real numbers satisfying $\epsilon \leq \zeta/72$ and $\delta \leq \frac{\beta\epsilon}{3}$. There is an $(\epsilon, \delta)$-DP algorithm that takes as input a function $f : \mathcal{U}^* \to \mathbb{B}_\kappa^d$ as well as the parameters $m$, $\zeta$, $\epsilon$ and $\delta$, runs in time $(nd/m)^{O_\alpha(1)} \text{poly} \log(1/\kappa)$ plus $O(n/m)$ times the running time for evaluating $f$ on a dataset of size $m$, and whenever $f$ is $(m, r, \zeta)$-stable on $S$, with probability $1 - \beta$, the algorithm outputs an $(m, (1+\alpha)r, \frac{\zeta}{8})$-stable point of $f$ on $S$, provided that $n \geq m \cdot O_\alpha\left(\frac{\sqrt{d}}{\epsilon} \cdot \text{poly} \log\left(\frac{nd}{\epsilon\delta\beta}\right) + \frac{1}{\epsilon} \cdot \log(\frac{1}{\beta\delta}) \cdot 9^{\log^*(d/\kappa)}\right)$.*

We point out that our Theorem 72 obtains a $1 + \alpha$ approximation to the radius (where $\alpha$ is an arbitrarily small positive constant) whereas [NSV16] obtained an approximation ratio of $O(\sqrt{\log n})$, the prior work of [NRS07] had obtained an approximation ratio of $O(\sqrt{d})$, and a constant factor is subsequently implied by [NS18].

## G  Agnostic Learning of Halfspaces with a Margin

In this section, we prove Theorem 20. We start with some definitions.

**Halfspaces.** Let $\mathrm{sgn}(x)$ be equal to $+1$ if $x \geq 0$, and to $-1$ otherwise. A *halfspace* (aka *hyperplane* or *linear threshold function*) is a function $h_{u,\theta}(x) = \mathrm{sgn}(u \cdot x - \theta)$ where $u \in \mathbb{R}^d$ and $\theta \in \mathbb{R}$, and where $u \cdot x = \langle u, x \rangle$ denotes the dot product of the vectors $u$ and $x$. Without loss of generality, we henceforth focus on the case where $\theta = 0$.[22] A halfspace $h_u$ correctly classifies the labeled point $(x, y) \in \mathbb{R}^d \times \{\pm 1\}$ if $h_u(x) = y$.

**Margins.** The *margin* of a point $x$ with respect to a hypothesis $h$ is defined as the largest distance $r$ such that any point of $x$ at distance $r$ is classified in the same class as $x$ by hypothesis $h$. In the special case of a halfspace $h_u(x) = \mathrm{sgn}(u \cdot x)$, the margin of point $x$ is equal to $\frac{|\langle u, x \rangle|}{\|u\| \cdot \|x\|}$.

**Error rates.** For a distribution $D$ on $\mathbb{R}^d \times \{\pm 1\}$,

- the *error rate* of a halfspace $h_u$ on $D$ is defined as $\mathrm{err}^D(u) := \Pr_{(x,y) \sim D}[h(x) \neq y]$,

- for any $\mu > 0$, the $\mu$-*margin error rate* of a halfspace $h_u$ on $D$ is defined as

$$\mathrm{err}_\mu^D(u) := \Pr_{(x,y) \sim D}\left[ y \frac{\langle u, x \rangle}{\|u\| \cdot \|x\|} \leq \mu \right].$$

Furthermore, let $\mathrm{OPT}_\mu^D := \min_{u \in \mathbb{R}^d} \mathrm{err}_\mu^D(u)$. For the ease of notation, we may write $\mathrm{err}^S(u)$ where $S \subseteq \mathbb{R}^d \times \{\pm 1\}$ to denote the error rate on the uniform distribution of $S$; $\mathrm{err}_\mu^S(u)$ is defined similarly.

We study the problem of learning halfspaces with a margin in the agnostic PAC model [Hau92, KSS94], as stated below.

**Definition 73** (Proper Agnostic PAC Learning of Halfspaces with Margin). *Let $d \in \mathbb{N}$, $\beta \in (0, 1)$, and $\mu, t \in \mathbb{R}^+$. An algorithm properly agnostically PAC learns halfspaces with margin $\mu$, error $t$, failure probability $\beta$ and sample complexity $m$, if given as input a training set $S = \{(x^{(i)}, y^{(i)})\}_{i=1}^m$ of i.i.d. samples drawn from an unknown distribution $D$ on $\mathcal{B}(0, 1) \times \{\pm 1\}$, it outputs a halfspace $h_u : \mathbb{R}^d \to \{\pm 1\}$ satisfying $\mathrm{err}^D(u) \leq \mathrm{OPT}_\mu^D + t$ with probability $1 - \beta$.*

When not explicitly stated, we assume that $\beta = 0.01$, it is simple to decrease this failure probability by running the algorithm $\log(1/\beta)$ times and picking the best.

**Related Work.** In the non-private setting, the problem has a long history [BS00, BM02, McA03, SSS09, BS12, DKM19, DKM20]; in fact, the perceptron algorithm [Ros58] is known to PAC learns halfspaces with margin $\mu$ in the realizable case (where $\mathrm{OPT}_\mu^D = 0$) with sample complexity $O_t(1/\gamma^2)$ [Nov62]. In the agnostic setting (where $\mathrm{OPT}_\mu^D$ might not be zero), Ben-David and Simon [BS00] gave an algorithm that uses $O\left(\frac{1}{t^2\gamma^2}\right)$ samples and runs in time $\mathrm{poly}(d) \cdot (1/t)^{O(1/\gamma^2)}$. This is in contrast with the perceptron algorithm, which runs in $\mathrm{poly}(d/t)$ time. It turns out that this is not a coincidence: the agnostic setting is NP-hard even for constant $t > 0$ [BEL03, BS00]. Subsequent works [SSS09, DKM19, DKM20] managed to improve this running time, albeit at certain costs. For example, the algorithm in [SSS09] is *improper*, meaning that it may output a hypothesis that is not a halfspace, and those in [DKM19, DKM20] only guarantee that $\mathrm{err}^D(h_u) \leq (1 + \eta) \cdot \mathrm{OPT}_\mu^D + t$ for an arbitrarily small constant $\eta > 0$.

Nguyen et al. [NUZ20] were the first to study the problem of learning halfspaces with a margin in conjunction with differential privacy. In the realizable setting, they give an $\epsilon$-DP (resp. $(\epsilon, \delta)$-DP) algorithm with running time $(1/t)^{O(1/\gamma^2)} \cdot \mathrm{poly}\left(\frac{d \log(1/\delta)}{\epsilon t}\right)$ (resp. $\mathrm{poly}\left(\frac{d \log(1/\delta)}{\epsilon t}\right)$) and sample complexity $O\left(\mathrm{poly}\left(\frac{1}{\epsilon t \gamma}\right) \cdot \mathrm{poly} \log\left(\frac{1}{\epsilon t \gamma}\right)\right)$ (resp. $O\left(\mathrm{poly}\left(\frac{1}{\epsilon t \gamma}\right) \cdot \mathrm{poly} \log\left(\frac{1}{\epsilon t \delta \gamma}\right)\right)$). Due to the aforementioned NP-hardness of the problem, their efficient $(\epsilon, \delta)$-DP algorithm cannot be extended to the agnostic setting. On the other hand, while not explicitly analyzed in the paper, their $\epsilon$-DP algorithm also works in the agnostic setting with similar running time and sample complexity.

Here, we provide an alternative proof of the agnostic learning result, as stated below. This will be shown via our DensestBall algorithm together with a known connection between DensestBall and learning halfspaces with a margin [BS00, BES02].

**Theorem 74.** *For every $0 < \epsilon \leq O(1)$ and $0 < \beta, \mu, t < 1$, there is an $\epsilon$-DP algorithm that runs in time $\left(\frac{\log(1/\beta)}{\epsilon t}\right)^{O_\mu(1)} + \text{poly}\left(O_\mu\left(\frac{d}{\epsilon t}\right)\right)$, and properly agnostically PAC learns halfspaces with margin $\mu$, error $t$, failure probability $\beta$ and sample complexity $O_\mu\left(\frac{1}{\epsilon t^2} \cdot \text{poly} \log\left(\frac{1}{\epsilon \beta t}\right)\right)$.*

To prove Theorem 74, we will use the following reduction[23]:

**Lemma 75** ([BS00, BES02])**.** *Let $\mu \in (0,1)$ and $\alpha, t > 0$ such that $1 + \alpha < 1/\sqrt{1 - \mu^2}$. There is a polynomial-time transformation that, given as input a set $S = \{(x^{(i)}, y^{(i)})\}_{i=1}^m$ of labeled points, separately transforms each $(x^{(i)}, y^{(i)})$ into a point $z^{(i)}$ in the unit ball such that a solution to DensestBall on the set $\{z^{(i)}\}_{i=1}^m$ with radius $\sqrt{1 - \mu^2}$, approximation ratio $1 + \alpha$ and additive error $t$ yields a halfspace with $\mu'$-margin error rate on $S$ at most $\text{OPT}_\mu^S + \frac{t}{m}$ where $\mu' = \sqrt{1 - (1 - \mu^2)(1 + \alpha)^2}$.*

By combining Lemma 75 and our Theorem 62, we immediately obtain the following:

**Lemma 76.** *For every $\epsilon, \beta > 0$ and $0 < \mu < 1$, there exists an $\epsilon$-DP algorithm that runs in time $(md)^{O_\mu(1)}$, takes as input a set $S = \{(x^{(i)}, y^{(i)})\}_{i=1}^m$ of labeled points, and with probability $1 - \beta$, outputs a halfspace with $\mu'$-margin error rate on $S$ at most $\text{OPT}_\mu^S + \frac{t}{m}$ where $\mu' = \sqrt{1 - (1 - \mu^2)(1 + \alpha)^2}$ and $t = O_\alpha\left(\frac{d}{\epsilon} \cdot \log\left(\frac{d}{\beta}\right)\right)$.*

As is usual in PAC learning results, we will need a generalization bound:

**Lemma 77** (Generalization Bound for Halfspaces with Margin [BM02, McA03])**.** *Let $S = \{(x^{(i)}, y^{(i)})\}_{i=1}^m$ be a multiset of i.i.d. samples from a distribution $D$ on $\mathbb{R}^d \times \{\pm 1\}$, where $m = \Omega(\log(1/\beta)/(t^2 \mu^2))$. Then, with probability $1 - \beta$ over $S$, for all vectors $u \in \mathbb{R}^d$, it holds that $\text{err}^D(u) \leq \text{err}_\mu^{\mathbb{U}(S)}(u) + t$.*

The above lemmas do not yet imply Theorem 74; applying them directly will lead to a sample complexity that depends on $d$. To prove Theorem 74, we will also need the following dimensionality-reduction lemma from [NUZ20] which allows us to focus on the low-dimensional case.

**Lemma 78** (Properties of JL Lemma [NUZ20])**.** *Let $A \in \mathbb{R}^{d' \times d}$ be a random matrix such that $d' = \Theta\left(\frac{\log(1/\beta_{JL})}{\mu^2}\right)$ and $A_{i,j} = \begin{cases} +\frac{1}{\sqrt{d'}} & \text{w.p. } \frac{1}{2} \\ -\frac{1}{\sqrt{d'}} & \text{w.p. } \frac{1}{2} \end{cases}$ independently over $(i,j)$. Let $u \in \mathbb{R}^d$ be a fixed vector. Then, for any $(x, y) \in \mathbb{R}^d \times \{\pm 1\}$ such that $y \cdot \frac{\langle u, x \rangle}{\|u\| \cdot \|x\|} \geq \mu$, we have*

$$\Pr_A\left[y \cdot \frac{\langle Au, Ax \rangle}{\|Au\| \cdot \|Ax\|} > 0.9\mu\right] \geq 1 - 4\beta_{JL}.$$

*Proof of Theorem 74.* Our algorithm works as follows. We first draw a set $S$ of $m$ training samples, and, then apply the JL lemma (with a matrix $A$ sampled as in Lemma 78) in order to project to $d'$ dimensions, where $m, d'$ are to be specified below. Let $S_A$ be the projected training set (i.e., $S_A$ is the multiset of all pairs $(Ax, y)$ where $(x, y) \in S$). We then use the algorithm from Lemma 76 with $\alpha = 0.01\mu^2$ to obtain a halfspace $u' \in \mathbb{R}^{d'}$. Finally, we output $A^T u'$.

We will now prove the algorithm's correctness. Consider any $u^* \in \arg\min_{u \in \mathbb{R}^d} \text{err}_\mu^D(u)$. Let $D'$ denote the distribution of $(x, y) \sim D$ conditioned on $(x, y)$ being correctly classified by $u^*$ with margin at least $\mu$. (Note that $\text{err}_\mu^{D'}(u) = 0$.) Furthermore, let $D_A$ denote the distribution of $(Ax, y)$ where $(x, y) \sim D$, and $D_A'$ denote the distribution of $(Ax, y)$ where $(x, y) \sim D'$.

Let $\beta_{JL} = 0.01t\beta$ and $d' = \Theta\left(\frac{\log(1/\beta_{JL})}{\mu^2}\right) = \Theta\left(\frac{\log(1/(t\beta))}{\mu^2}\right)$ be as in Lemma 78, which implies that $\mathbb{E}_A[\text{err}_{0.9\mu}^{D_A'}(Au^*)] \leq 0.04t\beta$. Hence, by Markov's inequality, we have $\Pr_A[\text{err}_{0.9\mu}^{D_A'}(Au^*) >$

$0.2t] \leq 0.2\beta$. Combining this with the definitions of $u^*$ and $D'$, we have

$$\Pr_A \left[ \mathrm{err}^{D_A}_{0.9\mu}(Au^*) > \mathrm{OPT}^D_\mu + 0.2t \right] \leq 0.2\beta. \tag{36}$$

When $m \geq \Omega(\log(1/\beta)/(t^2\mu^2))$, the Chernoff bound implies that

$$\Pr_S \left[ \mathrm{err}^{S_A}_{0.9\mu}(Au^*) > \mathrm{err}^{D_A}_{0.9\mu}(Au^*) + 0.2t \right] \leq 0.2\beta. \tag{37}$$

Combining (36) and (37), we have

$$\Pr_{A,S} \left[ \mathrm{err}^{S_A}_{0.9\mu}(Au^*) \leq \mathrm{OPT}^D_\mu + 0.4t \right] \geq 1 - 0.4\beta. \tag{38}$$

Lemma 76 then ensures that, with probability $1 - 0.2\beta$, we obtain a halfspace $u' \in \mathbb{R}^{d'}$ satisfying

$$\mathrm{err}^{S_A}_{0.5\mu}(u') \leq \mathrm{err}^{S_A}_{0.9\mu}(Au^*) + t', \tag{39}$$

where $t' = O_\mu \left( \frac{d'}{\epsilon m} \cdot \log \left( \frac{d'}{\beta} \right) \right)$. When we select $m = \Theta_\mu \left( \frac{d'}{\epsilon t} \cdot \log \left( \frac{d'}{\beta} \right) \right) = \Theta_\mu \left( \frac{1}{\epsilon t^2} \cdot \mathrm{poly} \log \left( \frac{1}{\epsilon \beta t} \right) \right)$, we have $t' \leq 0.1t$.

Next, we may apply the generalization bound from Lemma 77, which implies that

$$\Pr_S[\mathrm{err}^{D_A}(u') \leq \mathrm{err}^{S_A}_{0.5\mu}(u') + 0.1t] \geq 1 - 0.2\beta. \tag{40}$$

Using the union bound over (38), (39) and (40), the following holds with probability at least $1 - \beta$:

$$\mathrm{err}^D(A^T u') = \mathrm{err}^{D_A}(u') \leq \mathrm{OPT}^D_\mu + t,$$

which concludes the correctness proof. The claimed running time follows from Lemma 76. □

## H  ClosestPair

In this section, we give our history-independent data structure for ClosestPair (Theorem 22). Before we do so, let us briefly discuss related previous work.

**Related Work.** ClosestPair is among the first problems studied in computational geometry [SH75, BS76, Rab76] and there have been numerous works on lower and upper bounds for the problem since then. Dynamic ClosestPair has also long been studied [Sal91, Smi92, LS92, KS96, Bes98]. To the best of our knowledge, each of these data structures is either history-dependent or has update time $2^{\omega(d)} \cdot \mathrm{poly} \log n$. We will not discuss these results in detail. As alluded to in the main body of the paper, the best known history-independent data structure in the "small dimension" regime is that of Aaronson et al. [ACL+20] whose running time is $d^{O(d)} \mathrm{poly} \log n$. Our result improves the running time to $2^{O(d)} \mathrm{poly} \log n$. We also remark that, due to a result of [KM19], the update time cannot[24] be improved to $2^{o(d)} \mathrm{poly} \log n$ assuming the strong exponential time hypothesis (SETH); in other words, our update time is essentially the best possible.

We finally note that, in the literature, ClosestPair is sometimes referred to the optimization variant, in which we wish to determine $\min_{1 \leq i < j \leq n} \|x_i - x_j\|_2^2$. In the offline setting, the two versions have the same running time complexity to within a factor of $\mathrm{poly}(L)$ (both in the quantum and classical settings) because, to solve the optimization variant, we may use binary search on $\xi$ and apply the algorithm for the decision variant. However, our dynamic data structure (Section H.1) does not naturally extend to the optimization variant and it remains an interesting open question to extend the algorithm to this case.

**Proof Overview.** We will now briefly give an outline of the proof of Theorem 22. Our proof in fact closely follows that of Aaronson et al. [ACL+20]. As such, we will start with the common outline before pointing out the differences. At a high-level, both algorithms partition the space $\mathbb{R}^d$ into small cells $C_1, C_2, \ldots$, each cell having a diameter at most $\sqrt{\xi}$. Two cells $C, C'$ are said to be *adjacent* if there are $x \in C, x' \in C'$ for which $\|x - x'\|_2^2 \leq \xi$. The main observations here are that (i) if there are two points from the same cell, then clearly the answer to ClosestPair is YES and (ii) if no two points are from the same cell, it suffices to check points from adjacent cells. Thus, the algorithm maintains a map from each present cell to the set of points in the cell, and the counter $p_{\leq \xi}$ of the number of points from different cells that are within $\sqrt{\xi}$ in Euclidean distance. A data structure to maintain such a map is known [Amb07, BJLM13] (see Theorem 79). As for $p_{\leq \xi}$, adding/removing a point only requires one to check the cell to which the point belongs, together with the adjacent cells. Thus, the update will be fast, as long as the number of adjacent cells (to each cell) is small.

The first and most important difference between the two algorithms is the choice of the cells. [ACL+20] lets each cell be a $d$-dimensional box of length $\sqrt{\xi/d}$, which results in the number of adjacent cells being $d^{O(d)}$. On the other hand, we use a $(0.5\sqrt{\xi})$-cover from Lemma 29 and let the cells be the Voronoi cells of the cover. It follows from the list size bound at distance $(1.5\sqrt{\xi})$ that the number of adjacent cells is at most $2^{O(d)}$. This indeed corresponds to the speedup seen in our data structure.

A second modification is that, instead of keeping all points in each cell, we just keep their (bit-wise) XOR. The reason behind this is the observation (i) above, which implies that, when there are more than one point in a cell, it does not matter anymore what exactly these points are. This helps simplify our proof; in particular, [ACL+20] needs a different data structure to handle the case where there is more than one solution; however, our data structure works naturally for this case.

There are several details that we have glossed over; the full proof is given in the next section.

## H.1 History-Independent Dynamic Data Structure

As stated in the proof overview above, we will use a history-independent data structure for maintaining a map $M : \{0,1\}^{\ell_k} \to \{0,1\}^{\ell_v}$, where $\ell_k, \ell_v$ are positive integers. In this setting, the map starts of as the trivial map $k \mapsto 0 \ldots 0$. Each update is of the form: set $M[k]$ to $v$, for some $k \in \{0,1\}^{\ell_k}, v \in \{0,1\}^{\ell_v}$. The data structure should support a lookup of $M[k]$ for a given $k$.

Similarly to before, we say that a randomized data structure is history-independent if, for any two sequences of updates that result in the same map, the distributions of the states are the same.

Ambainis [Amb07] gives a history-independent data structure for maintaining a map, based on skip lists. However, this results in probabilistic guarantees on running time. As a result, we will use a different data structure due to [BJLM13] based on radix trees, which has a deterministic guarantee on the running time. (See also [Jef14] for a more detailed description of the data structure.)

**Theorem 79.** *[BJLM13] Let $\ell_k, \ell_v$ be positive integers. There is a history-independent data structure for maintaining a map $M : \{0,1\}^{\ell_k} \to \{0,1\}^{\ell_v}$ for up to $n$ updates, such that each update and lookup takes $\mathrm{poly}(\ell_k, \ell_v)$ time and the required memory is $O(n \cdot \mathrm{poly}(\ell_k, \ell_v))$.*

With the above in mind, we are now ready to prove our main result of this section.

*Proof of Theorem 22.* Let $C := C_{0.5\sqrt{\xi}} \subseteq \mathbb{R}^d$ be the lattice cover from Lemma 29 with $\Delta = 0.5\sqrt{\xi}$. It follows from the construction of Micciancio [Mic04] that every point $c \in C$ satisfies $\frac{3^{d+1}}{\sqrt{\xi}} c \in \mathbb{Z}^n$ (i.e., every coordinate of $c$ is an integer multiple of $\frac{\sqrt{\xi}}{3^{d+1}}$). As a result, we have that every point $c \in C^* := C \cap \mathcal{B}(0, 10\sqrt{d 2^L})$ can be represented as an $\ell_k = \mathrm{poly}(L, d)$ bit integer.

Our data structure maintains a triple $p_{\leq \xi}^{\mathrm{total}}, q^{\mathrm{marked\text{-}cell}}$ and $\mathcal{H}$, where $p_{\leq \xi}^{\mathrm{total}}, q^{\mathrm{marked\text{-}cell}}$ are integers between 0 and $n$ (inclusive) and $\mathcal{H}$ is the data structure from Theorem 79 for maintaining a map $M$ with $\ell_k$ as above and $\ell_v = 2\lceil \log n \rceil + dL$. Each key of $M$ is thought of as an encoding of a point $c$ in the cover $C^*$. Furthermore, each value is a triplet $(n_{count}, p_{\leq \xi}, x_{\oplus})$ where $n_{count}$ is an integer between 0 and $n$ (inclusive), $p_{\leq \xi}$ is an integer between 0 and $n$ (inclusive), and $x_{\oplus}$ is a $dL$-bit string.

Let $\psi : (\mathbb{Z} \cap [0, 2^L])^d \to C$ denote the mapping from $x$ to $\operatorname{argmin}_{c \in C} \|x - c\|_2$ where ties are broken arbitrarily, and let $\mathcal{V}_c := \psi^{-1}(c)$ denote the Voronoi cell of $c$ (with respect to $C$). Observe that $\psi$ can be computed in time $2^{O(d)} \cdot \operatorname{poly}(L)$ using the CVP algorithm from Theorem 32. Furthermore, since $C$ is a $0.5\sqrt{\xi}$ cover, we have that $\|\psi(x) - x\|_2 \le 0.5\sqrt{\xi}$, which implies that $\psi(x) \in C^*$.

For a set $S$ of input points and $c \in C^*$, if $|\mathcal{V}_c \cap S| = 1$, we use $x(c, S)$ to denote the unique element of $\mathcal{V}_c \cap S$. When $S$ is clear from the context, we simply write $x(c)$ as a shorthand for $x(c, S)$.

We will maintain the following invariants for the entire run of the algorithm (where $S$ is the current set of points):

- First, for all $c \in C^*$, $M[c] = (n_{count}, p_{\le \xi}, x_\oplus)$ where the values of $n_{count}, p_{\le \xi}, x_\oplus$ are as follows:

    - $n_{count} = |\mathcal{V}_c \cap S|$,
    - $x_\oplus = \bigoplus_{x \in \mathcal{V}_c \cap S} x$ where each $x \in \mathcal{V}_c \cap S$ is thought of as a $dL$-bit string resulting from concatenating each bit representation of the coordinate,
    - $p_{\le \xi}$ depends on whether $|\mathcal{V}_c \cap S| = 1$. If $|\mathcal{V}_c \cap S| \ne 1$, $p_{\le \xi} = 0$. Otherwise, i.e., if $|\mathcal{V}_c \cap S| = 1$, then $p_{\le \xi} = |\{c' \in C \setminus \{c\} \mid |\mathcal{V}_{c'} \cap S| = 1, \|x(c) - x(c')\|_2^2 \le \xi\}|$, i.e., the number of other cells $c'$ with unique input point $x(c')$ such that $x(c)$ and $x(c')$ are within $\sqrt{\xi}$ in Euclidean distance.

- $q^{\text{marked-cell}}$ is equal to $|\{c \in C^* \mid |\mathcal{V}_c \cap S| \ge 2\}|$.

- $p_{\le \xi}^{\text{total}}$ is equal to $|\{c, c' \in C^* \mid c \ne c', |\mathcal{V}_c \cap S| = |\mathcal{V}_{c'} \cap S| = 1, \|x(c) - x(c')\|_2^2 \le \xi\}|$, i.e., the number of pairs of cells with unique input points such that the corresponding pair of input points are within $\sqrt{\xi}$ in Euclidean distance.

We now describe the operations on the data structure. Throughout, we use the following notation:

$$\Lambda((n_{count}, p_{\le \xi}, x_\oplus), (n'_{count}, p'_{\le \xi}, x'_\oplus)) := \begin{cases} 1 & \text{if } n_{count} = n'_{count} = 1 \text{ and } \|x_\oplus - x'_\oplus\|_2^2 \le \xi, \\ 0 & \text{otherwise.} \end{cases}$$

Note that, when these two states correspond to cells $c$ and $c'$, this is the contribution of $c, c'$ to $p_{\le \xi}^{\text{total}}$. Notice also that $\Lambda$ does not depend on $p_{\le \xi}$ and $p'_{\le \xi}$, but we leave them in the expression for simplicity.

**Lookup.** To determine whether the current point set $S$ contains two distinct points that are at most $\sqrt{\xi}$ apart, we simply check whether $q^{\text{marked-cell}} \ge 1$ or $p_{\le \xi}^{\text{total}} \ge 1$.

**Insert.** To insert a point $x$ into the data structure, we perform the following:

1. Use the algorithm for Closest Vector Problem (Theorem 32) to compute $c = \psi(x)$.

2. Let $(n_{count}^{old}, p_{\le \xi}^{old}, x_\oplus^{old}) = M[c]$.

3. Let $n_{count}^{new} = n_{count}^{old} + 1, p_{\le \xi}^{new} = 0$ and $x_\oplus^{new} = x_\oplus^{old} \oplus x$.

4. Using the list-decoding algorithm (from Lemma 29), compute the set $C_{\text{close}}$ of all $c' \in C$ within distance $2\sqrt{\xi}$ of $c$. Then, for each $c' \in C_{\text{close}}$, do the following:

    (a) Compute $\Lambda^{old} = \Lambda(M[c'], (n_{count}^{old}, p_{\le \xi}^{old}, x_\oplus^{old}))$.
    (b) Compute $\Lambda^{new} = \Lambda(M[c'], (n_{count}^{new}, p_{\le \xi}^{new}, x_\oplus^{new}))$.
    (c) If $\Lambda^{old} - \Lambda^{new} \ne 0$, increase $p_{\le \xi}$ of $M[c']$ by $\Lambda^{old} - \Lambda^{new}$.
    (d) Increase $p_\xi^{new}$ by $\Lambda^{new}$.

5. Update $M[c]$ to $(n_{count}^{new}, p_{\le \xi}^{new}, x_\oplus^{new})$

6. If $n_{count}^{new} = 2$, increase $q^{\text{marked-cell}}$ by one.

**Delete.** To remove a point $x$ from the data structure, we perform the following:

1. Use the algorithm for the Closest Vector Problem (Theorem 32) to compute $c = \psi(x)$.

2. Let $(n_{count}^{old}, p_{\leq \xi}^{old}, x_{\oplus}^{old}) = M[c]$.

3. Let $n_{count}^{new} = n_{count}^{old} - 1, p_{\leq \xi}^{new} = 0$ and $x_{\oplus}^{new} = x_{\oplus}^{old} \oplus x$.

4. Using the list-decoding algorithm (from Lemma 29), compute the set $C_{close}$ of all $c' \in C$ within distance $2\sqrt{\xi}$ of $c$. Then, for each $c' \in C_{close}$, do the following:

    (a) Compute $\Lambda^{old} = \Lambda(M[c'], (n_{count}^{old}, p_{\leq \xi}^{old}, x_{\oplus}^{old}))$.
    (b) Compute $\Lambda^{new} = \Lambda(M[c'], (n_{count}^{new}, p_{\leq \xi}^{new}, x_{\oplus}^{new}))$.
    (c) If $\Lambda^{old} - \Lambda^{new} \neq 0$, increase $p_{\leq \xi}$ of $M[c']$ by $\Lambda^{old} - \Lambda^{new}$.
    (d) Increase $p_{\xi}^{new}$ by $\Lambda^{new}$.

5. Update $M[c]$ to $(n_{count}^{new}, p_{\leq \xi}^{new}, x_{\oplus}^{new})$

6. If $n_{count}^{new} = 1$, decrease $q^{\text{marked-cell}}$ by one.

**Time and memory usage.** It is obvious that a lookup takes $\text{poly}(d, L, \log n)$ time. For an insertion or a deletion, recall that the CVP algorithm and the list-decoding algorithm run in time $2^{O(d)}\text{poly}(L, \log n)$. Furthermore, from the list size bound, $C_{close}$ is of size at most $2^{O(d)}$, which means that we only invoke at most $2^{O(d)}$ lookups and updates of the map $M$. As a result, from the running time guarantee in Theorem 79, we can conclude that the total runtime for each update is only $2^{O(d)}\text{poly}(L, \log n)$.

**Correctness.** It is simple to verify that the claimed invariants hold. Notice also that these invariants completely determine $p_{\leq \xi}^{\text{total}}, q^{\text{marked-cell}}$ and $M$ based on the current point set $S$ alone (regardless of the history). As a result, from the history-independence of $\mathcal{H}$, we can conclude that our data structure is also history-independent. $\qquad\square$

## Footnotes

[11]The claim in [Mic04] states the running time as $d^{O(d)}$. However, this was just because, at the time of publication of [Mic04], only $d^{O(d)}$-time algorithms were known for CVP. By plugging the $2^{O(d)}$-time algorithm for CVP of [MV13] into the first step of the construction in [Mic04], the running time of the construction immediately becomes $2^{O(d)}$.

[12]See also Section 3.6 of [DR14] for a concise description of how [DNR+09] can be applied to Selection.

[13]Note that $1 - e^{-\epsilon/2} \geq 0.5 \min\{1, \epsilon\}$, which implies that $t = O\left(\frac{1}{\epsilon} \log\left(\frac{n\ell}{\min\{\epsilon, 1\} \cdot \delta\beta}\right)\right)$.

[14]Notice that $t = \frac{2}{\epsilon} \ln \left( \frac{2|\mathcal{U}|}{\beta p} \right) = O\left( \frac{1}{\epsilon} \ln \left( \frac{1}{\beta p} \right) \right)$, where the inequality holds because $p \leq 1/|\mathcal{U}|$.

[15]Here we only require the approximation ratio to be some constant for DensestBall, which is fixed to 2 in the algorithm itself.

[16] We assume throughout that $n_j^* > 0$. This is without loss of generality in the case where $n \geq k$. When $n < k$, our DP algorithms can output anything, since the allowed additive errors are larger than $k$.

[17]This holds for any $(\zeta r)$-cover that is also a $\Omega(\zeta r)$-packing. For example, covers described in Section B.1 satisfy this property.

[18]Note that the precise theorem statement in [GLM+10] is only for $k$-median. However, the same argument applies for $(k, p)$-Clustering for any $p \geq 1$.

[19]Specifically, if $k \leq \frac{\log n}{\log \log n}$, it holds that $(\log n)^{O(k)} \leq \text{poly}(n)$; on the other hand, if $k > \frac{\log n}{\log \log n}$, then $(\log n)^{O(k)} \leq k^{O(k)} = 2^{O(k \log k)}$.

[20]Here we use that fact that, since $\epsilon' \leq 1$, we have $e^{\epsilon'} - 1 < 10\epsilon'$.

[21]Recall that $\tilde{\psi}$ is an extension of $\psi$ iff $\tilde{\psi}(x) = \psi(x)$ for all $x \in X$.

[22] As a non-homogeneous halfspace (i.e., one with $\theta \neq 0$) can always be thought of as a homogeneous halfspace (i.e., with $\theta = 0$) with an additional coordinate whose value is $\theta$.

[23]This reduction is implicit in Claim 2.6 and Lemma 4.1 of [BES02].

[24]Specifically, [KM19] shows, assuming SETH, that (offline) ClosestPair cannot be solved in $O(n^{1.499})$ time even for $d = O(\log n)$. If one had a data structure for dynamic ClosestPair with update time $2^{o(d)} \mathrm{poly} \log n$, then one would be able to solve (offline) ClosestPair in $n \cdot 2^{o(d)} \mathrm{poly} \log n = n^{1+o(1)}$ time for $d = O(\log n)$.