[Reviews · NeurIPS 2020]

Review 1

Summary and Contributions: They study the task of differentially private clustering. For several basic clustering problems, authors give efficient differentially private algorithms that achieve essentially the same approximation ratios as those for non-private algorithm, while incurring only small additive errors. This improves upon existing efficient algorithms that only achieve some large constant approximation factors. Results also imply an efficient differentially private algorithm for the agnostic learning of halfpsaces with a margin, as well as an improved algorithm for the sample and Aggregate privacy framework.

Strengths: The algorithms run in polynomial time (in n and d) and obtain tight approximation ratios, which are very good results. The main technical contribution is the implement the Exponential Mechanism more efficiently.

Weaknesses: As authors say, there are still some open questions for this problem, such as to achieve a better additive error. The result expressions for term t and the running time all ignore the dependency for, it is better to give an illustration for about the dependency on alpha, to show the differences to previous works more clearly.

Correctness: Without checking proofs in supplementary materials step by step, the claims in main paper looks correct for me. The methodologies are quite standard and interesting.

Clarity: The paper is well written, but there are still something need to be improved. For example, the clearer illustration between previous works about 1-cluster problem. And why they just ignore the dependence on alpha in Table 1. There are some typos, such as Row 54, the -> these, is->are, complexity->complexities

Relation to Prior Work: It is clearly discussed how this work differs from previous contributions. The works from NSV16 and NS18 are for the 1-cluster problem, the “inverse” of DenseBall. Authors show that the computational complexity of these two problems are essentially the same by the binary search. I’ve some questions about this binary search. Authors don’t show the data universe and the precision of the binary search, then what is the time for this binary search? Intuitively, it will depend on them. Authors are expected to explain that before saying they are the same.

Reproducibility: Yes

Additional Feedback: The feedback answers my question.


Review 2

Summary and Contributions: The paper deals with problems related to differentially private clustering, including DensestBall, k-means, and k-median. The authors obtain differentially private approximation algorithms that improve upon the additive errors of the prior works. The two key contributions of this work include a) an efficient implementation of the list decodable covers which allows one of effectively sample potential candidates from needed by the exponential mechanism (for the case of DensestBall) b) a general framework which allows them to convert any non-private algorithm for k-means and k-median to the differentially private counterparts, with a provably small loss in guarantees. ======= Update ==== Thank you authors for the response. My evaluation remains positive with a recommendation that making this approach practical would be the next key step.

Strengths: This is a classic problem that has received a lot of attention in ML and data mining community. The paper is theoretically quite interesting and well written. The authors clearly identify which parts are novel (for example, efficient implementation of the list-decodable covers). 

Weaknesses: The key drawback is lack of experimental analysis. Most differentially private algorithm suffer from too much noise in practice. What is the key deterrent in not performing such an analysis? 

Correctness: They seem correct, but I have not verified them carefully. No empirical methodology adopted. 

Clarity: Yes, it's quite well written.

Relation to Prior Work: Yes

Reproducibility: Yes

Additional Feedback: The paper is a solid theoretical contribution and I vote for acceptance. I would have loved to see an experimental analysis -- so I am curious, why no experimental analysis? especially for the k-means, there are so many standard implementations, why not pick one and apply the generic framework of the second result?


Review 3

Summary and Contributions: This paper studies differentially private (DP) clustering (under the central notion of DP). The authors give both pure and approximate DP algorithms for the problems of Euclidean k-means and k-median, and in the process, they also develop DP algorithms for Densest Ball and 1-Cluster. The algorithms achieve approximation ratios that are up to (1+a) times the non-private approximation ratios, in polynomial time with respect to the dimension of the space and the size of the input (where the polynomial depends on 1/a). The authors also apply these new results to give DP algorithms for the problems of learning a halfspace classifier with small margin-error in the non-realizable case, and finding a "stable point" of a function f, which is an important primitive for the Sample&Aggregate technique of DP.

Strengths: -This paper improves the state of the art for the DP k-means and k-medians problem. Previous work (NeurIPS 2018) has managed to achieve a constant approximation factor for k-means in polynomial time. This paper achieves an approximation of w*(1+a), where w is the approximation of the best non-private algorithm for the problem, and a similar or slightly improved additive factor, in polynomial time. (Also, note that it is known that the tight approximation ratio can be achieved in exponential time.) -In the process, they give DP algorithms for Densest Ball and 1-Cluster, which are simpler problems, as well as DP sparse selection, and (non-DP) how to find an efficient list-decodable cover. All these could be of independent interest. The authors already apply their results for 1-cluster to finding stable-points (for Sample and Aggregate), and 1-cluster has been useful as a building block in other clustering problems. Densest Ball is also applied to retrieve a DP agnostic learner for halfspaces. -The paper uses and combines a number of techniques, from differential privacy and sketching. The core of the solution is the private algorithm for Densest Ball, which is used in a reduced dimension, which already uses ideas from lattices to create a small enough cover of the space, and a clever sampling scheme to privately choose a center (the DP sparse selection algorithm). Then this is used iteratively to retrieve a coarse centroid set and then a coreset, on which any non-private algorithm can be run. Although private coresets and some of the ideas used have appeared in previous work, the combination of them and several ideas in these proofs are new.

Weaknesses: -Given that a constant approximation ratio algorithm already exists, one could argue that improving it to whatever the constant for the non-private problem is (for k-means and k-medians) is a slightly narrow contribution. However I find that given how clustering is used in practice, this improvement could be significant enough.

Correctness: The paper fully proves all of its claims.

Clarity: The paper is very well written, especially the proofs in the supplementary. The main body gives a good intuition about the proof sketch.

Relation to Prior Work: Prior work is clearly discussed and to the best of my knowledge there are not any omissions.

Reproducibility: Yes

Additional Feedback: I think this is a very well written manuscript. Some low-level/stylistic comments: -I think a more detailed statement of the last known bound from NS18 (approximation ratio, additive error, and running time) might have been useful for the reader. -The citation for NUZ20 has some typos (written as LNUZ+20): Last name of first author is Nguyen, and there is no et al. -Line 31: "except that the distances are not squared in the definition of the clustering cost." I found this a bit misleading for someone who does not know what k-means/median is (ell_2 is not just the square of ell_1 beyond the real line). -Line 47: "On the other hand, without privacy constraints," This makes me think that the previous sentence was regarding the problem with privacy constraints, and it's not. So I would remove "without privacy constraints" from this sentence. -Line 96: "Agnostic learning with a margin" This sentence was confusing to me, until I saw the supplementary. I consider learning with a margin to be on separable data, that is, realizable, by default. I would try to find another title for it. -Line 211: "The gives" -> This gives Supplementary: -I would put all explanations within equations to the right e.g. using \tag{}. (see (triangle inequality) in line 69 and so on) -Please re-state a lemma before proving it and if this is a restated lemma from the main body, include its numbering -Footnote 13: fwhich -> which -Line 420 (above): Shouldn't this be a less than or equal, rather than an equal? -Algorithm 9: APXCLUSTERING -Eq. (23): in the last derivation, I am not sure how eq. (21) has already been applied ======================================== Thank you for your response. The main reason I like this paper is that it uses clear and simple techniques and ideas to approach the problem, and the secondary reason is that I think it lays the foundations for a practical private clustering algorithm in the future. So I vote for its acceptance.


Review 4

Summary and Contributions: This paper considers the problem of clustering algorithms (k-median and k-means) which achieve differential privacy. The point of departure is a solution to the densest ball and related 1-cluster problems. The first step, as usual is dimension reduction via the JL-transform, and then leverage the reduced dimension to allow exponential time algorithms. However, achieving polynomial running time for computing a single ball (thereby improving on the exponential mechanism) still remains surprising difficult, requiring list-decodable covers along with a packing bound. From here, the clustering problems can be solved by also introducing a coreset extraction algorithm, but this step is still intricate. Finally, the authors demonstrate an application to learning halfspaces with margin.

Strengths: The topic is interesting, the results significant, the techniques involved, and the presentation thorough.

Weaknesses: Additive approximation terms, perhaps unavoidable.

Correctness: The proof approach and claimed results are plausible.

Clarity: Overall, yes. It suffers from the problem of trying to stuff a 38 page paper into 8 pages, but does this reasonably well.

Relation to Prior Work: Yes

Reproducibility: Yes

Additional Feedback:

[Author Response · NeurIPS 2020]

We thank the reviewers for their careful reading of the paper and their insightful feedback. Please find below answers to the questions that were raised. For clarity, we sometimes use blue text to quote from the reviews.

**Review #1.**

*R: The result expressions for term $t$ and the running time all ignore the dependency for $\alpha$, it is better to give an illustration for about the dependency on $\alpha$, to show the differences to previous works more clearly.*

We quickly remark that the dependencies on $\alpha$ both in the exponent of the running time and the additive error are all polynomial. We will write out the dependencies more clearly in the revision. Thank you for the suggestion.

*R: Authors show that the computational complexity of these two problems [Densest Ball and 1-Cluster] are essentially the same by the binary search. I've some questions about this binary search. Authors don't show the data universe and the precision of the binary search, then what is the time for this binary search? Intuitively, it will depend on them. Authors are expected to explain that before saying they are the same.*

There are two directions in the equivalence. When reducing from 1-Cluster to Densest Ball, we binary-search on the target radius. In this case, the number of iterations needed for the binary search depends logarithmically on the ratio between the maximum possible distance between two input points and the minimum possible distance between two distinct input points. This is explained in more detail in Appendix F of the Supplementary Material.

Conversely, when reducing from Densest Ball to 1-Cluster, we binary-search on the number of points inside the optimal ball. Here the number of iterations will be logarithmic in the number of input points.

We will add a remark regarding these in the main body of the revision. Thank you for pointing this out.

**Review #2.**

*R: The paper is a solid theoretical contribution and I vote for acceptance. I would have loved to see an experimental analysis – so I am curious, why no experimental analysis? especially for the $k$-means, there are so many standard implementations, why not pick one and apply the generic framework of the second result?*

Thank you for your suggestion. The main issue facing an experimental evaluation of our algorithms is that they would rely on constructions of lattices and solvers for the Closest Vector Problem on lattices. Unfortunately, these are currently not efficient at scales that would be interesting from a practical clustering point of view. Nevertheless, obtaining a differentially private clustering algorithm that is also practical is one of the directions that we are currently pursuing.

**Review #3.**

Thank you very much for the careful reading and helpful comments. We will incorporate the low-level/stylistic suggestions.

*R: Line 420 (above): Shouldn't this be a less than or equal, rather than an equal?*

Yes, the expression before line 420 in the Supplementary Material should indeed be $\leq$ instead of $=$.

*R: Eq. (23): in the last derivation, I am not sure how eq. (21) has already been applied*

The last line in the derivation of (23) should indeed be removed.

**Review #6.**

*R: Additive approximation terms, perhaps unavoidable.*

Yes, the additive approximation term is indeed inevitable under differential privacy requirements. We will emphasize this in the text.

[Meta-Review · NeurIPS 2020]

The paper offers strong theoretical results and is well written. The practicability of the approach is questionable, but this paper gives an important step towards further developments of the topic, including a more practical algorithm.